# Integrating Hi-C and FISH data for modeling of the 3D organization of chromosomes

Ahmed Abbas[1], Xuan He[1], Jing Niu[2], Bin Zhou[3], Guangxiang Zhu[1], Tszshan Ma[2], Jiangpeikun Song[3], Juntao Gao[4], Michael Q. Zhang[5,2,4] & Jianyang Zeng[1,4]

The new advances in various experimental techniques that provide complementary information about the spatial conformations of chromosomes have inspired researchers to develop computational methods to fully exploit the merits of individual data sources and combine them to improve the modeling of chromosome structure. Here we propose GEM-FISH, a method for reconstructing the 3D models of chromosomes through systematically integrating both Hi-C and FISH data with the prior biophysical knowledge of a polymer model. Comprehensive tests on a set of chromosomes, for which both Hi-C and FISH data are available, demonstrate that GEM-FISH can outperform previous chromosome structure modeling methods and accurately capture the higher order spatial features of chromosome conformations. Moreover, our reconstructed 3D models of chromosomes revealed interesting patterns of spatial distributions of super-enhancers which can provide useful insights into understanding the functional roles of these super-enhancers in gene regulation.

[1] Institute for Interdisciplinary Information Sciences, Tsinghua University, Beijing 100084, China. [2] Department of Basic Medical Sciences, School of Medicine, Tsinghua University, Beijing 100084, China. [3] School of Life Science, Tsinghua University, Beijing 100084, China. [4] MOE Key Laboratory of Bioinformatics; Bioinformatics Division, Center for Synthetic and Systems Biology, BNRist; Department of Automation, Tsinghua University; Center for Synthetic and Systems Biology, Tsinghua University, Beijing 100084, China. [5] Department of Biological Sciences, Center for Systems Biology, the University of Texas at Dallas, Richardson, TX 75080-3021, USA. Correspondence and requests for materials should be addressed to J.Z. (email: zengjy321@tsinghua.edu.cn)

etermining the three-dimensional (3D) structure of a chromosome can provide important mechanistic insights into understanding the underlying mechanisms of the 3D folding of the genome and the functional roles of high-order chromatin compaction in gene regulation. For instance, the 3D organization of a chromosome and the spatial proximity of genomic loci can reveal essential relationships between functional elements and their distal targets along the genome sequence, which can shed light on their regulatory functions in controlling gene activities. Recently, the chromosome conformation capture (3C) technique[1], which measures the interaction frequencies between pairs of genomic loci through a proximity ligation strategy, has significantly advanced the studies of higher-order chromatin structure. The extended 3C techniques, such as Hi-C[2] and ChIA-PET[3], have enabled one to study the genome-wide landscape of 3D genome structure at different resolutions (i.e., ranging from Mbps to Kbps) and in various cell types, organisms, and conditions.

Based on their proposed Hi-C technique, Liberman-Aiden et al.[2] discovered that chromosomes are generally partitioned into two compartments, i.e., A and B, which are enriched with active and inactive chromatin marks, respectively. Using Hi-C maps with a resolution in the order of tens of Kbps, several research groups introduced the concept of topologically associated domains (TADs)[4–7], which are defined as the regions that have higher contact frequencies within a domain than across different domains in the Hi-C maps. With Hi-C maps of a relatively high resolution (in a range of 1–5 Kbps), Rao et al.[8] were able to study a finer scale of chromatin structure and investigate the formation of chromatin loops.

Despite the recent significant progress in the studies of higher-order architecture of the genome using the 3C-based techniques, our current understanding on the 3D packing of chromosomes still remains largely incomplete. For example, there still exists a gap in understanding the spatial organizations of the A/B compartments relative to each other in individual chromosomes. Also, if two genomic loci have relatively low contact frequency, it is usually difficult to infer their relative spatial positions only from the Hi-C maps. On the other hand, the fluorescent in situ hybridization (FISH) technique, which measures the spatial distances between a pair of distal genomic loci over a number of cells through a direct imaging strategy, can provide a complementary tool to investigate the 3D organizations of chromosomes.

Wang et al.[9] applied a multiplexed FISH method to study the spatial organizations of TADs and compartments in Chromosomes 20, 21, 22, and X of human diploid (XX) IMR90 cells. They observed that the relation between spatial and genomic distances might deviate from the 1/3 power law expected from the ideal fractal globule model[10], especially when the genomic distance exceeds 7 Mbps. In addition, they found that the A/B compartments are usually arranged in a spatially polarized manner relative to each other with different compartmentalization schemes for the active (ChrXa) and inactive (ChrXi) states of X-Chromosomes. In particular, for ChrXi, the A/B compartments are separated by the DXZ4 macrosatellite, while for ChrXa, these two compartments correspond to the p and q arms.

A large number of computational methods have been developed in the past few years to determine the 3D structures of chromosomes from Hi-C maps[1,11–35]. Many of these methods estimate the pairwise spatial distances between genomic loci using the formula $f \propto 1/d^{\alpha}$, where $f$ and $d$ stand for the contact frequency and the estimated spatial distance between a pair of loci, respectively, and $\alpha$ is a constant. Recently, our group has developed a new manifold learning based approach, called GEM[36], which combines both Hi-C data and conformational energy derived from our current available biophysical knowledge about a 3D polymer model to calculate the 3D structure of a chromosome. GEM does not depend on any specific assumption about the relation between the Hi-C contact frequencies and the corresponding spatial distances, and directly embeds the neighboring proximity from Hi-C space to 3D Euclidean space. Comprehensive comparison tests have demonstrated that GEM can achieve better performance in modeling the 3D structures of chromosomes than other state-of-the-art methods[36].

Despite the recent new advances in FISH techniques[37–40], obtaining a high-resolution pairwise distance map similar to a Hi-C contact map in the same high-throughput manner is still out of reach[41]. On the other hand, the large amount of available FISH data provide an important source of complementary constraints to Hi-C maps for modeling the 3D architectures of chromosomes. However, integrating both Hi-C and FISH data into a unified framework for modeling 3D chromosome structures is not a trivial task, and requires the development of a systematic data integration approach to fully exploit the strengths of individual data types to improve the modeling accuracy. To our best knowledge, no computational approach has been proposed previously to integrate both Hi-C and FISH data for reconstructing the 3D models of chromosomes.

In this paper, we propose a divide-and-conquer based method, called GEM-FISH, which is an extended version of GEM[36] and an attempt to systematically integrate FISH data with both Hi-C data and the prior biophysical knowledge of a polymer model to reconstruct the 3D organizations of chromosomes. GEM-FISH fully exploits the complementary nature of FISH and Hi-C data constraints to improve the modeling process and reveal the finer details of the chromosome packing. In particular, it first uses both Hi-C and FISH data to calculate a TAD-level resolution 3D model of a chromosome and reconstruct the 3D conformations of individual TADs using the intra-TAD interaction frequencies from Hi-C maps and the radii of gyration derived from FISH data. After that, an assembly algorithm is used to integrate the intra-TAD conformations with the TAD-level resolution model to derive the final 3D model of the chromosome. We have demonstrated that GEM-FISH can obtain better 3D models than using Hi-C data only, with more accurate spatial organizations of TADs and compartments in the 3D space. In addition, we have shown that the final 3D models reconstructed by GEM-FISH can also accurately capture the spatial proximity of loop loci, the colocalization of loci belonging to the same subcompartments, and the tendency of expressed genes and interaction sites of the nuclear pore complex (NPC) component Nup153 to lie closer to the chromosome surface. Based on our modeled 3D organizations of chromosomes, we have also found interesting patterns of the spatial distributions of super-enhancers on the three autosomes investigated (i.e., Chrs 20, 21, and 22). This finding can provide useful mechanistic insights into understanding the regulatory roles of super-enhancers in controlling gene activities.

## Results

**Integrating Hi-C and FISH data for 3D chromosome modeling**. We propose a divide-and-conquer based method, called GEM-FISH, to determine the 3D spatial organization of a chromosome through systematically integrating both Hi-C and FISH data. Our framework consists of the following three main steps (Fig. 1). First, we determine the 3D spatial arrangements of individual TADs at TAD-level resolution using both Hi-C and FISH data. Second, we compute the 3D coordinates of genomic loci within individual TADs using a sufficient number of geometric constraints derived from Hi-C data. During the elucidation of both intra-TAD and inter-TAD structures in the previous two steps, we also consider the prior biophysical knowledge of 3D

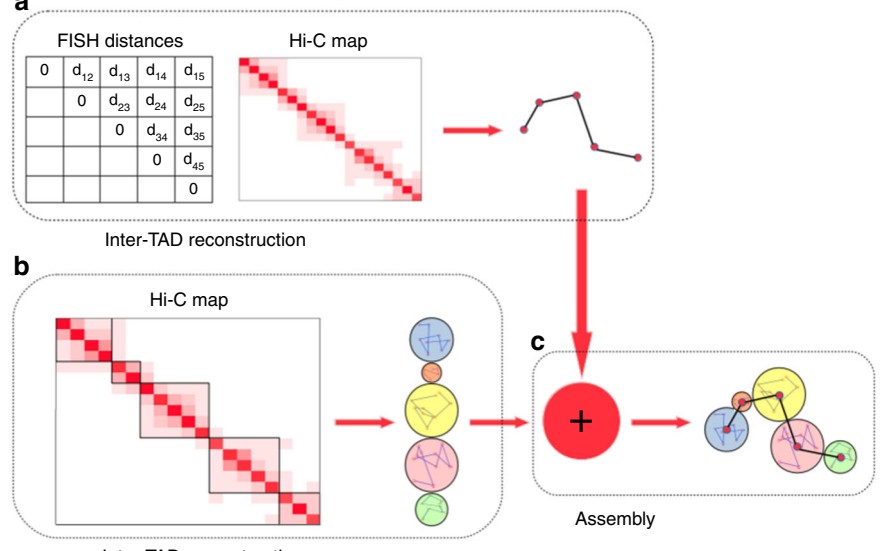

**Fig. 1** The schematic overview of GEM-FISH, which applies a divide-and-conquer strategy to reconstruct the 3D organization of a chromosome by systematically integrating both Hi-C and FISH data. **a** The 3D chromosome model at TAD-level resolution is calculated by integrating Hi-C and FISH data as well as prior biophysical knowledge of a 3D polymer model. **b** The 3D conformations of individual TADs are determined using the intra-TAD geometric restraints derived from the input Hi-C map and prior biophysical knowledge of polymer models. **c** The final complete 3D structure of the chromosome is obtained by assembling the modeling results from the previous two steps, i.e., placing the previously determined intra-TAD conformations into the TAD-level resolution model through translation, rotation, and reflection operations. More details can be found in the main text

polymer models to compute biophysically feasible and structurally stable chromosome structure. In the last step, the modeling results from the previous two steps are assembled through a series of translation, rotation, and reflection operations on individual TAD conformations. More details of the GEM-FISH framework can be found in the "Methods" section.

**GEM-FISH yields more accurate 3D chromosome models**. To evaluate the modeling performance of GEM-FISH, we used it to compute the 3D models of Chromosomes 20, 21, 22, and X of human diploid IMR90 cells for which both Hi-C[8] and FISH[9] data are available. We used 5 Kbp resolution to model the intra-TAD 3D models of chromosomes. For Chromosome X, we calculated both its active and inactive states 3D models, denoted by ChrXa and ChrXi, respectively. We assessed the accuracy of our reconstructed models by measuring the relative error (denoted by RE) in the distance between each pair of TADs, which is defined as,

$$\text{RE}_{ij} = \frac{|d_{ij} - F_{ij}|}{F_{ij}}, \qquad (1)$$

where the term $F_{ij}$ stands for the average distance between TADs $i$ and $j$ obtained from FISH imaging data, and the term $d_{ij}$ denotes the distance between the centers of two TADs $i$ and $j$ in case of the low-resolution (i.e., TAD-level resolution) chromosome model or the average pairwise distance over all pairs of genomic loci between TADs $i$ and $j$ in case of the final complete model.

Figure 2a, b shows the TAD-level resolution model and the corresponding final model of an example chromosome (i.e., Chr21). We calculated the relative errors of both TAD-level resolution and final models resulting from GEM-FISH using both Hi-C and FISH data (Fig. 2c, e). For comparison, we also calculated the relative errors of both TAD-level resolution and final models reconstructed by GEM[36] using only Hi-C data (Fig. 2d, f). We found that the integration of FISH constraints with Hi-C data significantly decreased the relative errors in the spatial distances especially between TADs far away along the

**Table 1 The average relative errors of the TAD-level resolution and final models reconstructed by GEM-FISH using both Hi-C and FISH data, and by GEM using Hi-C data only for Chrs 20, 21, 22, Xa, and Xi**

|  | TAD-level model (GEM-FISH) | Final model (GEM-FISH) | TAD-level model (GEM) | Final model (GEM) |
|---|---|---|---|---|
| Chr20 | 0.18 | 0.16 | 0.31 | 0.30 |
| Chr21 | 0.16 | 0.14 | 0.31 | 0.30 |
| Chr22 | 0.17 | 0.16 | 0.22 | 0.20 |
| ChrXa | 0.17 | 0.16 | 1.36 | 1.37 |
| ChrXi | 0.22 | 0.21 | 1.92 | 1.93 |

In GEM, due to the availability of only the non-allele-specific Hi-C maps (which do not distinguish between ChrXa and ChrXi) for the IMR90 cell line[8], we only considered one 3D model for ChrX

genome that usually have relatively low Hi-C contact frequencies (Fig. 2c, d). In addition, when compared to the TAD-level resolution models, the relative errors in the spatial distances between adjacent TADs along the genome slightly decreased in the final model (see the diagonal elements in Fig. 2e, f), which basically indicated that our method was able to compute the correct relative orientations of adjacent TADs. Table 1 summarizes the relative errors obtained for all five tested chromosomes, and Supplementary Figs. 1–4 show the corresponding modeling results for Chrs 20, 22, Xa, and Xi, respectively. All these results indicated that incorporating FISH data can significantly improve the accuracy of modeling the 3D organizations of chromosomes.

In addition, we conducted more validation tests to further evaluate the reasonableness of the 3D models calculated by GEM-FISH. More specifically, we compared the curves of the spatial vs. genomic distances between TADs for the 3D models reconstructed by both GEM-FISH and GEM with the corresponding curves derived from the FISH experimental data[9], conducted an additional 10-fold cross-validation procedure, measured the

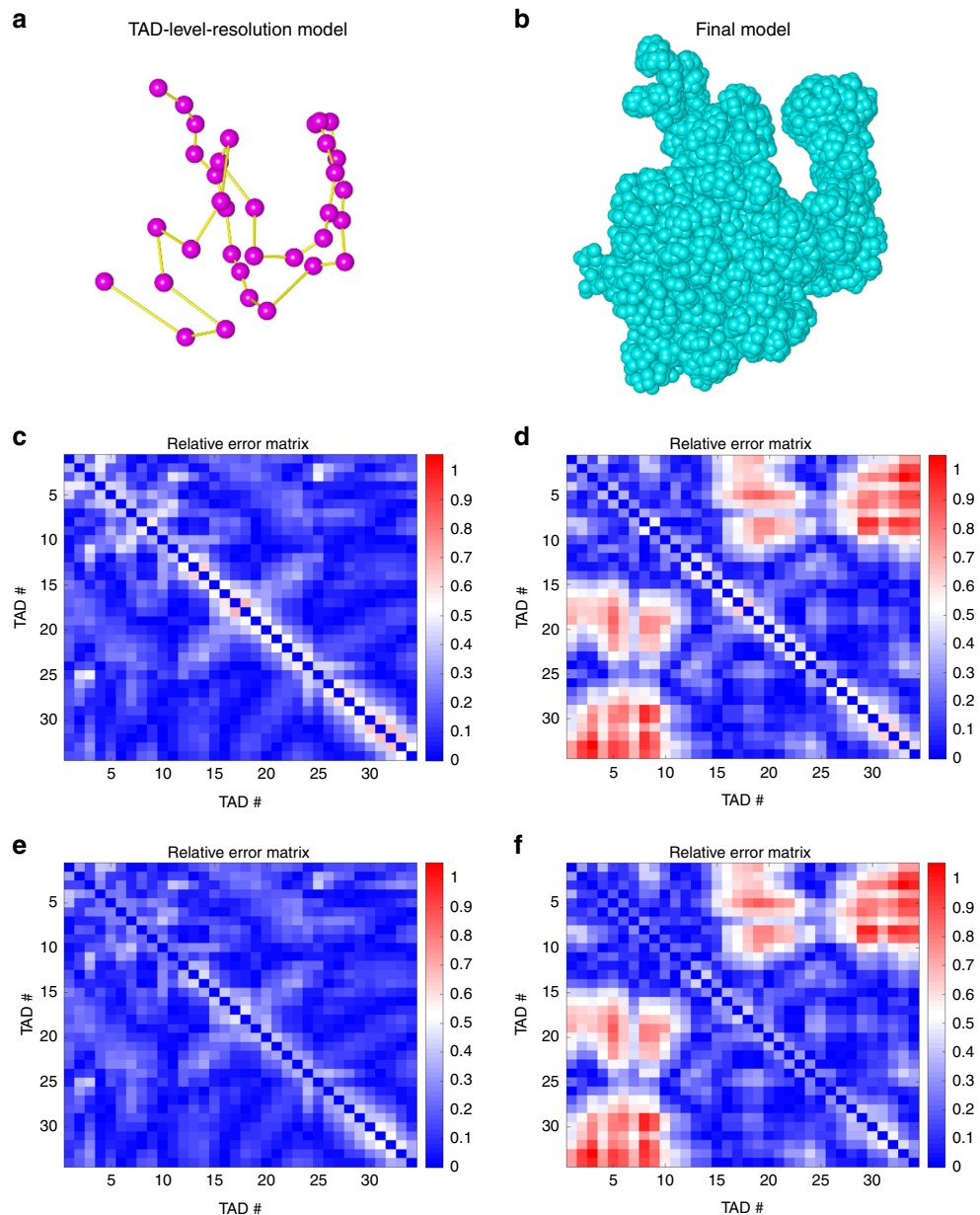

**Fig. 2** The modeling results of human Chromosome 21 (Chr21). **a** The TAD-level resolution 3D structure of Chr21 calculated by GEM-FISH, where each dot represents the center of a TAD. **b** The final 3D structure of Chr21 reconstructed by GEM-FISH. The visualization in (**a**) and (**b**) was performed using UCSF Chimera[61]. **c**, **d** The relative error matrices of the TAD-level resolution models computed by GEM-FISH using both Hi-C and FISH data, and by GEM using only Hi-C data, respectively. **e**, **f** The relative error matrices of the final models computed by GEM-FISH using both Hi-C and FISH data, and GEM using only Hi-C data, respectively

deviation of the reconstructed 3D models from being spherical (hereafter referred to as asphericity values), and inspected the radial distributions of the expressed genes and interaction sites of the NPC component Nup153 in the final 3D models reconstructed by GEM-FISH (see Supplementary Note 1 for details). Moreover, we demonstrated the necessity of incorporating the Hi-C data constraints in GEM-FISH to ensure the consistency between the reconstructed 3D models with both Hi-C and FISH data (see Supplementary Note 2). The results of all these additional validation tests further supported the superiority of our modeling approach.

**GEM-FISH yields accurate compartment partitioning**. It has been widely observed that chromosomes are partitioned into two compartments (i.e., A and B) based on Hi-C maps[2]. TADs within the same compartment are generally spatially closer to each other and have higher contact frequencies than those belonging to different compartments. Using FISH experiments, Wang et al.[9] observed that the two compartments are typically spatially arranged in a polarized fashion relative to each other. In addition, they showed that individual compartments are relatively enriched with different epigenetic marks.

Following the same strategy as in ref. [9], we assigned TADs to A/B compartments for the 3D chromosome models calculated by GEM-FISH (using both Hi-C and FISH data) and GEM[36] (using only Hi-C data). For the examined autosomes (i.e., Chrs 20, 21, and 22), the average accuracy of assigning TADs of the 3D models reconstructed by GEM-FISH to A/B compartments was 89.6% vs. 81.0% for those models calculated by GEM (Table 2,

Fig. 3). We found that the 3D models computed by GEM-FISH displayed approximately similar relative enrichment patterns of different epigenetic marks in A/B compartments for the three autosomes, which were close to those derived from experimental FISH data (Supplementary Fig. 5). This observation indicated that the few TADs that were wrongly assigned to the A and B compartments in the models reconstructed by GEM-FISH probably had more noisy epigenetic properties of one compartment over the other. On the other hand, the difference was

relatively more obvious for the 3D model of Chr20 calculated by GEM (Supplementary Fig. 5a). This was likely due to the relatively low accuracy in assigning TADs to A/B compartments on this model when using only Hi-C data to reconstruct its 3D chromosome structure (73.3%, Table 2).

The final 3D models calculated by GEM-FISH for ChrXa and ChrXi were clearly different and can be easily distinguished through visual inspection (Fig. 4a, b). The 3D model of ChrXi was notably more compact compared to that of ChrXa, which was consistent with its inactive nature. The quantitative comparison of the compactness of the 3D models of ChrXi and ChrXa showed that the densities of the TADs of ChrXi were significantly higher than those of ChrXa (Fig. 4c).

Although Wang et al.[9] observed that the X-Chromosome can be partitioned into two compartments, they found that the compartmentalization scheme was different for its active state ChrXa and inactive state ChrXi. The two compartments of ChrXa corresponded to its p and q arms. For ChrXi, there were two continuous compartments separated along the genomic sequence by the DXZ4 macrosatellite. Here, the 3D models calculated by GEM-FISH (Fig. 4a, b) resulted in 97.5% and 92.5% accuracy in assigning TADs to the two compartments for ChrXa and ChrXi, respectively (Table 2). More importantly, in the 3D models derived from GEM-FISH, the separation position between the two compartments was correctly captured for both ChrXa and ChrXi (Fig. 4d, e). On the other hand, since so far only the non-

**Table 2 The accuracy of assigning TADs of the 3D chromosome models calculated by GEM-FISH (which uses both Hi-C and FISH data) and GEM (which uses only Hi-C data) to the two different compartments**

|       | GEM-FISH | GEM   |
|-------|----------|-------|
| Chr20 | 28/30    | 22/30 |
| Chr21 | 32/34    | 30/34 |
| Chr22 | 22/27    | 22/27 |
| ChrXa | 39/40    | 35/40 |
| ChrXi | 37/40    | 24/40 |

The table entries are filled with the format of "number of correctly assigned TADs/total number of TADs". In GEM, due to the availability of only the non-allele-specific Hi-C maps (which do not distinguish between ChrXa and ChrXi) for the IMR90 cell line[8], we only considered one 3D model for ChrX

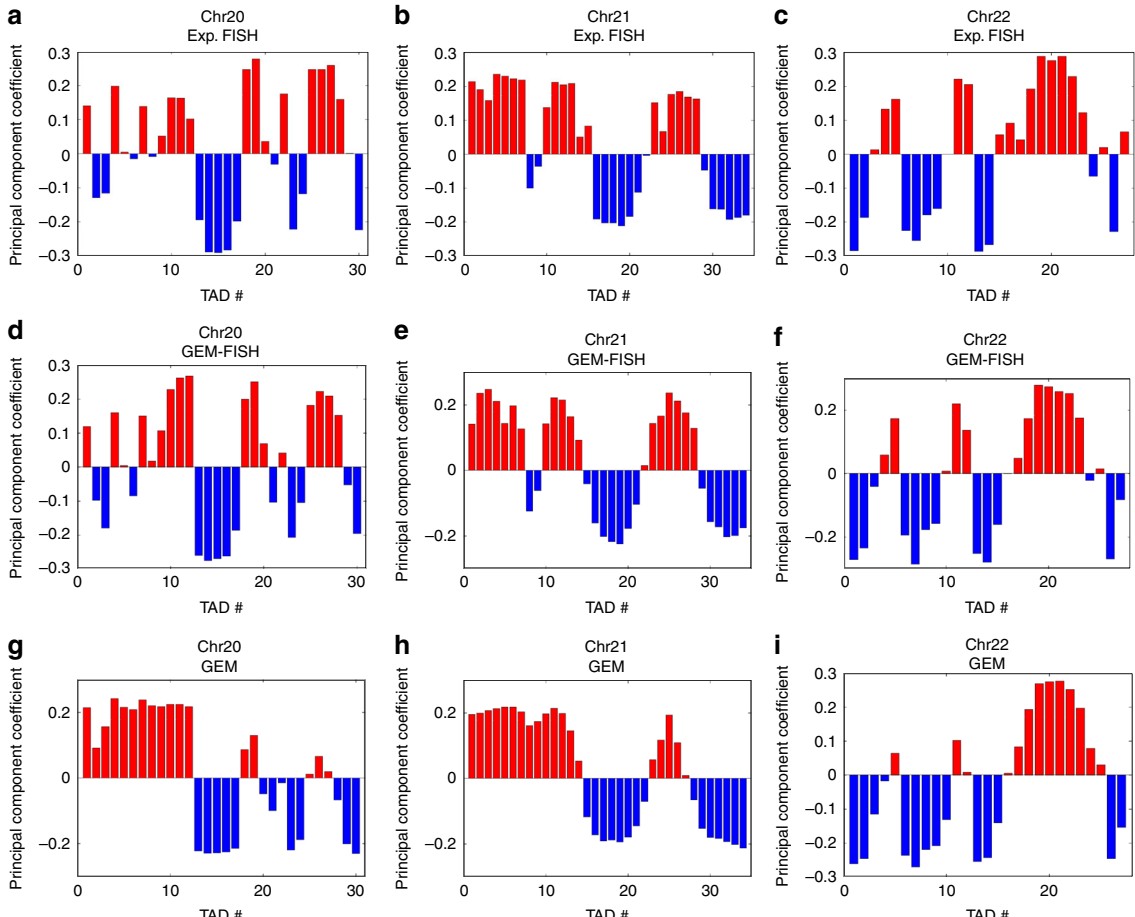

**Fig. 3** Assignment of TADs to A/B compartments for Chrs 20, 21, and 22. **a–c** Assignment of TADs for Chrs 20 (**a**), 21 (**b**), and 22 (**c**) using the experimental FISH data (obtained from ref. [9]). **d–f** Assignment of TADs of the 3D chromosome models calculated by GEM-FISH using both Hi-C and FISH data for Chrs 20 (**d**), 21 (**e**), and 22 (**f**). **g–i** Assignment of TADs of the 3D chromosome models calculated by GEM using only Hi-C data for Chrs 20 (**g**), 21 (**h**), and 22 (**i**)

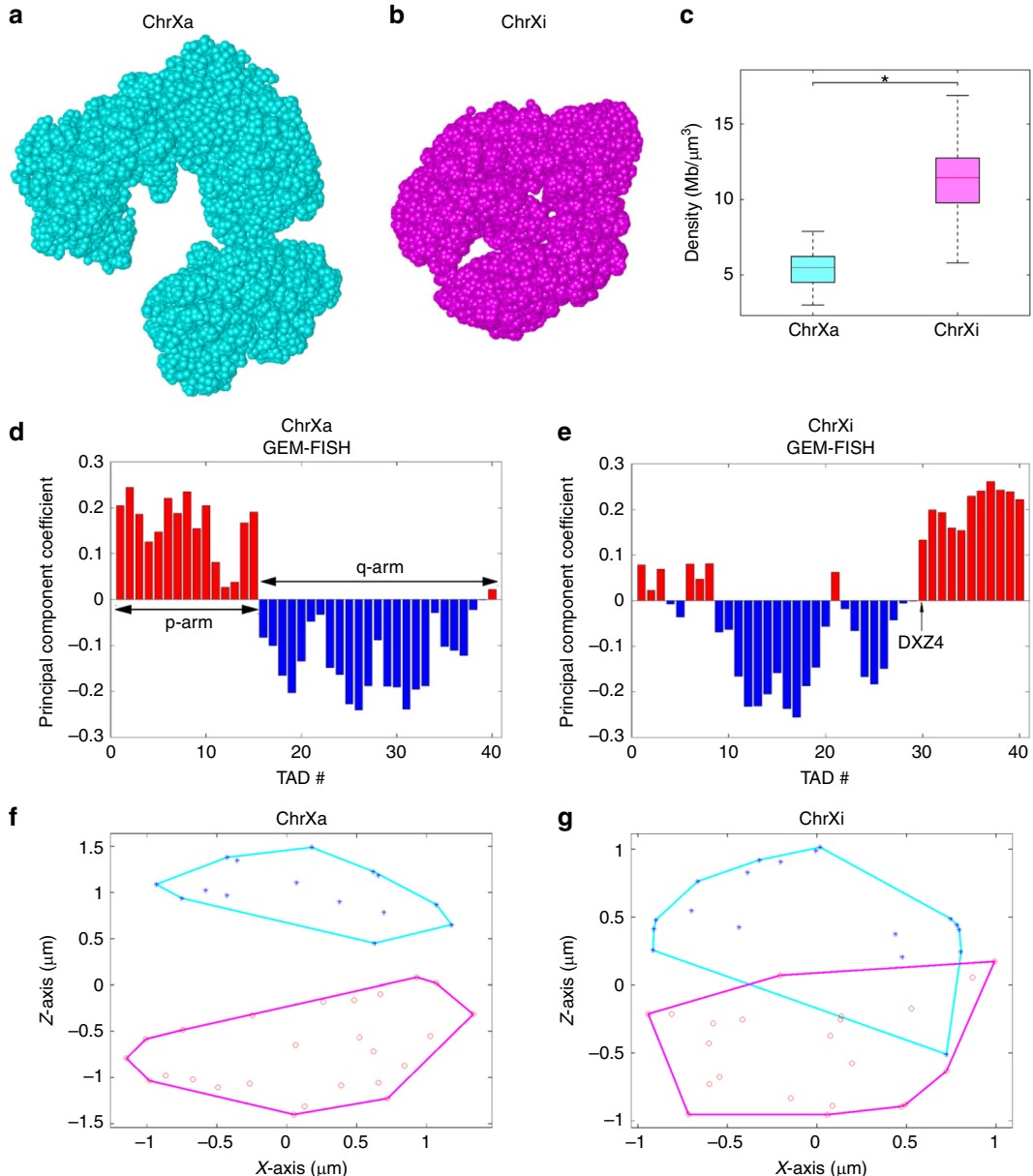

**Fig. 4** The modeling results of GEM-FISH for the human X-Chromosome (including both active state ChrXa and inactive state ChrXi). **a**, **b** Visualization of the final 3D models of ChrXa (**a**) and ChrXi (**b**) in UCSF Chimera[61]. (**c**) Comparison of the compactness of TADs between the 3D models of active (ChrXa) and inactive (ChrXi) states of human X-Chromosome. $N = 40$ TADs for both Chrs Xa and Xi. *$p$-value $< 10^{-13}$, one-tailed Wilcoxon rank-sum test. **d**, **e** The assignment of TADs of ChrXa (**d**) and ChrXi (**e**) to the A/B compartments. **f**, **g** Projection of the 3D convex hull plots of the A and B compartments of ChrXa (**f**) and ChrXi (**g**) to the XZ plane. For the boxplots, the top and bottom lines of each box represent the 75th and 25th percentiles of the samples, respectively. The line inside each box represents the median of the samples. The upper and lower lines above and below the boxes are the whiskers

allele-specific Hi-C maps (which do not distinguish between ChrXa and ChrXi) were available for the IMR90 cell line[8] (to our best knowledge), the 3D models calculated by GEM (which only takes Hi-C data as input) resulted in lower accuracy in assigning their TADs to the two compartments (87.5% and 60.0% for ChrXa and ChrXi, respectively). The significant improvement in the accuracy of compartmentalization after integrating both Hi-C and FISH data into the modeling process especially for the X-Chromosomes demonstrated the main advantage of GEM-FISH in fully exploiting the available FISH data to derive more accurate higher-order arrangements of sub-chromosomal regions.

We also examined how the two compartments of the X-Chromosomes were spatially arranged relative to each other. Our modeling results from GEM-FISH showed that the two compartments of all the obtained 3D chromosome models were placed in a polarized fashion relative to each other, with the degree of polarization in ChrXi less than that in ChrXa, which was consistent with the previous finding in the previous study[9] (Fig. 4f, g, and Supplementary Fig. 6). In addition, we found that the pairwise spatial distances between TADs within the same compartment were significantly smaller than those between TADs across different compartments for all five tested chromosomes (Supplementary Fig. 7), which was also consistent with the previous result[2].

**GEM-FISH outperforms the state-of-the-art methods.** We further benchmarked GEM-FISH against several state-of-the-art

methods in chromosome structure modeling. We first calculated the 3D models of the five chromosomes using classical multi-dimensional scaling (MDS)[42] with the average FISH distances as input. In addition, we reconstructed the 3D models of Chrs 20, 21, 22, and X using Shrec3D[18], chromosome3D[35], and ChromSDE[43] with the TAD-level resolution Hi-C maps as input. We then compared the quality of the reconstructed 3D models using these four methods vs. that of the 3D models reconstructed by GEM-FISH with respect to the accuracy in assigning TADs to the A/B compartments. For the 3D models reconstructed by classical MDS, we also compared them with those reconstructed by GEM-FISH in terms of the average relative errors with respect to the pairwise FISH distances. For the sake of fair comparison, we did not calculate this metric (i.e., average relative error) for those 3D models reconstructed by Shrec3D, chromosome3D, and ChromSDE. We found that GEM-FISH outperformed these four baseline methods, in terms of the accuracy of TAD assignment to A/B compartments, and surpassed classical MDS with respect to the average relative errors (Supplementary Table 1). We also calculated the asphericity values for the 3D models reconstructed by these baseline methods and compared them to the results derived from GEM-FISH. The high asphericity values for the 3D models obtained by Shrec3D, chromosome3D, and ChromSDE indicated that these conformations generally had an extended shape rather than a spherical one (Supplementary Table 1), probably due to the relatively weak Hi-C contact signals between those TADs far away along the genomic distances.

**Analyses of subcompartments**. It has been observed that there are at least six nuclear subcompartments defined based on their long-range interaction patterns[8]. Two subcompartments associate with compartment A, hence called A1 and A2, and the other four subcompartments associate with compartment B, hence called B1, B2, B3, and B4. The genomic loci belonging to different sub-compartments tend to exhibit distinct genomic and epigenomic properties. For instance, those loci belonging to subcompartments A1 and A2 are enriched with activating chromatin marks, such as H3K27ac, H3K36me3, H3K4me1, and H3K79me2[8]. On the other hand, the loci belonging to subcompartment B1 correlate positively with the repressive mark H3K27me3 and negatively with the activating mark H3K36me3, while the loci belonging to subcompartment B2 lack all the marks mentioned above[8].

Here we annotated the loci that carry the epigenomic content of subcompartments B1 or B2 and investigated their folding properties in the 3D models reconstructed by GEM-FISH (see Supplementary Note 3). We mainly considered the subcompart-ment types B1 and B2, particularly because they are expected to have different folding properties due to their distinct epigenomic content. For instance, the genomic regions belonging to subcompartment B1 with a repressive nature are expected to be more densely packed than those belonging to subcompartment B2 with an inactive nature[38]. We found that the genomic loci belonging to a certain subcompartment (B1 or B2) tend to colocalize in the 3D models reconstructed by GEM-FISH even if they are far away along the genomic distances (Fig. 5a, c, e, and Supplementary Figs. 8–13), which agreed well with the previous finding[8]. We also calculated the densities of regions belonging to the two subcompartments in the reconstructed 3D models and found that the densities of the regions from subcompartment B1 were significantly higher than those of the regions from subcompartment B2 (Fig. 5b, d, f). Such a finding was also consistent with the previous result[38].

In addition, we conducted a new FISH experiment to validate the relative spatial distances between a triplet of genomic loci and showed that loci belonging to the same subcompartment tend to lie spatially closer to each other than those loci belonging to different subcompartments even at a larger genomic distance. In particular, we designed probes for three genomic loci, denoted by L1, L2, and L3, respectively, where the genomic distance between L1 and L2 is larger than that between L1 and L3 (Fig. 6a). The two loci L1 and L2 are depleted from the marks H3K27me3, H3K36me3, H3K27ac, H3K4me1, and H3K79me2, and hence considered belonging to subcompartment B2[8]. On the other hand, the locus L3 is enriched with H3K27me3 and depleted from H3K36me3, and hence considered belonging to subcompartment B1[8] (Fig. 6a). We examined the spatial distances between L1 and L2 and between L1 and L3 derived from this new FISH experiment, and found that the spatial distance between the two loci L1 and L2 is consistently smaller than that between L1 and L3 (Fig. 6b, c). This observation was consistent with the modeling results obtained by GEM-FISH, and also provided an experimental evidence to show that the loci belonging to the same subcompartment tend to locate spatially closer to each other.

**Analyses of the final 3D models derived from GEM-FISH**. Next, we further analyzed the details of the final 3D models derived from GEM-FISH. We first compared the packing densities of the regions between loop anchor loci with those between control loci, i.e., loci that do not form anchor points of a loop (see Supple-mentary Note 4). We found that the DNA packing densities of the regions between loop anchor loci were significantly higher than those of the regions between control loci for all five tested chromosomes (Supplementary Fig. 14), which provided another evidence to support the reasonableness of the 3D models recon-structed by GEM-FISH.

In addition, we investigated the 3D conformations of individual TADs that strongly carry the epigenetic nature of A or B compartments (see Supplementary Note 5). We found that the TADs that carry the inactive nature of compartment B tend to have compact 3D models, while the 3D models of TADs that strongly have the active nature of compartment A tend to be more open (Supplementary Fig. 15), which was consistent with previous findings[38].

We also examined the amount of overlapping between adjacent TADs in the final 3D models reconstructed by GEM-FISH (see Supplementary Note 6). We observed that adjacent TADs along the genomic distance that belong to the same compartment tend to display higher overlapping than those that belong to different compartments (Supplementary Table 2), which was also con-sistent with the previous findings[38].

**Analysis of the spatial distributions of super-enhancers**. Super-enhancers are a group of enhancers that are in close genomic proximity and span a genomic interval in a range of tens of kilobase pairs. A key feature that distinguishes super-enhancers from common enhancer is the relatively high enrichment of specific transcription coactivators such as mediator Med1 or activating histone marks such as H3K27ac. They are usually found close to the cell-type-specific genes that define the cell identity and regulate their expression[44,45].

We first obtained the positions of super-enhancers of Chrs 20, 21, and 22 from the super-enhancer database dbSUPER[44], and then examined their locations in the final 3D models calculated by GEM-FISH. We found that super-enhancers tend to lie on the surfaces of the reconstructed 3D chromosome models (Fig. 7 and Supplementary Fig. 16), which was consistent with the previous finding that active gene regions tend to lie on the surface of the chromosome territory[46,47].

In addition, for Chr21, we found that four of its five super-enhancers lie in the G-band q22.3. After closely examining this

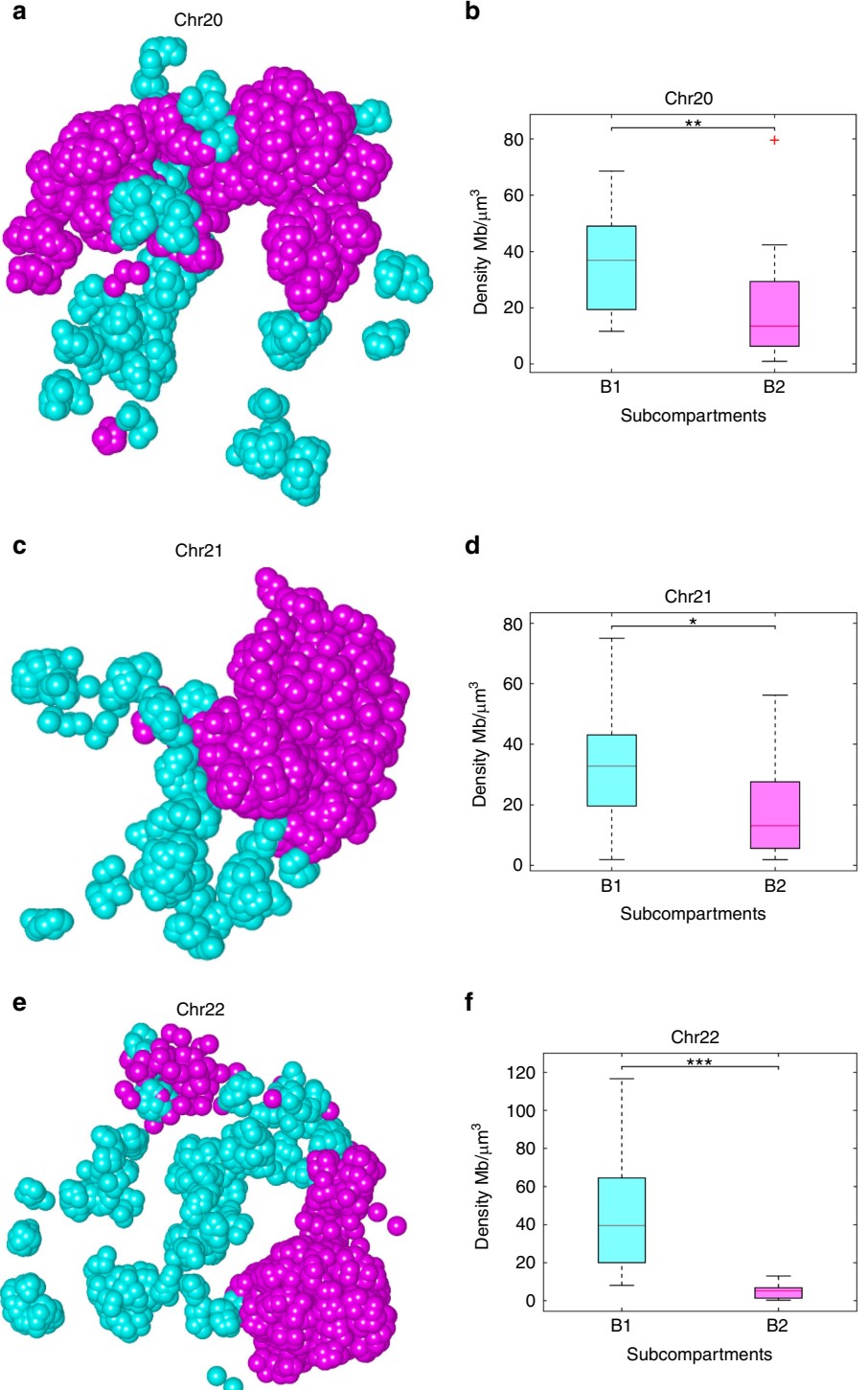

**Fig. 5** Regions belonging to the same subcompartment tend to colocalize in the 3D space. **a**, **c**, **e** Visualization of the regions belonging to subcompartments B1 (cyan) and B2 (magenta) in the 5 Kbps-resolution 3D models reconstructed by GEM-FISH for Chr20, Chr21, and Chr22, respectively. Only regions that belong to either B1 or B2 are shown. The visualization was performed using UCSF Chimera[61]. **b**, **d**, **f** Boxplots on the densities of regions belonging to subcompartments B1 and B2 for Chr20, Chr21, and Chr22, respectively. $N_{B1} = 39, 20, 44$ genomic segments belonging to subcompartment B1 for Chrs 20, 21, and 22, respectively. $N_{B2} = 23, 25, 9$ genomic segments belonging to subcompartment B2 for Chrs 20, 21, and 22, respectively. *p-value < 0.007, **p-value < $10^{-4}$, ***p-value < $10^{-4}$. All tests were performed using the one-tailed Wilcoxon rank-sum test. For the boxplots, the top and bottom lines of each box represent the 75th and 25th percentiles of the samples, respectively. The line inside each box represents the median of the samples. The upper and lower lines above and below the boxes are the whiskers. Red points marked by '+' represent outliers, which represent the observations beyond 1.5 times interquartile range away from the top or the bottom of the box

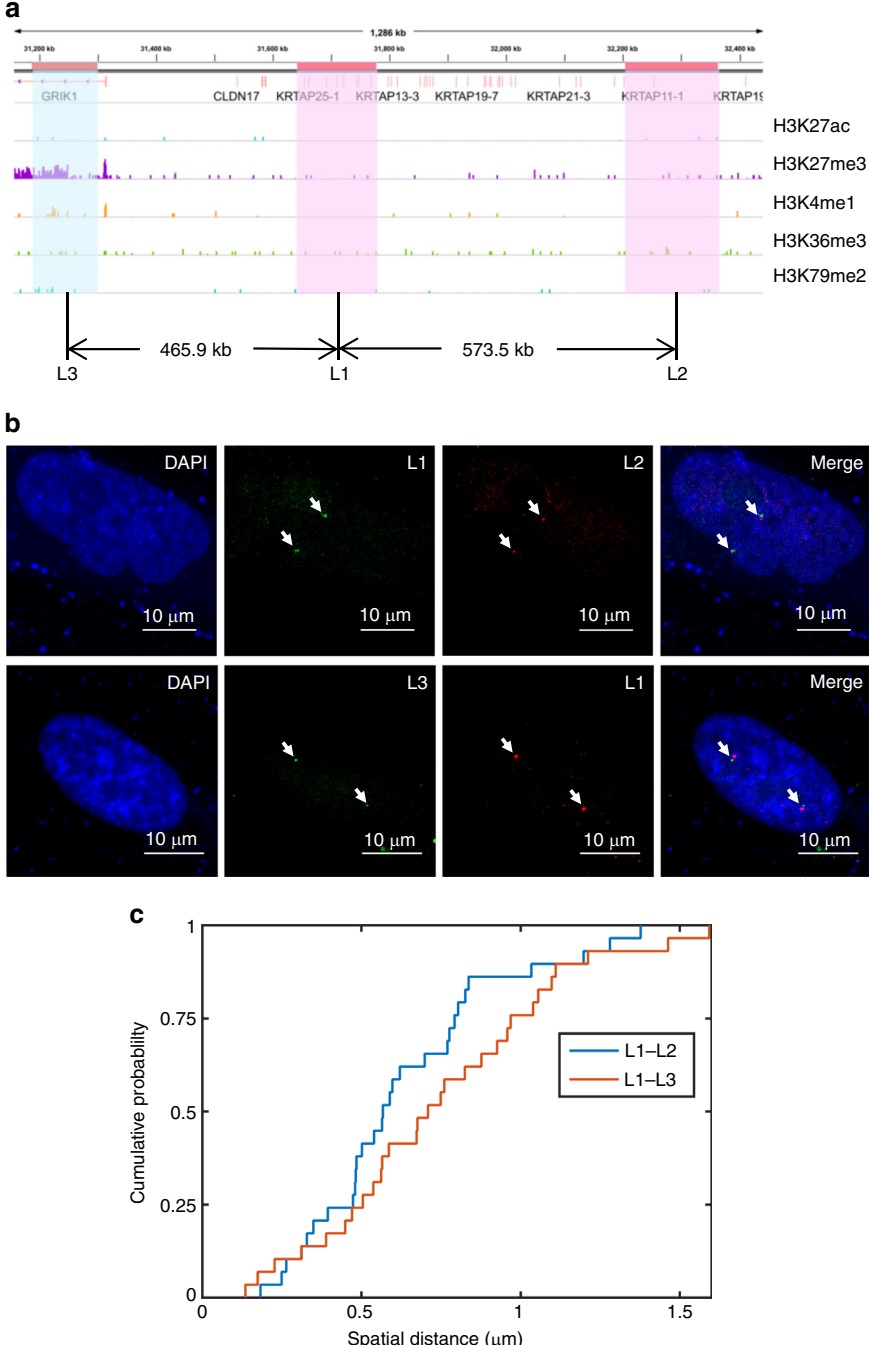

**Fig. 6** An experimental evidence that genomic loci from the same subcompartment tend to colocalize. **a** The ChIP-Seq profiles of different histone marks for the three loci L1, L2, and L3 examined in the FISH experiment. Locus L3 is relatively enriched with the mark H3K27me3 and depleted from the mark H3K36me3, and hence considered belonging to subcompartment B1. On the other hand, the two loci L1 and L2 are depleted from the marks H3K27me3, H3K27ac, H3K36me3, H3K4me1, and H3K79me2, and hence considered belonging to subcompartment B2. **b** One example of the experimental FISH images for the two loci L1 and L2 (top), and for the two loci L1 and L3 (bottom). **c** The cumulative distribution function (CDF) curves of the distances between L1 and L2 and between L1 and L3, indicating the tendency of the two loci L1 and L2 (belonging to the same subcompartment B2) to lie spatially closer to each other than the other two loci L1 and L3 (belonging to subcompartments B2 and B1, respectively), in spite of the smaller genomic distance between L1 and L3 than between L1 and L2. Data from 35 individual imaged copies of Chr21 were used to generate (**c**)

band, we found that it covers about 40% of the currently known protein-coding genes in Chr21 [http://www.uniprot.org/docs/humchr21], although it forms only <12% of the size of the chromosome. We also found the expression values of the genes in the G-band q22.3 are significantly higher than those of the genes in the other regions of the same chromosome in the IMR90 cell line (Supplementary Fig. 17). If we set the FPKM value '20' as a

threshold to classify genes into 'ON' and 'OFF' (as in the previous work[48]), we found that 12 genes in the band q22.3 are 'ON', forming 38.7% of all the active genes in Chr21 in the IMR90 cell line. In other words, we found that the density of expressed genes in the q22.3 region is 2.17 expressed genes/Mbp vs. 0.59 expressed genes/Mbp in the other regions of Chr21 in the IMR90 cell line. Moreover, from the visualization of this band in the final 3D

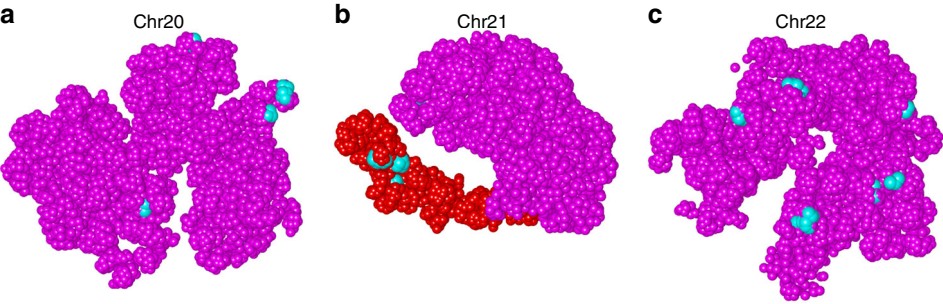

**Fig. 7** Super-enhancers tend to lie on the surface of the 3D models of chromosomes. **a–c** Super-enhancers (cyan) lie on the surfaces of the 3D models reconstructed by GEM-FISH for Chrs 20, 21, and 22, respectively. For Chr21 (**b**), four of the five super-enhancers were found in the gene-rich G-band q22.3 region, which is shown in red. The visualization was performed using UCSF Chimera[61]

model reconstructed by GEM-FISH, we found that the G-band q22.3 region formed an arm-like structure extending out from the chromosome body, and appeared to be accessible from all directions (Fig. 7b), which was consistent with the active nature of the whole band. By contrast, the discrimination of this gene-rich band was not that obvious in the final model reconstructed by GEM using Hi-C data alone (Supplementary Fig. 18), in which other parts of the chromosome also formed arm-like structures accessible from all directions. Thus, our modeling results may provide useful insights into understanding the regulatory roles of super-enhancers and the functional roles of the G-band q22.3 region in controlling the gene activities for Chr21.

Moreover, we also investigated whether our findings are specific to super-enhancers by comparing the spatial distributions of super-enhancers vs. regular enhancers (see Supplementary Note 7). As shown in Supplementary Fig. 19, the analysis on our reconstructed 3D models demonstrated that super-enhancers tend to lie closer to the chromosome surface than regular enhancers. Since super-enhancers are usually associated with cell-type-specific genes, their tendency to lie closer to the chromosome surface, and hence their higher accessibility relative to regular enhancers, supported the hypothesis that local interactions between genomic loci are the driving forces that lead to particular chromosome conformations[46,48,49].

## Discussion

Both Hi-C and FISH techniques have been widely used to study the 3D genome structure. Hi-C data provide the contact frequencies between genomic loci and can be interpreted as a measure of how frequently a pair of two genomic loci come close to each other in the 3D spatial space, while FISH directly measures the spatial distances between genomic loci through the imaging techniques. Usually, both Hi-C and FISH provide consistent measures, i.e., the Hi-C contact frequency between a pair of genomic loci is normally negatively correlated to their spatial distance measured by FISH. In some cases, however, the results from these two methods may seem to contradict each other. This apparent contradiction has brought up an interesting research problem, driving scientists to read the results of these two techniques in an attempt to reconcile them[41,50].

Inspired by Fudenberg and Imakaev[41], we also tried to investigate the consistency between Hi-C and FISH data used in our computational experiments (Supplementary Note 8). As shown in Supplementary Table 3, the average consistency between Hi-C and FISH data for Chr20, Chr21, and Chr22 was 82.03%, much higher than that of the control maps. Thus, we can conclude that both Hi-C and FISH data used in our study were reasonably consistent with each other.

Due to dynamic nature of chromosome structures and uncertainty in experimental data, it would be generally better for a chromosome structure modeling approach to compute an ensemble of chromosome conformations rather than just a single solution. In principle, in this study, we could also derive an ensemble of 3D models for both TAD-level resolution and intra-TAD conformations, using the same strategy as in GEM[36], in which an ensemble of multiple conformations with mixing proportions are computed. However, since GEM-FISH involves an assembly step to combine the intra-TAD conformations with the TAD-level resolution structure, the original optimization technique for computing multiple conformations with mixing proportions cannot be directly used to compute the final complete structures.

Nevertheless, to further evaluate the stability or uncertainty of the reconstructed 3D models, we first applied a strategy that has been widely used in computing an ensemble of conformations in current popular protein structure modeling methods, such as Xplor-NIH[51] and Rosetta[52]. More specifically, we ran GEM-FISH and the baseline method GEM 100 times, and evaluated both the average relative errors with respect to experimental FISH distances and the number of TADs correctly assigned to A/B compartments. This additional test demonstrated the good stability of the 3D models calculated by GEM-FISH (Supplementary Tables 4 and 5). In addition, we computed an ensemble of TAD-level resolution models in GEM-FISH using the same optimization technique as in GEM[36]. We found that computing such an ensemble of 3D conformations can lead to a slight improvement in the modeling performance (Supplementary Table 6). For more discussions about the dynamics of chromosome conformations, see "Supplementary Discussion".

In addition, we argue that GEM-FISH is not sensitive to the TAD calling method. The TAD-level resolution model reconstructed by GEM-FISH in the first step acts as the backbone for the final chromosome model, in which each 3D point corresponds to an imaged TAD. Thus, as long as the genomic locations of the TADs identified by the new TAD calling method are imaged, GEM-FISH should be able to build an accurate backbone model. In addition, the TADs identified by any calling method generally contain a sufficient number of Hi-C contact frequencies for GEM-FISH to reconstruct the high-resolution 3D models of individual TADs. Therefore, we believe that the modeling performance of GEM-FISH is robust to different TAD calling methods.

In this study, we presented a divide-and-conquer based method for modeling the 3D organizations of chromosomes. Our approach integrates both Hi-C and FISH data, as well as our current biophysical knowledge about a 3D polymer model. These different sources of information provide complementary

constraints that allow the reconstruction of more accurate 3D models that can capture both global and local geometric features of chromosomes. On the one hand, the global features were validated through the highly accurate assignment of TADs to A/B compartments, the reasonable placement of compartments in a polarized fashion relative to each other, the clear proximity of TADs within the same compartment, and the reasonable spherical shapes of the reconstructed chromosome conformations. On the other hand, the local features were validated through the proximity of loop loci, the colocalization and different epigenomic properties of the genomic segments belonging to the same subcompartment, and the tendency of expressed genes and interaction sites of the NPC component Nup153 to lie closer to the chromosome surface. In addition, the 3D models of chromosomes reconstructed by our method revealed interesting patterns of the spatial distributions of super-enhancers. Such an interesting finding will provide important hints for further investigating the functional roles of super-enhancers in controlling gene activities.

In general, every source of available data has its own merits and limitations. Integrating multiple sources of data constraints can help fully exploit their benefits and overcome their weaknesses during the 3D chromosome structure modeling process. On the way to calculate the whole 3D genome model, more data sources will be needed to further increase the accuracy of the reconstructed structure, and also compensate the limited availability and modeling power of existing input data. For instance, the geometric constraints derived from lamina-DamID experiments can also be used to infer the proximity of a chromatin region to the nuclear envelope[53]. In addition, the epigenomic profiles derived from ChIP-Seq can provide additional useful information to reconstruct the 3D architectures of chromosomes[54]. In principle, our framework can be easily extended to integrate all these different types of data constraints for modeling the 3D structures of chromosomes, which will thus further improve our current understanding of the underlying functional roles of 3D genome folding in gene regulation.

## Methods

**Determining the 3D chromosome models at TAD-level resolution**. We first calculate a relatively low-resolution (i.e., at TAD-level resolution) model of the chromosome of interest that is consistent with both input Hi-C and FISH data and also biophysically stable. Although a Hi-C map generally provides a relatively less number of geometric constraints between TADs than within individual TADs, these inter-TAD interaction frequencies can still provide useful restraints for pinning down the spatial arrangements of individual TADs at TAD-level resolution. On the other hand, FISH techniques directly image the 3D coordinates of different TADs from a number of cells, based on which we can also derive the average spatial distance between a pair of TADs over all cells. Overall, Hi-C and FISH data can provide useful and complementary restraints to determine the 3D organization of a chromosome at TAD-level resolution.

More specifically, to calculate the global chromosome structure at TAD-level resolution, we optimize a cost function that simultaneously incorporates the constraints derived from Hi-C data, FISH data, and prior biophysical knowledge about a 3D polymer model. In particular, the cost function $C_g$ is defined as,

$$C_g = C_1 + \lambda_E C_2 + \lambda_F C_3, \tag{2}$$

where $C_1, C_2$, and $C_3$ stand for the cost terms corresponding to the restraints derived based on Hi-C data, prior biophysical knowledge, and FISH data, respectively, and $\lambda_E$ and $\lambda_F$ represent the corresponding coefficients that weigh the relative importance of individual terms.

The term $C_1$ is defined using the same strategy as in the previous work[36], that is,

$$C_1 = \sum_i \text{KL}(P_i || Q_i) = \sum_i \sum_j p_{ij} \log \frac{p_{ij}}{q_{ij}}, \tag{3}$$

where KL(.) represents for the Kullback-Leibler (KL) divergence between two distributions, $p_{ij}$ represents the neighboring affinity between two genomic loci $l_i$ and $l_j$ in Hi-C space and $q_{ij}$ represents the probability that two 3D points $s_i$ and $s_j$ (corresponding to loci $l_i$ and $l_j$, respectively) are close to each other in the reconstructed 3D chromosome model. Here, the neighboring affinity is derived

according to the normalized interaction frequencies, that is,

$$p_{ij} = \frac{f_{ij}}{\sum_{i \neq j} f_{ij}}, \tag{4}$$

where $f_{ij}$ stands for the interaction frequency between the two genomic loci $l_i$ and $l_j$. In addition, $q_{ij}$ is defined as follows,

$$q_{ij} = \frac{(1 + ||s_i - s_j||)^{-1}}{\sum_{k \neq l} (1 + ||s_k - s_l||)^{-1}}, \tag{5}$$

where $||.||$ stands for the Euclidean distance between two 3D points.

In the cost function defined in Eq. (2), the term $C_2$ represents the conformation energy of a 3D polymer model, which is defined using the same strategy as in the previous studies[20,36]. The term $C_3$ is a new term that we add to incorporate the average spatial distance constraints derived from FISH imaging data. Let $F_{ij}$ denote the average spatial distance between two TADs $t_i$ and $t_j$ measured from FISH experiments. Then $C_3$ is defined as,

$$C_3 = \sum_i \sum_j (||s_i - s_j|| - F_{ij})^2, \tag{6}$$

where $s_i$ and $s_j$ stand for the coordinates of the centers of TADs $t_i$ and $t_j$, respectively.

More technical details about optimizing the cost function $C_g$ and selecting the optimal parameters $\lambda_E$ and $\lambda_F$ can be found below.

**Determining the 3D conformations of individual TADs**. Since FISH data only provide the geometric restraints on the 3D chromosome models at TAD-level resolution, and do not provide high-resolution information about the internal structure of each TAD, we cannot use them as pairwise distance restraints to determine the 3D coordinates of genomic loci within individual TADs. Nevertheless, we can still use FISH data to obtain a rough estimate of the radius of gyration (denoted by $\hat{R}_g$) of every TAD (more details can be found in Supplementary Note 10). In principle, incorporating $\hat{R}_g$ as an additional constraint to determine the 3D structures of intra-TAD chromosome fragments can further improve the modeling accuracy. Note that a similar scheme has also been used to incorporate FISH data to model the 3D genome structures from single-cell Hi-C data[55]. To incorporate the estimated radius of gyration into our modeling process, we define a new term $C_4$,

$$C_4 = |R_g^2 - \hat{R}_g^2|, \tag{7}$$

where $|.|$ stands for the L1 norm and $R_g$ stands for the radius of gyration of the reconstructed 3D model of the corresponding TAD and is calculated as follows,

$$R_g^2 = \frac{1}{N} \sum_{i=1}^{N} ||y_i - \bar{y}||^2, \tag{8}$$

where $N$ stands for the number of genomic loci in the 3D model, $y_i$ is the 3D coordinates of individual loci, $\bar{y} = \frac{1}{N} \sum_i y_i$, and $||.||$ stands for the Euclidean distance between two loci in the reconstructed 3D model.

Then the cost function for calculating the local 3D chromosome structures within individual TADs is defined as,

$$C_t = C_1 + \lambda_E C_2 + \lambda_R C_4, \tag{9}$$

where $C_1$ and $C_2$ stand for the terms representing the constraints from Hi-C data and prior biophysical knowledge, respectively, and $\lambda_E$ and $\lambda_R$ stand for the coefficients that weigh the relative importance of the corresponding terms. More details about selecting the optimal parameters $\lambda_E$ and $\lambda_R$ can be found below.

**Obtaining the final 3D model of the chromosome**. The TAD-level resolution 3D model of a chromosome is composed of a list of 3D points, each corresponding to a TAD. To integrate individual 3D models of TADs with this TAD-level resolution model of the chromosome, we first translate every TAD model such that its center coincides with the corresponding point in the TAD-level resolution model. Then, we adjust the orientation of every TAD model relative to its adjacent TADs while preserving the location of its center. This task can be achieved by rotating every TAD around its center to minimize the distance gaps between the current TAD and its adjacent ones. Reflection of a TAD model through a mirror plane passing by its center is also considered during this optimization process.

A rough estimate of the spatial distance between two adjacent TADs $i$ and $i + 1$ along the genome can be derived either from the contact frequency between the last genomic locus of TAD$_i$ and the first genomic locus of TAD$_{i+1}$, or from the relation between genomic and spatial distances in that particular chromosome if the contact frequency between these two loci is equal to zero. That is,

$$d_{i,i+1} = \begin{cases} f_{i,i+1}^{\alpha} & f_{i,i+1} \neq 0 \\ c \times g_{i,i+1}^{\beta} & f_{i,i+1} = 0 \end{cases}, \tag{10}$$

where $d_{i,i+1}$ stands for the estimated spatial distance between the last locus of TAD$_i$ and the first locus of TAD$_{i+1}$, $f_{i,i+1}$ and $g_{i,i+1}$ stand for the contact frequency and genomic distance between those two loci, respectively. According to the relation

between spatial distance and contact frequency for a pair of TADs derived in ref. [9], $\alpha$ is set to $-0.25$. The proportionality constant $c$ and the scaling exponent $\beta$ are derived from the FISH data[9] for each inspected chromosome.

The problem of placing the intra-TAD structures to the TAD-level resolution model can be formulated as an optimization problem, with the goal to minimize the following cost function $C_{\text{integration}}$,

$$C_{\text{integration}} = \sum_{i=1}^{n} (\|y_{s_{i+1}} - y_{e_i}\| - d_{i,i+1})^2, \tag{11}$$

where $y_{s_i}$ and $y_{e_i}$ are the first and last points in the model of TAD$_i$, respectively.

We use gradient descent to optimize $C_{\text{integration}}$. In particular, the gradient of $C_{\text{integration}}$ with respect to $y_{s_{i+1}}$ can be given by,

$$\frac{\partial C_{\text{integration}}}{\partial y_{s_{i+1}}} = 2\left(\|\Delta y_{i,i+1}\| - d_{i,i+1}\right) \times \frac{\Delta y_{i,i+1}}{\|\Delta y_{i,i+1}\|}, \tag{12}$$

where $\Delta y_{i,i+1} = y_{s_{i+1}} - y_{e_i}$.

In each iteration, until convergence of $C_{\text{integration}}$ is reached, $y_{s_{i+1}}$ should be updated by adding a value proportional to the negative of $\frac{\partial C_{\text{integration}}}{\partial y_{s_{i+1}}}$. Thus, the new value of $y_{s_{i+1}}$, denoted by $y'_{s_{i+1}}$, can be given by,

$$y'_{s_{i+1}} = y_{s_{i+1}} - \alpha \frac{\partial C_{\text{integration}}}{\partial y_{s_{i+1}}}, \tag{13}$$

where $\alpha$ is the learning rate. However, the points associated with TAD$_{i+1}$ are only allowed to rotate around its center. Based on the positions of $y_{s_{i+1}}$, $y'_{s_{i+1}}$, and the center of TAD$_{i+1}$, we can force the angle and axis of rotation of the point $y_{s_{i+1}}$ to fit into the direction towards point $y'_{s_{i+1}}$. We then rotate all the points within the TAD$_{i+1}$ model around its center using the same rotation angle.

**Optimization of the cost function**. We use gradient descent to minimize the cost functions $C_g$ and $C_t$ defined in Eqs. (2) and (9), respectively. The gradient of $C_g$ with respect to the coordinate $s_i$ is calculated as follows,

$$\frac{\partial C_g}{\partial s_i} = \frac{\partial C_1}{\partial s_i} + \lambda_E \frac{\partial C_2}{\partial s_i} + \lambda_F \frac{\partial C_3}{\partial s_i}. \tag{14}$$

More details about the calculation of $\frac{\partial C_1}{\partial s_i}$ and $\frac{\partial C_2}{\partial s_i}$ can be found in ref. [36]. In addition, $\frac{\partial C_3}{\partial s_i}$ is calculated as follows,

$$\frac{\partial C_3}{\partial s_i} = 2 \sum_j \frac{(\|s_i - s_j\| - dF_{ij})}{\|s_i - s_j\|}(s_i - s_j). \tag{15}$$

Similarly, the gradient of $C_t$ with respect to the coordinate $y_i$ is calculated as follows,

$$\frac{\partial C_t}{\partial y_i} = \frac{\partial C_1}{\partial y_i} + \lambda_E \frac{\partial C_2}{\partial y_i} + \lambda_R \frac{\partial C_4}{\partial y_i}, \tag{16}$$

where $\frac{\partial C_4}{\partial y_i}$ is calculated as follows,

$$\frac{\partial C_4}{\partial y_i} = \frac{2}{N} \text{sign}(R_g^2 - R_{\text{gest}}^2)(y_i - \bar{y}). \tag{17}$$

**Parameter selection**. In GEM-FISH, we optimize the cost function $C_g$ in Eq. (2) to calculate the TAD-level resolution 3D model of the whole chromosome, and the cost function $C_t$ in Eq. (9) to calculate the models of individual TADs. Each of these two cost functions has two parameters ($\lambda_E$ and $\lambda_F$ in Eq. 2, and $\lambda_E$ and $\lambda_R$ in Eq. 9). These parameters need to be chosen in a principled way that the calculated models best interpret the input Hi-C and FISH data, and the prior knowledge of a 3D polymer model. From the available FISH data, we can obtain rough estimates of the volumes of the whole chromosome and individual TADs (see Supplementary Note 10). Following the same strategy as in the previous work[36], we can select a pair of parameter values that maximize the following scoring function,

$$S = (1 - C_1) \times \frac{v}{|v - v'|}, \tag{18}$$

where $C_1$ (defined in Eq. 3) ranges between 0 and 1 and measures the degree of mismatch between the calculated 3D model and the input Hi-C data, $v$ is the estimated volume of the chromosome (or TAD) obtained from the prior knowledge, and $v'$ is the corresponding volume of the reconstructed model.

We use grid search to find the best pair of parameter values that yield high scores of $S$ for both Chr21 and Chr22. We found that the values $\lambda_E = 5 \times 10^{12}$ and $\lambda_F = 10^{-8}$ lead to the highest score for Chr22 and a reasonably high score for Chr21. Thus, we choose them as the default values for these two parameters when computing the TAD-level resolution 3D models of chromosomes (more analyses are provided in Supplementary Note 11).

Similarly, we found that the values $\lambda_E = 5 \times 10^{11}$ and $\lambda_R = 10^{-7}$ yield reasonably high scores of $S$ for individual TADs. Thus, we choose them as the default values for these two parameters when computing the 3D conformations of individual TADs.

**The computational efficiency of GEM-FISH**. We first analyzed the running time needed by GEM-FISH to reconstruct a 3D model for each of Chrs 20, 21, 22, Xa, and Xi at the TAD-level resolution. As shown in Supplementary Table 7, the running time for GEM-FISH to reconstruct TAD-level resolution models was in a range of minutes.

For reconstructing the intra-TAD models, the time taken by GEM-FISH depends on the resolution of the intra-TAD Hi-C maps used. Since the 3D structures of all TADs of a given chromosome can be calculated in parallel, the time taken to calculate all the intra-TAD models is the maximum running time for reconstructing the intra-TAD structures of that chromosome. As shown in Supplementary Table 8, the time needed by GEM-FISH to calculate the most time-demanding intra-TAD models at 5 Kbp resolution for Chrs 20, 21, 22, Xa, and Xi was approximately 26 h. Supplementary Table 8 also summarizes the time needed to calculate the most time-demanding intra-TAD models at 10 Kbp, 25 Kbp, and 50 Kbp resolutions, respectively.

As for the memory usage, for the TAD-level resolution modeling, the size of the largest matrix that needs to be stored in memory for the five tested chromosomes was $40 \times 40$ (Chrs Xa and Xi). For the intra-TAD modeling, the size of the Hi-C map depends mainly on the resolution used. In our computational experiments, with 5 Kbp resolution, the size of the Hi-C map corresponding to the largest TAD had $1576 \times 1576$ entries. In other words, the largest matrix needed around 20 Mb of memory.

According to the above analysis, GEM-FISH does not need extensive memory resource, and the final 3D models of the chromosomes at 5 Kbp resolution can be calculated within one day.

**FISH protocol based on in situ nick translation**. Human IMR90 fibroblast cells (ATCC, CCL-186) were purchased from ATCC and cultured in Dulbecco's Modified Eagle's medium (DMEM) (high glucose, Gibco cell culture medium, Fisher Scientific Company) with 1% penicillin-streptomycin solution (Gibco), 1% non-essential amino acids (Gibco) and 10% foetal bovine serum (FBS, Gibco), and then attached on slides before DNA FISH.

DNA probe libraries were generated with bacterial artificial chromosome (BAC) (L3 covered by CTD-2053M15, Chr21:31,187,325-31,298,669; L1 covered by RP11-1133B5, Chr21:31,641,247-31,776,629; and L2 covered by CTD-3175I1, Chr21: 32,203,928-32,361,056) using nick translation.

The 3D FISH method was adapted from the previous work[56] with slight modification. More specifically, the cells were fixed with 4% paraformaldehyde, washed twice with PBS, and then treated with 0.1 M pH 7.4 Tris-HCl. Next, the cells were treated with 0.1% saponin and 0.1% Triton X-100 in PBS for 10 min and washed twice with PBS. Then the cells were incubated with 20% glycerin at room temperature for 20 min, and freeze-thawed in liquid nitrogen three times. After being washed with PBS, the cells were treated with 0.1 M HCl solution at room temperature for 30 min, and then washed with PBS. After that, 0.5% saponin and 0.5% Triton X-100 solution were employed for membrane penetration. After being cleaned with PBS, the cells were balanced in 50% formamide and 2× SSC solution for at least 10 min. Probes and cells were incubated on the hybrid instrument (Thermobrite, IRIS) after being mixed, and then were incubated at 75 °C for 5 min, 37 °C for 12–18 h. After hybridization, the cells were washed three times with washing buffer which included 0.2% CA-630 and 2× SSC, and sealed with mounting medium containing DAPI (2 µg/ml).

The nick translation reaction system contained 10 x DNA polymerase buffer, 10 x DNase I buffer, 1 mM dATP/dCTP/dGTP mixture, 1 mM 2:1 ratio dTTP/ Fluorescent dUTP (Alexa Fluor 488-dUTP, Alexa Fluor 594-dUTP, Alexa Fluor 647-dUTP, Invitrogen, U.S.A.), 200U DNase I (NEB), 10U DNA polymerase I (NEB), 1 ~ 2 µg BAC plasmid DNA (Thermofisher) and ddH₂O. The mixture was incubated at 15 °C for 1.5 ~ 2.5 h. Then 2 µl 0.5 M EDTA was added, incubated at 65 °C for 5 min. After that, 50 µg salmon sperm DNA, 5 µg human Cot-1 DNA, 3 M sodium acetate, and anhydrous ethanol were added and mixed, stored at −80 °C overnight. After centrifuging at 4 °C, the probes were dissolved with hybridization buffer.

The slides were imaged with 100× oil immersion objective on LSM780 (Zeiss Company, Jena, Germany) and 100× oil immersion objective on Nikon A1 (Nikon Instruments Inc., Japan). The spatial distances between every two signals and the volumes of nuclei were measured accordingly.

**Quantification of FISH images**. Surface rendering, 3D reconstruction, measurement of the nucleus volume, and the distance measurement between every two signals in every cell were performed with the help of the commercial software Imaris (version 9.2.1, Bitplane AG, Switzerland). Every distance was normalized by the corresponding nuclear volume, which was derived based on the quantification by Imaris through 3D reconstruction and quantification.

**Datasets used**. The Hi-C and FISH data of the human IMR90 cell line can be downloaded from NCBI GEO GSE63525[8] and [https://www.sciencemag.org/content/353/6299/598/suppl/DC1][9], respectively. The ChIP-Seq data for the IMR90 cell line can be downloaded from NCBI GEO: GSE38442 (H3K9me3)[57], GSM469966 (H3K27ac)[58,59], GSM469970 (H3K4me3)[58,59], GSM521895 (H3K4me1)[58,59], GSM521890 (H3K36me3)[58,59], GSM469968 (H3K27me3)[58,59],

GSM521911 (H3K79me2)[58,59], GSM521895 (H3K4me1)[58,59], GSM521900 (H3K4me2)[58,59], and GSM521933 (Control)[58,59]. The RNA-Seq data and the DamID-Seq data of Nup153 for the IMR90 cell line can be downloaded from GSE87831[60].

**Reporting summary**. Further information on research design is available in the Nature Research Reporting Summary linked to this article.

## Data availability

All relevant data supporting the key findings of this study are available within the article and its Supplementary Information files or from the corresponding author upon reasonable request. The source data underlying Supplementary Tables 2, 4–7, 10–13 and Supplementary Figures 17, 42, 43, 44, 45, 46, 47, 48, and 49 are provided as a Source Data file. A reporting summary for this article is available as a Supplementary Information file.

## Code availability

GEM-FISH was implemented using Matlab 2014b, and its source code can be downloaded from https://github.com/ahmedabbas81/GEM-FISH.

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

## Acknowledgements

We thank Mohamed Ibrahim (King Abdullah University of Science and Technology) for helpful discussions. Molecular graphics and analyses were performed with the UCSF Chimera package. Chimera is developed by the Resource for Biocomputing, Visualization, and Informatics at the University of California, San Francisco (supported by NIGMS P41-GM103311). This work was supported in part by the National Natural Science Foundation of China (61472205 and 81630103), State Key Research Development Program of China (2017YFA0505503), the National Natural Science Foundation of China under Grant (91729301, 31671383, 61475010, 61729501, and 11671005), the China's Youth 1000-Talent Program, and the funds from Beijing Advanced Innovation Center for Structural Biology, Tsinghua University. We acknowledge the support of NVIDIA Corporation with the donation of the Titan X GPU used for this research.

## Author contributions

A.A., J.G., M.Z., and J.Z. conceived the research project. J.Z. supervised the research project. A.A., G.Z., and J.Z. designed the computational pipeline. A.A. implemented GEM-FISH. A.A., X.H., B.Z., Z.M., J.G., M.Z., and J.Z. discussed the modeling and validation results. J.N., P.J.S., and J.G. performed the FISH validation experiment and analyzed the corresponding imaging data. A.A. and J.Z. wrote the manuscript with support from all authors.

## Additional information

**Competing interests:** The authors declare no competing interests.

