## [Peer Review File · Nature Communications]

Reviewers' comments:

Reviewer #1 (Remarks to the Author):

In the manuscript, the authors developed a divide-and-conquer based method called GEM-FISH, for integrating high-throughput chromatin conformation capture Hi-C data with FISH imaging data and polymer physics to reconstruct the 3D model of chromosome. The GEM-FISH method consists of three main steps. First, it builds a TAD-level-resolution model constrained by inter-TAD contact frequencies from Hi-C, polymer conformation energy, and the average spatial distance between pairs of TADs from FISH. Second, the internal conformation with each individual TAD is built by integrating intra-TAD contact frequencies from Hi-C, polymer model, and the rough estimate of the radius of gyration of TAD from FISH. In the first two steps, the authors added weights to the polymer model and FISH information into the cost function, and the default weights were selected by a grid search which maximizes a specific scoring function. The last step of GEM-FISH is to align the individual TAD structures to the TAD-level-resolution model to obtain the final model, by first translation of the TADs to match their centers to the position in the global model, then rotation and/or reflection of the TADs so that the spatial distance between adjacent TADs best fit the estimates from Hi-C data.

The authors evaluated the performance of GEM-FISH using chromosome 20, 21, 22 and X data in IMR90 cells. They observed that GEM-FISH significantly decreased the relative error between the model and FISH data, particularly for the pairs of TADs that are far away along the genome and have few contact frequencies, comparing to the GEM model using only Hi-C data. Overall, GEM-FISH showed higher accuracy of assignment of TADs in the 3D model to A/B compartments, and TADs within the same compartment are closer than across different compartments. For local features, the chromatin loop anchor loci are significantly closer in the GEM-FISH model. For subcompartments B1 and B2, loci within the same subcompartment tend to colocalize, and regions from B1 are more compacted than from B2. In addition, the super-enhancers show a novel spatial distribution that tend to lie on the surface of the chromosome, and a gene/super-enhancer-rich band on chr21 extend out of the chromosome body, forming an arm-like structure.

Integrating Hi-C and FISH data for modeling 3D chromatin structure is an important question and of great interest to the field. The proposed GEM-FISH method is a first attempt to address this question. Overall the method is nicely developed and clearly written. My main concern is about the use of FISH data and the evaluation metrics.

Major Comments

1. I have a concern about the method evaluation part. The authors used both Hi-C and FISH data to optimize their 3D model and then used the same FISH data to evaluate the model. It was not surprising that their method performed better than the original GEM method using Hi-C data only. It will be more convincing if the authors could use independent evidence(s) to conduct the evaluation.

2. Since the spatial distance from FISH used in the model is averaged over many single cells, would the cell-to-cell variability in the FISH data affect the results? In Fig. S7-S11, the volume and size of chromosomes vary widely among cells. It would greatly enhance the method if the authors could take the advantage of the single cell feature in the FISH data and incorporate the cell-to-cell variability into their model.

Minor Comments

1. It would be good to include the computational efficiency of GEM-FISH (running time, memory usage) for different resolution/length of the chromosome.
2. It is not clear what resolution was used for the 3D modeling.
3. What is the average size of the TADs? According to Table 2, there are about 30-40 TADs for each chromosome, which seems fewer than expected.

4. It would be better to perform some colocalization test to show the loci within the same subcompartment tend to colocalize in the 3D models.
5. Similarly, could the authors provide some statistics (for example, radial distribution) to demonstrate that super-enhancers tend to lie on surface.
6. When calculating the accuracy of assigning TADs to compartments, is the "ground truth" here the assignment inferred from FISH or Hi-C data? If not Hi-C data, it would be good to compare with the compartment assignments called from Hi-C.

Reviewer #2 (Remarks to the Author):

In this paper the authors describe an approach, GEM-FISH for integrating Hi-C data with FISH data to more reliably infer 3D model of the chromosome. This method extends the previous approach, GEM, developed by the authors which used only Hi-C data and biophysical constraints. The GEM method works by minimizing the KL divergence between the distribution of two pairs of regions to have high contact counts and the probability distribution two pairs of regions to be close in 3D space, subject to minimizing constraints imposed by biophysical properties. The authors here incorporate additional constraints derived from FISH data.

To assess the advantage of GEM-FISH over GEM, the authors compare the spatial distance inferred from the 3D models constructed to those from FISH, ability to place TADs in the right compartments, ability to recover 3D models for chrX and finally chromatin mark information for different subcompartments. Overall this seems like an interesting piece of work, but I feel that the advantage of the GEM-FISH over GEM is not significant. Most of the findings are validation of what has been shown and it is not clear what new insights are being derived. There are also several aspects of the paper that need clarification.

Major

1. How were TADs called in this method? How sensitive are the TAD calls to the robustness of the method?
2. The within TAD models are not studied in a lot of detail and only examined in the subcompartment analysis which is done in an adhoc manner. In this analysis the type of region is defined in an arbitrary manner. It is not clear what 3 and 10 are, and the methods don't get into the details of this. Why not use what is published in Di Perro [39] or a simpler model?
3. The analysis of super enhancers on the 3D model is interesting. It seems the exposed region where the super enhancers lie are gene rich. Are the genes expressed in IMR90?
4. The comparison of the TAD spatial distance estimation from GEM and GEM-FISH in figure 2 and supp figs S1, S2, S3 and S4 is not fair because GEM-FISH is using the spatial distance as a constraint which GEM is not.
5. For the users of this tool, providing a sense of range of parameters would be very helpful. I think showing how the objective and the reconstructed model change as a function of different parameters would be beneficial.
6. The authors don't mention how fast the algorithm runs. Providing some numbers for estimating the within TAD and TAD-resolution model will be helpful.
7. Equation 6, which norm is used?
8. The authors show the reconstructed 3D models of entire chromosomes, but they also have this at the individual TAD level. It would be interesting to see the structure of the individual TADs and how different chromatin marks are organized within individual TADs.
9. Is always a single 3D structure found, or can there be multiple equally good solutions? Does the FISH data enable to improve the stability?

Minor:

1. Label the color maps in Figure 2 to indicate this is error.
2. Supp fig 2-5 are not referred in the text

Reviewer #3 (Remarks to the Author):

Transparency statement: this review was written by Guillaume Filion (CRG, Barcelona).

The manuscript of Ahmed Abbas et al. titled "Integrating Hi-C and FISH data for modeling 3D organizations of chromosomes" presents a novel method called GEM-FISH to reconstruct the 3D conformation of chromosomes from Hi-C and FISH data. The principle of GEM-FISH is to simultaneously maximize: i. the agreement with contact frequencies derived from Hi-C ii. the conformation energy and iii. the agreement with distances derived from FISH. It builds on a strategy initially developed in the GEM method (reference [32]) borrowing concepts from manifold learning. The authors show that GEM-FISH improves substantially the results of GEM in terms of reconstruction (measured by agreement with FISH, A/B compartments, chromatin marks or epigenetic state of the X chromosome).

I found the article extremely well written. This is a model of clarity and concision, and I highly encourage the authors to continue writing in this outstanding style. The concepts of using manifold learning is very interesting and it should be further explored in this field of research. Overall, the manuscript describes a method with great potential.

However, I found some issues in the manuscript. While none of them is fatal, fixing them may require major changes to the manuscript and / or the software, so the authors may prefer to submit to another journal if time is an issue. Below is the list of issues I think the authors should address. I hope that they will find it useful to improve their manuscript.

MAJOR ISSUES

1. The authors acknowledge 25 citations presenting methods to reconstruct the 3D structure from Hi-C maps (references [1,8-31]), but they compare GEM-FISH to none of them, arguing that GEM already outperformed other state-of-the-art methods. Since this is a different data set, and since the reconstruction criteria are different in this study, the authors should again compare GEM-FISH to these state-of-the-art methods. This is important to really establish the performance of GEM-FISH versus to other methods. Since these other methods are more popular than GEM, it would also allow more readers to judge the quality of GEM-FISH.

2. Several conformations are usually compatible with a set of restraints given by Hi-C and/or FISH. It seems that GEM deals with ensemble solutions, but GEM-FISH does not (as far as I understand, it uses gradient descent to find one local minimum of the cost function). The output is presented as "the" structure of the chromosome, but it may not even be the optimum. Even though there is some value in proposing a structure, it seems unreasonable to not quantify the uncertainty around this structure. Providing ensemble solutions is one way, but there may be others.

To summarize these last two points, from the perspective of a user, it is not acceptable to take the single solution provided by GEM-FISH as correct, based only on the comparison with GEM on one data set. Inference is always associated with a certain degree of certainty but in its present form, GEM-FISH neither shows that the performance is generalizable, nor does it provide any information regarding the confidence the user should have in the results.

3. Section 4.3 explains how the mixing parameters of the cost function were chosen. This is based on a single data set at a single resolution of the Hi-C data (one can assume that the TAD resolution is always the same, but not the Hi-C). What are the reasons to think that these parameters are acceptable for other datasets at other resolutions? In the absence of this information, how can the user trust the solution? The authors should show that these values are

robust across resolutions and datasets, and if they are not, provide a way for the user to obtain meaningful results in different conditions.

4. The models are validated by FISH data (see for instance Fig 2g and similar panels from supplement figures), but the same data was used to fit the model. This leads to a phenomenon known as overfitting that must be avoided. At the very least, the authors should use a 'leave-one-out' strategy where they fit the model with all FISH data points except one, and then use only this point for validation. Alternatively, they could use k-fold cross-validation or any other approach to reduce overfitting. Ideally, they should test their model with orthogonal FISH data (from another laboratory for instance).

5. Another point related to the validation by FISH data is that Fig 2g and similar panels from supplement figures show a black curve labelled "Exp. FISH" that is perfectly smooth. The legend says that they are "derived from the experimental FISH data", but I could not find the description of this "derivation". Why not show the original data points? With the current representation, there is no information of the variability around the fitted line. It would also be interesting to see whether the line is better at fitting one outlier to the detriment of all other points. The authors should either represent the original data, or clearly explain why they think it is better not to. Either way, they should explain what the fitted line is.

6. Figure 4 is interesting but it cannot be used for validation. Chromatin marks correlate with A/B compartments, but since the authors can compute the A/B compartments from the original Hi-C map, they learn nothing new about the structure of the chromosome by comparing the distributions of histone marks. The authors may present this result, but not as a validation of GEM-FISH. Instead, they should include tests that are orthogonal to the ones they have already included.

7. In section 2.3 the authors claim that GEM-FISH performs better than GEM on the X chromosome because FISH allowed to distinguish the two X chromosomes. The argument is interesting and it could be presented as a main feature of GEM-FISH (because the heterozygosity is probably not high enough to distinguish homologs by Hi-C in most cell lines). But in the current manuscript, it does not speak in favor of GEM-FISH against GEM: there exists allele-specific Hi-C data in mouse, so if the authors want to show that the GEM-FISH method surpasses the GEM method, they should test both on such data. Alternatively, they could rewrite the manuscript in order to show that the main advantage of GEM-FISH is that it makes better use than GEM of existing FISH data.

8. Section 2.6 about the super enhancers does not show any control. I would suggest to remove this section and the associated results altogether, unless the authors can show some evidence that the results they observe are specific to super enhancers versus typical enhancers. If they can show these controls, it would be appropriate to substantiate the claim that "These novel patterns of the spatial distributions of super-enhancers can (...) provide important insights into revealing their functional roles in gene regulation..." (last sentence of the section). The authors should explain in the discussion what kind of insight they are talking about. If the authors cannot or do not want to explain why this pattern is important, they should remove the claim.

9. What is the evidence that there is no bug in GEM-FISH, or that the code actually does what the authors claim? The authors should provide some test code to show some evidence that the code is correct.

MINOR ISSUES

1. There is no discussion of the computational performance of GEM-FISH and of the hardware requirements. Even ballpark estimates would be useful. Due to the availability of FISH data, the

largest chromosome used here is the X. It would be interesting to know if chromosome 1 can be modeled at the same Hi-C resolution.

2. In the Code availability section, it should be specified that GEM-FISH is written in Matlab (among others because Matlab is nonfree, so this is relevant for availability).

3. The instructions on the page of GEM-FISH are unclear. For instance it says "Download the program" but there are over 40 files and none of them is obviously "the program". Also, GitHub repositories are usually cloned, so the meaning of "download" is not clear. Statements like "go into the folder" are ambiguous, and so are statements like "run the m-file". In all these cases, the authors should show the commands the users have to write.

Reviewer #4 (Remarks to the Author):

In this manuscript, the authors developed a computational framework to integrate Hi-C and multiplexed FISH data to generate 3D models of chromosome organization. Briefly, the authors took advantage of both Hi-C and FISH data to model chromosome conformation at the TAD-to-chromosome length scale, then used HiC data to guide the modeling at sub-TAD length scale, and finally combined the models at the two scales together using HiC data. This framework performs significantly better in capturing the correct large-scale organization than previous modeling using HiC data alone, and retains the ability to model sub-TAD structures without FISH data at the sub-TAD resolution. Using this framework, the authors obtained a series of structural features consistent with previous reports, and further reported new findings regarding the spatial localization of super enhancers and the G-band q22.3 of Chr21. The study is innovative, technically sound and clearly presented, and represents a significant progress in the computational modeling of chromosome organization that would be of broad interest to the genome biology and cell biology fields. I would recommend this manuscript for publication in Nature Communications as long as the following points are addressed:

Major points:

1. The authors clearly demonstrated that at the TAD-to-chromosome level, combining Hi-C data and FISH data in the modeling is better than using Hi-C data alone. But it is unclear how much the modeling at this scale benefits from the inclusion of Hi-C data in the GEM-FISH strategy. In other words, during the optimization of the cost function C_g , does the third term with C_3 dominate this process? If the authors use FISH data alone to model large scale chromosome conformation, will that give comparable results to the current GEM-FISH results? I would love to see in Fig. 2 comparisons among GEM-FISH, Hi-C only and FISH-only modeling. This comparison will not undermine the value of this work even if it shows that the FISH-only modeling is sufficient to address the large-scale fitting, but will only make the work more thorough and clear.

2. Another analysis that could be very helpful is to show how much the individual chromosome conformations measured by the multiplexed FISH deviate from the authors' averaged modeling result. This analysis could help evaluate how stereotypic/dynamic the individual chromosome conformations are at this scale.

3. The authors should state why they used the KL divergence function to define C_1 but instead used the sum-square function for C_3 . In both cases the authors are essentially comparing the differences between two distributions, so why not use the KL divergence for C_3 as well, or adopt the sum-square function in the C_1 definition?

4. On Page 22, in order to infer the radius of gyration of individual TADs, the authors first plotted the distributions of volumes and sizes for individual copies of chromosomes, and selected "the values corresponding to the highest values to be the expected volume and size of that chromosome". What is the rationale for choosing the highest values of the distributions as the

expected values here, rather using the means or medians?

5. One potential oversimplification in this framework could be the assumption that adjacent TADs are always well insulated and do not overlap. Recent super-resolution imaging investigation of TADs showed that adjacent TADs could substantially overlap, depending on the epigenetic states of the TADs (Nature 529 418-422 (2016)). I am not sure if adjacent TADs are allowed to overlap in the authors' framework here. It would be great if the authors could clarify. If such an assumption was implemented, maybe the authors could discuss potential ways to improve this front in their future work.

Minor points:

6. On Page 3 the authors stated that "Dixon et al [4], introduced the concept of topologically associated domains (TADs)". TADs were independently discovered by four groups at about the same time: J. R. Dixon et al., Nature 485, 376–380 (2012). E. P. Nora et al., Nature 485, 381–385 (2012). T. Sexton et al., Cell 148, 458–472 (2012). C. Hou, L. Li, Z. S. Qin, V. G. Corces, Mol. Cell 48, 471–484 (2012). I recall the very terminology of "TADs" was first used in the Nora et al paper. The authors should cite all four papers.

7. On Page 9, Equation 7 is missing a square sign on the right end, given the common definition of radius of gyration.

8. On Page 14, the authors pointed out the lack of allele-specific Hi-C data for distinguishing ChrXa and ChrXi in IMR90 cells. How did the authors still manage to get "3D models calculated by GEM (which only takes Hi-C data as input)" for ChrXa and ChrXi to perform the calculations in Table 2?

9. On Page 17 and Fig. 7, the authors tried to show that "a certain subcompartment tend to colocalize in 3D models reconstructed by GEM-FISH". How much of this colocalization is due to the genomic proximity among regions in the same subcompartment? Do regions far away on the genomic map still colocalize in a subcompartment? The authors should include a genomic map of the subcompartments and identify the genomic regions on the 3D model.

10. On Page 19, the authors stated that "Hi-C contact frequency between a pair of genomic loci is normally inversely proportional to their spatial distance measured by FISH". I think the authors meant to say "Hi-C contact frequency between a pair of genomic loci is normally negatively correlated with their spatial distance measured by FISH", since the exact scaling relationship is "inversely proportional to the 4th power" as the authors cited before.

11. The authors should quantify the surface distribution of the super-enhancers, and compare them with randomly picked regions of the same size in the 3D models.

12. On Page 22, the authors wrote "n stands for the number of cells". n should be the total measured copies of the chromosome, given that IMR-90 is a diploid cell line.

Reviewer 1

General comments

In the manuscript, the authors developed a divide-and-conquer based method called GEM-FISH, for integrating high-throughput chromatin conformation capture Hi-C data with FISH imaging data and polymer physics to reconstruct the 3D model of chromosome. The GEM-FISH method consists of three main steps. First, it builds a TAD-level-resolution model constrained by inter-TAD contact frequencies from Hi-C, polymer conformation energy, and the average spatial distance between pairs of TADs from FISH. Second, the internal conformation with each individual TAD is built by integrating intra-TAD contact frequencies from Hi-C, polymer model, and the rough estimate of the radius of gyration of TAD from FISH. In the first two steps, the authors added weights to the polymer model and FISH information into the cost function, and the default weights were selected by a grid search which maximizes a specific scoring function. The last step of GEM-FISH is to align the individual TAD structures to the TAD-level-resolution model to obtain the final model, by first translation of the TADs to match their centers to the position in the global model, then rotation and/or reflection of the TADs so that the spatial distance between adjacent TADs best fit the estimates from Hi-C data.

The authors evaluated the performance of GEM-FISH using chromosome 20, 21, 22 and X data in IMR90 cells. They observed that GEM-FISH significantly decreased the relative error between the model and FISH data, particularly for the pairs of TADs that are far away along the genome and have few contact frequencies, comparing to the GEM model using only Hi-C data. Overall, GEM-FISH showed higher accuracy of assignment of TADs in the 3D model to A/B compartments, and TADs within the same compartment are closer than across different compartments. For local features, the chromatin loop anchor loci are significantly closer in the GEM-FISH model. For subcompartments B1 and B2, loci within the same subcompartment tend to colocalize, and regions from B1 are more compacted than from B2. In addition, the super-enhancers show a novel spatial distribution that tend to lie on the surface of the chromosome, and a gene/super-enhancer-rich band on chr21 extend out of the chromosome body, forming an arm-like structure.

Integrating Hi-C and FISH data for modeling 3D chromatin structure is an important question and of great interest to the field. The proposed GEM-FISH method is a first attempt to address this question. Overall the method is nicely developed and clearly written. My main concern is about the use of FISH data and the evaluation metrics.

Response: Thank you for the nice summary. We have performed several additional tests and made a number of major changes in the revised manuscript to address your main concerns. Below please find our point-to-point response to your comments.

Major comments

Comment 1: I have a concern about the method evaluation part. The authors used both Hi-C and FISH data to optimize their 3D model and then used the same FISH data to evaluate the model. It was not surprising that their method performed better than the original GEM method using Hi-C data only. It will be more convincing if the authors

could use independent evidence(s) to conduct the evaluation.

Response: Thank you for pointing this out. To make a more fair comparison, we have conducted an additional 10-fold cross-validation procedure to further evaluate the performance of our approach. In each fold, we built the average FISH distance matrix, denoted by F_1 , of a specific chromosome using only 90% of the available FISH data and used it to optimize the 3D models in GEM-FISH. Then, we evaluated the reconstructed structures using the average FISH distance matrix obtained from the remaining 10% of FISH data, denoted by F_2 , of that chromosome (which can be considered completely independent data). We also assessed the 3D structures reconstructed by GEM against the same distance matrix F_2 . The average relative errors obtained are listed in Table R1.

We also performed another three independent validation tests. We first measured the asphericity values of the 3D models reconstructed by both GEM-FISH and GEM [1], based on the eigenvalues of their gyration tensors [2]. We observed that the models reconstructed by GEM-FISH tended to be spherical in shape, while those reconstructed by GEM tended to be more extended (Table R2 and Fig. R1). A potential reason for these extended structures obtained by GEM is the relatively weak restraints of Hi-C contact frequencies between those TADs far away in the genomic distance.

In addition, we inspected the radial distributions of expressed genes in the final 3D models of Chrs 20, 21, and 22 obtained by GEM-FISH and GEM, respectively. We observed that the expressed genes tend to lie closer to the surface of the chromosome territories in the models reconstructed by GEM-FISH than those in the models reconstructed by GEM (Fig. R2), in agreement with the fact that active transcription regions tend to lie closer to the chromosome surface [3].

Moreover, we used a previous finding derived from an independent FISH dataset about the spatial arrangement of genomic loci related to the nuclear pore complex (NPC) component Nup153 [4,5], to further verify the reasonableness of our reconstructed 3D models. In particular, we first derived a number of genomic positions that are known to interact with Nup153 in Chrs 20, 21, and 22 from the DNA adenine methyltransferase identification (DamID) data [4]. We inspected the radial distribution of these genomic sites in the final 3D models of Chrs 20, 21, and 22 reconstructed by GEM-FISH and compared them to that of randomly selected loci of the same genomic lengths (denoted by Control-Nup loci). We observed that most of the genomic sites interacting with Nup153 generally tend to lie closer to the surface of the chromosome territories than the Control-Nup loci (Fig. R3). Nevertheless, we also found that some sites interacting with Nup153 lie relatively close to the center of the chromosome. This observation was also consistent with the fact derived from the previous FRAP (fluorescence recovery after photobleaching) analysis that Nup153 is a nuclear basket component with a stable subpopulation at the nuclear envelope-embedded NPC, and a dynamic subpopulation exchanging between the NPC and the nucleoplasm [5].

Furthermore, we have conducted a new FISH experiment to validate the relative spatial distances between a triplet of genomic loci and show that loci belonging to the same

subcompartment tend to lie spatially closer to each other than those loci belonging to different subcompartments even at a larger genomic distance. In particular, we designed probes for three genomic loci, denoted by L1, L2, and L3, respectively, where the genomic distance between L1 and L2 is larger than that between L1 and L3 (Fig. R4(a)). The two loci L1 and L2 are depleted from the marks H3K27me3, H3K36me3, H3K27ac, H3K4me1, and H3K79me2, and hence considered belonging to subcompartment B2 [6]. On the other hand, the locus L3 is enriched with H3K27me3 and depleted from H3K36me3, and hence considered belonging to subcompartment B1 [6] (Fig. R4(a)). We examined the spatial distances between L1-L2 and between L1-L3 derived from this new FISH experiment, and found that the spatial distance between the two loci L1-L2 is consistently smaller than that between L1-L3 (Fig. R4(b) and R4(c)). This result provided a new experimental evidence to show that the loci belonging to the same subcompartment tend to locate spatially closer to each other.

All the above additional validation tests have further supported the superiority of our new modeling approach with independent evidences. We have added the results of all the above tests to the revised manuscript (please refer to the highlighted parts in Section 2.2 and Section 2.5).

Comment 2: Since the spatial distance from FISH used in the model is averaged over many single cells, would the cell-to-cell variability in the FISH data affect the results? In Fig. S7-S11, the volume and size of chromosomes vary widely among cells. It would greatly enhance the method if the authors could take the advantage of the single cell feature in the FISH data and incorporate the cell-to-cell variability into their model.

Response: Thank you for the perspective comment. Following your suggestion, we have proposed a new variant of GEM-FISH called GEM-FISH* in which we tried to take into account the cell-to-cell variability of FISH data by assigning different weights to distinct pairwise TAD distances. The intuition was to assign higher weights to those distances with smaller variances. In particular, in GEM-FISH*, the term corresponding to the FISH distances (C_3) is now defined as follows,

$$C_3 = \sum_i \sum_j \frac{1}{\sigma_{ij}} (||s_i - s_j|| - F_{ij})^2, \quad (1)$$

where F_{ij} and σ_{ij} denote the mean and standard deviation of spatial distances between the two TADs t_i and t_j measured from FISH experiments, respectively, and s_i and s_j stand for the coordinates of the centers of TADs t_i and t_j , respectively.

We ran GEM-FISH* 100 times on the datasets of Chrs 20, 21, 22, Xa, and Xi, and compared the results with those obtained using GEM-FISH. We observed that the modeling results from GEM-FISH and GEM-FISH* were quite close to each other (Fig. R3). Thus, we stucked to using the original version of GEM-FISH.

We have added detailed discussions about this point in the revised manuscript. Please refer to the last paragraph in Section 3.2 (highlighted in blue).

Minor comments

Comment 1: It would be good to include the computational efficiency of GEM-FISH (running time, memory usage) for different resolution/length of the chromosome.

Answer: Thank you for the comment. We first analyzed the running time needed by GEM-FISH to reconstruct a 3D model for each of Chrs 20, 21, 22, Xa, and Xi at the TAD-level resolution. As shown in Table R4, the running time for GEM-FISH to reconstruct TAD-level resolution models was in a range of minutes.

For reconstructing the intra-TAD models, the time taken by GEM-FISH depends on the resolution of the intra-TAD Hi-C maps used. Since the 3D structures of all TADs of a given chromosome can be calculated in parallel, the time taken to calculate all the intra-TAD models is the maximum running time for reconstructing the intra-TAD structures of that chromosome. As shown in Table R5, the time needed by GEM-FISH to calculate the most time-demanding intra-TAD models at 5 Kbp resolution for Chrs 20, 21, 22, Xa, and Xi was approximately 26 hours. Table R5 also summarizes the time needed to calculate the most time-demanding intra-TAD models at 10 Kbp, 25 Kbp, and 50 Kbp resolutions, respectively.

As for the memory usage, for the TAD-level-resolution modeling, the size of the largest matrix that needs to be stored in memory for the five tested chromosomes was 40×40 (Chrs Xa and Xi). For the intra-TAD modeling, the size of the Hi-C map depends mainly on the resolution used. In our computational experiments, with 5 Kbp resolution, the size of the Hi-C map corresponding to the largest TAD had 1576×1576 entries. In other words, the largest matrix needed around 20 Mb of memory.

According to the above analysis, GEM-FISH does not need extensive memory resource, and the final 3D models of the chromosomes at 5 Kbp resolution can be calculated within one day. We have added the above analysis of the computational efficiency of GEM-FISH to the revised manuscript (Section 5.5).

Comment 2: It is not clear what resolution was used for the 3D modeling.

Response: Thank you for the comment. GEM-FISH can calculate TAD-level resolution 3D models (i.e., each entry in the Hi-C map or the FISH distance matrix represents the contact frequency or the average distance between two TADs, respectively). In addition, GEM-FISH can calculate the high-resolution intra-TAD 3D models and integrate them into the TAD-level resolution model to obtain the final 3D structures of the chromosomes. In our experiments, we calculated the intra-TAD 3D models of chromosomes at 5 Kbp resolution. We have pointed this out in the revised manuscript.

Comment 3: What is the average size of the TADs? According to Table 2, there are about 30-40 TADs for each chromosome, which seems fewer than expected.

Response: Thank you for the comment. We followed the same TAD partition as in [7], which used the genomic locations of the TADs derived from Hi-C data in [8] and imaged the central 100 Kbp regions of those TADs. For Chrs 21 and 22, all the TADs were imaged. However, for Chrs 20 and X, a subset of the TADs that span the chromosomes at regular intervals was imaged. Table R6 summarizes the number of TADs in each of the five tested chromosomes (which was derived from [8]), the number of TADs imaged

in [7], and their average size.

Comment 4: It would be better to perform some colocalization test to show the loci within the same subcompartment tend to colocalize in the 3D models.

Response: Thank you for the comment. Reviewer 4 has raised a similar concern. As we described in our response to Reviewer 4 (point 4), we have included the genomic maps to show the genomic locations of subcompartments. We observed that the loci within the same subcompartment tend to colocalize even if they are far away along the genomic distance (Fig. R5, R6, and R7). We have added this result in the revised manuscript (please refer to the second paragraph in Section 2.5, highlighted in blue).

Comment 5: Similarly, could the authors provide some statistics (for example, radial distribution) to demonstrate that super-enhancers tend to lie on surface.

Response: Thank you for the comment. Following your suggestion, we calculated the normalized radial distances of super-enhancers in Chrs 20, 21, and 22, and compared them to those of regular enhancers in these three autosomes. As shown in Fig. R8, the analysis of the 3D models reconstructed by GEM-FISH demonstrated that super-enhancers tend to lie closer to the chromosome surface than regular enhancers. We have also added this analysis to the revised manuscript (please refer to the last paragraph in Section 2.7, highlighted in blue).

Comment 6: When calculating the accuracy of assigning TADs to compartments, is the ground truth here the assignment inferred from FISH or Hi-C data? If not Hi-C data, it would be good to compare with the compartment assignments called from Hi-C.

Response: Thank you for the comment. For analyzing the higher-order arrangements of sub-chromosomal regions, in general, FISH data are preferred because they directly offer the locations of interest at relatively long-range distances. Thus, we used the compartment assignments inferred from FISH data as the ground truth. For the sake of comparison, we also followed your suggestion and obtained the compartment assignments derived from Hi-C data. We found that the compartment assignments derived using Hi-C data were quite close to those obtained using FISH data (Fig. R9). Table R7 shows the number of TADs correctly assigned to A/B compartments when the ground truth was the assignment inferred from Hi-C data. This result was quite close to that obtained using the ground truth derived from FISH data (Table R7).

Reviewer 2

General comments

In this paper the authors describe an approach, GEM-FISH for integrating Hi-C data with FISH data to more reliably infer 3D model of the chromosome. This method extends the previous approach, GEM, developed by the authors which used only Hi-C data and biophysical constraints. The GEM method works by minimizing the KL divergence between the distribution of two pairs of regions to have high contact counts and the probability distribution two pairs of regions to be close in 3D space, subject to minimizing constraints imposed by biophysical properties. The authors here incorporate additional constraints derived from FISH data.

To assess the advantage of GEM-FISH over GEM, the authors compare the spatial distance inferred from the 3D models constructed to those from FISH, ability to place TADs in the right compartments, ability to recover 3D models for chrX and finally chromatin mark information for different subcompartments. Overall this seems like an interesting piece of work, but I feel that the advantage of the GEM-FISH over GEM is not significant. Most of the findings are validation of what has been shown and it is not clear what new insights are being derived. There are also several aspects of the paper that need clarification.

Response: Thank you for the nice summary. In the revised manuscript, we have added more validation tests with independent evidences, and also provided more descriptions about the new insights derived from our modeling results. Below please find our point-to-point response to your comments.

Major comments

Comment 1: How were TADs called in this method? How sensitive are the TAD calls to the robustness of the method?

Response: Thank you for the comment. In [7], the authors used the genomic locations of the TADs derived from Hi-C data in [8] and imaged the central 100 Kbp regions of these TADs. Since we used the FISH data obtained from [7], here we used the same TAD partition.

Comment 2: The within TAD models are not studied in a lot of detail and only examined in the subcompartment analysis which is done in an adhoc manner. In this analysis the type of region is defined in an arbitrary manner. It is not clear what 3 and 10 are, and the methods don't get into the details of this. Why not use what is published in Di Perro [39] or a simpler model?

Response: Thank you for the comment. In the revised manuscript, we have further performed comprehensive analyses on the within TAD models from our modeling results. More details can be found in Section 2.5 and also our response to point 2 of Reviewer 1's comments.

The purpose of section "Analyses of subcompartments" is to compare the foldings of the regions that belong to subcompartment B1 (that has a repressive nature with the enrichment of the H3K27me3 mark) and those belonging to subcompartment B2 (that

instead has an inactive nature). We divided the chromosome into loci of 50 Kbp length and assigned each locus with a discrete score ranging from 1 - 20 for different histone marks, in the same way as in Di Pierro et al. [9]. However, here we did not apply a recurrent neural network to classify these loci into different types of subcompartment as in [9], because this model did not yield high accuracy for this specific classification task. For instance, the accuracy of applying this neural network to classify loci into subcompartments A1, A2, B1, B2, and B3 were 72.8%, 71.8%, 61.4%, 19.6%, and 83.1% respectively (see Fig. S1 (B) in [9]). In other words, the neural network classifier had poor performance (only 19.6% accuracy) in classifying loci into subcompartment B2. Thus, here we adopted another scheme to determine the regions that carry the epigenomic content of subcompartment B1 or B2. More specifically, B2 subcompartment is depleted from the marks H3K27me3, H3K36me3, H3K27ac, H3K4me1, and H3K79me2 [6]. Thus, in our analysis, the loci with a discrete score < 3 (i.e., a score of at most ‘2’, which is equal to 0.1 of the maximum possible score) for all the marks mentioned above were considered belonging to subcompartment B2. Allowing the loci to have at most 0.1 of the maximum possible score was to account for the experimental noise that may affect the analysis. On the other hand, loci belonging to subcompartment B1 are generally positively correlated with the repressive mark H3K27me3 and negatively correlated with the active mark H3K36me3 [6]. Thus, we considered those loci with a score larger than or equal to ‘10’ (i.e., the median score) for the mark H3K27me3, and a score less than ‘3’ (i.e., at most 0.1 of the maximum possible score) for the mark H3K36me3 to belong to subcompartment B1.

To further verify the above simple approach for identifying subcompartments B1 and B2, we have also evaluated the chromatin states of our classified regions using an independent genome annotation strategy which was mainly adapted from [10]. More specifically, in [10], the authors defined the active regions as those enriched with the mark H3K79me3 or H3K4me2 and the repressed regions as those enriched with the mark H3K27me3. They also defined the inactive regions as those enriched with unmodified histone H3 and depleted of binding from PcG proteins and transcriptional activators. They calculated the enrichment of the mark H3K4me2 or H3K79me3 relative to H3 as $\log_2(\text{H3K4me2 reads}/\text{H3 reads})$ or $\log_2(\text{H3K79me3 reads}/\text{H3 reads})$. For the repressed domains, they calculated the enrichment of H3K27me3 as $\log_2(\text{H3K27me3 reads}/\text{H3 reads})$. For the inactive domains, they calculated the enrichment of unmodified histone H3 as $(1 - \log_2(\text{H3K4me2 reads}/\text{H3 reads}) - \log_2(\text{H3K79me3 reads}/\text{H3 reads}) - \log_2(\text{H3K27me3 reads}/\text{H3 reads}) - \log_2(\text{H3K9me2 reads}/\text{H3 reads}))$ [10]. For the cell line IMR90, the ChIP-Seq profiles for unmodified histone H3, H3K79me3 and H3K9me2 are not available on the ENCODE project web site. Thus, to define the active regions, we used H3K36me3 instead of H3K79me3. For the repressed regions, we used H3K27me3, and ‘Control’ from the ENCODE profiles as an alternative to unmodified histone H3. We then calculated the average enrichment of the marks H3K27me3, H3K36me3, H3K4me2, and unmodified histone H3 for the regions that we classified (using our simple method described previously) as sub-

compartments B1 and B2, using the same scheme as described in [10]. We found that the regions that we classified as carrying the epigenomic content of subcompartments B1 and B2 were relatively enriched with H3K27me3 and unmodified histone H3, respectively, and both were depleted of other marks (Table R8). This result was consistent with the repressive nature of subcompartment B1 and inactive nature of subcompartment B2, which thus supported the reasonableness of our simple scheme for classifying subcompartments B1 and B2. We have clarified this point in the revised manuscript (please refer to Section 2.5 and Section S3).

Comment 3: The analysis of super enhancers on the 3D model is interesting. It seems the exposed region where the super enhancers lie are gene rich. Are the genes expressed in IMR90?

Response: Thank you for the perspective comment. We have further investigated the gene expression values (in FPKM) of the genes in the q22.3 region of Chr21, and compared them with those in the other regions of the same chromosome. We found that the gene expression values of the genes in the q22.3 region are significantly higher than those in the other regions, which were referred to as control genes (Fig. R10). Also, taking the FPKM value ‘20’ as a threshold to classify the genes into ‘ON’ and ‘OFF’ states (as in [3]), we found that 12 genes in the region q22.3 are ‘ON,’ forming 38.7% of all the ‘ON’ genes in Chr21. In other words, we found that the density of the expressed genes in the q22.3 region is 2.17 expressed genes/Mbp vs. 0.59 expressed genes/Mbp in the other regions of Chr21. We have added this analysis result in the revised manuscript (please refer to the blue highlighted part in the third paragraph of Section 2.7).

Comment 4: The comparison of the TAD spatial distance estimation from GEM and GEM-FISH in figure 2 and supp figs S1, S2, S3 and S4 is not fair because GEM-FISH is using the spatial distance as a constraint which GEM is not.

Response: Thank you for pointing this out. Reviewer 1 has raised the same concern. As we described in our response to Reviewer 1 (Point 1), we have performed several additional tests to further verify our modeling results. First, to make a more fair comparison, we conducted an additional 10-fold cross-validation procedure to further evaluate the performance of our approach. In each fold, we built the average FISH distance matrix, denoted by F_1 , of a specific chromosome using only 90% of the available FISH data and used it to optimize the 3D models in GEM-FISH. Then, we evaluated the reconstructed structures using the average FISH distance matrix obtained from the remaining 10% of FISH data, denoted by F_2 , of that chromosome (which can be considered completely independent data). We also assessed the structures reconstructed by GEM against the same distance matrix F_2 . The average relative errors obtained are listed in Table R1.

In addition, we measured the asphericity values of the 3D models reconstructed by both GEM-FISH and GEM [1], based on the eigenvalues of their gyration tensors [2]. We observed that the 3D models reconstructed by GEM-FISH tended to be spherical in shape, while those reconstructed by GEM tended to be more extended (Table R2 and Fig. R1). A potential reason for these extended structures obtained by GEM is the relatively

weak restraints of Hi-C contact frequencies between those TADs far away in the genomic distance.

We also inspected the radial distributions of expressed genes in the final 3D models of Chrs 20, 21, and 22 obtained by GEM-FISH and GEM, respectively. We observed that the expressed genes tend to lie closer to the surface of the chromosome territories in the models reconstructed by GEM-FISH than those in the models reconstructed by GEM (Fig. R2). These observations are in agreement with the fact that active transcription regions tend to lie closer to the chromosome surface [3].

Moreover, we used a previous finding derived from an independent FISH dataset about the spatial arrangement of genomic loci related to the nuclear pore complex (NPC) component Nup153 [4,5], to further verify the reasonableness of our reconstructed 3D models. In particular, we first derived a number of genomic positions that are known to interact with Nup153 in Chrs 20, 21, and 22 from the DNA adenine methyltransferase identification (DamID) data [4]. We inspected the radial distribution of these genomic sites in the final 3D models of Chrs 20, 21, and 22 reconstructed by GEM-FISH and compared them to that of randomly selected loci of the same genomic lengths (denoted by Control-Nup loci). We observed that most of the genomic sites interacting with Nup153 generally tend to lie closer to the surface of the chromosome territories than the Control-Nup loci (Fig. R3). Nevertheless, we also found that some sites interacting with Nup153 lie relatively close to the center of the chromosome. This observation was also consistent with the fact derived from the previous FRAP (fluorescence recovery after photobleaching) analysis that Nup153 is a nuclear basket component with a stable subpopulation at the nuclear envelope-embedded NPC, and a dynamic subpopulation exchanging between the NPC and the nucleoplasm [5].

Furthermore, we have conducted a new FISH experiment to validate the relative spatial distances between a triplet of genomic loci and show that loci belonging to the same subcompartment tend to lie spatially closer to each other than those loci belonging to different subcompartments even at a larger genomic distance. In particular, we designed probes for three genomic loci, denoted by L1, L2, and L3, respectively, where the genomic distance between L1 and L2 is larger than that between L1 and L3 (Fig. R4(a)). The two loci L1 and L2 are depleted from the marks H3K27me3, H3K36me3, H3K27ac, H3K4me1, and H3K79me2, and hence considered belonging to subcompartment B2 [6]. On the other hand, the locus L3 is enriched with H3K27me3 and depleted from H3K36me3, and hence considered belonging to subcompartment B1 [6] (Fig. R4(a)). We examined the spatial distances between L1-L2 and between L1-L3 derived from this new FISH experiment, and found that the spatial distance between the two loci L1-L2 is consistently smaller than that between L1-L3 (Fig. R4(b) and R4(c)). This result provided a new experimental evidence to show that the loci belonging to the same subcompartment tend to locate spatially closer to each other.

All the above additional validation tests have further supported the superiority of our new modeling approach with independent evidences. We have added the results of all the

above tests to the revised manuscript (please refer to the highlighted parts in Section 2.2 and Section 2.5).

Comment 5: For the users of this tool, providing a sense of range of parameters would be very helpful. I think showing how the objective and the reconstructed model change as a function of different parameters would be beneficial.

Response: Thank you for the comment. Following your suggestion, we have performed this analysis in the revised manuscript. Fig. R11(a) shows the values of the scoring function when fixing $\lambda_E = 5 \times 10^{12}$ and changing λ_F from 1×10^{-6} to 1×10^{-10} . Similarly, Fig. R11(b) shows the values of the scoring function when fixing $\lambda_F = 1 \times 10^{-8}$ and changing λ_E from 5×10^{10} to 5×10^{14} .

To examine how different parameter settings may affect the reconstructed models, we ran GEM-FISH ten times with different combinations of parameters λ_E and λ_F . Fig. R12 shows the mean values of average relative errors obtained for Chrs 20, 21, and 22. Fig. R13 and R14 show the Pearson correlation coefficients between pairwise spatial distances between TADs derived from the 3D models reconstructed by GEM-FISH vs. inverse Hi-C contact frequencies and experimental FISH distances, respectively. We also calculated the harmonic means of Pearson correlation coefficients with both FISH distances and inverse Hi-C contact frequencies to measure the agreement between the reconstructed 3D models with both Hi-C and FISH data (Fig. R15). All these tests demonstrated that the values of our chosen parameters $\lambda_F = 1 \times 10^{-8}$ and $\lambda_E = 5 \times 10^{12}$ produced reasonably good modeling results. We have included these additional results on parameter selection in the SI appendix of the revised manuscript (see Section S2 “More analysis results on parameter selection”).

Comment 6: The authors don’t mention how fast the algorithm runs. Providing some numbers for estimating the within TAD and TAD-resolution model will helpful.

Response: Thank you for the comment. Table R4 shows the running time needed by GEM-FISH to reconstruct a 3D model at the TAD-level resolution for each of the chromosomes 20, 21, 22, Xa, and Xi. The time taken by GEM-FISH to reconstruct the intra-TAD models depends on the resolution of the intra-TAD Hi-C maps used. Since we can calculate the 3D structures of all TADs of a given chromosome in parallel, the time required for reconstructing all intra-TAD models is the maximum of running time for calculating the intra-TAD structures. Table R5 summarizes the time needed by GEM-FISH to calculate the most time-demanding intra-TAD models for Chrs 20, 21, 22, Xa, and Xi at 5 Kbp, 10 Kbp, 25 Kbp, and 50 Kbp resolutions, respectively. In the revised manuscript, we have added the above analysis about the computational efficiency of GEM-FISH (Section 5.5).

Comment 7: Equation 6, which norm is used?

Response: Thank you for your comment. In Equation 6, the L1 norm is used. We have clarified this point in the revised manuscript.

Comment 8: The authors show the reconstructed 3D models of entire chromosomes, but they also have this at the individual TAD level. It would be interesting to see the

structure of the individual TADs and how different chromatin marks are organized within individual TADs.

Response: Thank you for the perspective comment. Following your suggestion, we further investigated the 3D models reconstructed by GEM-FISH for individual TADs and the organizations of histone marks within them. We first calculated the relative enrichment of four histone marks H3K27ac, H3K4me1, H3K4me3, and H3K9me3 in individual TADs of Chrs 20, 21, and 22 using the same approach as described in [7]. More specifically, for every TAD, we calculated the read count density of each mark by dividing its read count by the genomic length of the corresponding TAD. After that, we divided the read count density of each mark in every TAD by the mean read count density of that mark along the whole chromosome to obtain its relative enrichment within the corresponding TAD. Tables R9, R10, and R11 show the relative enrichments of the four marks mentioned above within each TAD in the three autosomes. We focused on those TADs that are highly active or inactive and investigated their structures and how different histone marks are organized within them. We considered a TAD as highly enriched with a specific mark if its relative count density is at least three times its mean count density of the whole chromosome. Accordingly, TADs that are highly enriched with one or more of the active histone marks (i.e., H3K27ac, H3K4me1, and H3K4me3) were considered highly active (hereafter referred to as active TADs). Similarly, those TADs highly enriched with the mark H3K9me3 were considered highly inactive (hereafter referred to as inactive TADs).

For the two types of TADs that we focused on (i.e., active and inactive TADs), it was not surprising to see that most of the loci (of 5 Kbp size) within a TAD carry the same epigenetic nature of the whole TAD (Fig. R16). On the other hand, we found that some of the loci have read count densities less than the mean read count density of the whole chromosome for all of the four marks. Moreover, we found that few loci in the active TADs have inactive nature, and vice versa (Fig. R16). In addition, we noticed that the inactive TADs tend to have compact 3D models, while the 3D models of the active ones tend to be more open, which was consistent with previous findings [10] (Fig. R16). To quantitatively measure the compactness of the 3D models of both active and inactive TADs, we also calculated their mean asphericity values, which were 0.14 and 0.05 for active and inactive TADs, respectively. These results indicated that the inactive TADs tend to be more spherical, which was consistent with their compact 3D conformations, while the active TADs tend to be more extended, which was consistent with their open structures observed. We have added these results to the revised manuscript (please refer to the second paragraph in Section 2.6, highlighted in blue).

Comment 9: Is always a single 3D structure found, or can there be multiple equally good solutions? Does the FISH data enable to improve the stability?

Response: Thank you for the comment. We agree that due to the dynamic nature of chromosome structures and uncertainty in experimental data, it would be generally better for a chromosome structure modeling approach to compute an ensemble of chromosome conformations rather than just a single solution. In principle, in this study, we could

also derive an ensemble of 3D models for both TAD-level resolution conformations and intra-TAD conformations, using the same strategy as in GEM [11], in which an ensemble of multiple conformations with mixing proportions are computed. However, since GEM-FISH involves an assembly step to combine the intra-TAD conformations with the TAD-level resolution structure, the original optimization technique in GEM [11] for computing multiple conformations with mixing proportions cannot be directly used here to compute the final complete structures.

Nevertheless, to further evaluate the stability or uncertainty of the reconstructed 3D models, we first applied a strategy that has been widely used in popular structure modeling methods, such as Xplor-NIH [12] and Rosetta [13], to compute an ensemble of conformations. More specifically, we ran GEM-FISH and the baseline method GEM 100 times, starting from different random initial structures, and then evaluated the average relative errors with respect to the experimental FISH distances and the number of TADs correctly assigned to A/B compartments. This additional test demonstrated the good stability of the 3D models calculated by GEM-FISH (Table R12 and R13). We also computed an ensemble of TAD-level resolution models in GEM-FISH using the same optimization technique as in GEM [11]. We found that computing such an ensemble of 3D conformations can lead to a slight improvement in the modeling performance (Table R12, R13, and R14). We have added the above additional analysis results to the revised manuscript (please refer to the first two paragraphs in Section 3.2, highlighted in blue).

Minor comments

Comment 1: Label the color maps in Figure 2 to indicate this is error.

Response: Thank you for the comment. Following your suggestion, we have added a label to Fig. 2 and Fig. S2-S5 to indicate the meaning of each color map.

Comment 2: Supp fig 2-5 are not referred in the text

Response: Thank you for the comment. We have referred to Figures S1-S4 in the last sentence of the second paragraph in Section 2.2. As for Fig. S5 (Fig. S15 in the revised manuscript), we have referred to it in the last paragraph of Section 2.3 in the revised manuscript.

Reviewer 3

General comments

Transparency statement: this review was written by Guillaume Filion (CRG, Barcelona).

The manuscript of Ahmed Abbas et al. titled Integrating Hi-C and FISH data for modeling 3D organizations of chromosomes presents a novel method called GEM-FISH to reconstruct the 3D conformation of chromosomes from Hi-C and FISH data. The principle of GEM-FISH is to simultaneously maximize: i. the agreement with contact frequencies derived from Hi-C ii. the conformation energy and iii. the agreement with distances derived from FISH. It builds on a strategy initially developed in the GEM method (reference [32]) borrowing concepts from manifold learning. The authors show that GEM-FISH improves substantially the results of GEM in terms of reconstruction (measured by agreement with FISH, A/B compartments, chromatin marks or epigenetic state of the X chromosome).

I found the article extremely well written. This is a model of clarity and concision, and I highly encourage the authors to continue writing in this outstanding style. The concepts of using manifold learning is very interesting and it should be further explored in this field of research. Overall, the manuscript describes a method with great potential.

However, I found some issues in the manuscript. While none of them is fatal, fixing them may require major changes to the manuscript and / or the software, so the authors may prefer to submit to another journal if time is an issue. Below is the list of issues I think the authors should address. I hope that they will find it useful to improve their manuscript.

Response: Thank you for the nice summary. In the revised manuscript, we have conducted several additional experiments and made a number of changes to address your major concerns. Below please find our point-to-point responses to your comments.

Major comments

Comment 1: The authors acknowledge 25 citations presenting methods to reconstruct the 3D structure from Hi-C maps (references [1,8-31]), but they compare GEM-FISH to none of them, arguing that GEM already outperformed other state-of-the-art methods. Since this is a different data set, and since the reconstruction criteria are different in this study, the authors should again compare GEM-FISH to these state-of-the-art methods. This is important to really establish the performance of GEM-FISH versus to other methods. Since these other methods are more popular than GEM, it would also allow more readers to judge the quality of GEM-FISH.

Response: Thank you for the comment. In the revised manuscript, we have further compared the performance of our method to that of several other state-of-the-art modeling approaches, in addition to GEM. In particular, we first calculated the 3D models of the five chromosomes using classical multidimensional scaling (MDS) [14] with the average FISH distances as input. In addition, we calculated the 3D models of Chrs 20, 21, 22, and X using Shrec3D [15], chromosome3D [16], and ChromSDE [17] with the TAD-level-resolution Hi-C maps as input. We then compared the performance of GEM-FISH to that

of these four methods regarding the accuracy in assigning TADs to A/B compartments. We also compared the 3D models reconstructed by GEM-FISH to those obtained by MDS in terms of the average relative errors with respect to experimental FISH distances. For the sake of fair comparison, we did not look into the average relative errors with respect to the pairwise FISH distances for those 3D models obtained by Shrec3D, chromosome3D, and ChromSDE. Table R15 shows that GEM-FISH outperformed the other four state-of-the-art methods in terms of the accuracy in assigning TADs to A/B compartments, and surpassed MDS regarding the average relative errors for all the tested chromosomes. We also calculated the asphericity values for the 3D models obtained by GEM-FISH, MDS, Shrec3D, chromosome3D, and ChromSDE [1], based on the eigenvalues of their gyration tensors [2]. The high asphericity values for the 3D models obtained by Shrec3D, chromosome3D, and ChromSDE indicated that these conformations generally owned an extended shape rather than a spherical one (Table R15). A potential reason for these extended structures was the relatively weak Hi-C contact signals between those TADs far away along the genomic distances. In the revised manuscript, we have added a new section entitled “GEM-FISH outperforms the state-of-the-art methods for chromosome structure modeling” to show these comparison results (Section 2.4).

Comment 2: Several conformations are usually compatible with a set of restraints given by Hi-C and/or FISH. It seems that GEM deals with ensemble solutions, but GEM-FISH does not (as far as I understand, it uses gradient descent to find one local minimum of the cost function). The output is presented as the structure of the chromosome, but it may not even be the optimum. Even though there is some value in proposing a structure, it seems unreasonable to not quantify the uncertainty around this structure. Providing ensemble solutions is one way, but there may be others.

To summarize these last two points, from the perspective of a user, it is not acceptable to take the single solution provided by GEM-FISH as correct, based only on the comparison with GEM on one data set. Inference is always associated with a certain degree of certainty but in its present form, GEM-FISH neither shows that the performance is generalizable, nor does it provide any information regarding the confidence the user should have in the results.

Response: Thank you for the constructive comments. Reviewer 2 (point 9) also raised a similar concern. We agree that due to the dynamic nature of chromosome structures and uncertainty in experimental data, it would be generally better for a chromosome structure modeling approach to compute an ensemble of chromosome conformations rather than just a single solution. In principle, in this study, we could also derive an ensemble of 3D models for both TAD-level resolution conformations and intra-TAD conformations, using the same strategy as in GEM [11], in which an ensemble of multiple conformations with mixing proportions are computed. However, since GEM-FISH involves an assembly step to combine the intra-TAD conformations with the TAD-level resolution structure, the original optimization technique in GEM [11] for computing multiple conformations with mixing proportions cannot be directly used here to compute the final complete structures.

Nevertheless, to further evaluate the stability or uncertainty of the reconstructed 3D models, we first applied a strategy that has been widely used in popular structure modeling methods, such as Xplor-NIH [12] and Rosetta [13], to compute an ensemble of conformations. More specifically, we ran GEM-FISH and the baseline method GEM 100 times, starting from different random initial structures, and then evaluated the average relative errors with respect to the experimental FISH distances and the number of TADs correctly assigned to A/B compartments. This additional test demonstrated the good stability of the 3D models calculated by GEM-FISH (Table R12 and R13). We also computed an ensemble of TAD-level resolution models in GEM-FISH using the same optimization technique as in GEM [11]. We found that computing such an ensemble of 3D conformations can lead to a slight improvement in the modeling performance (Table R12, R13, and R14). We have added the above additional analysis results to the revised manuscript (please refer to the first two paragraphs in Section 3.2, highlighted in blue).

Comment 3: Section 4.3 explains how the mixing parameters of the cost function were chosen. This is based on a single data set at a single resolution of the Hi-C data (one can assume that the TAD resolution is always the same, but not the Hi-C). What are the reasons to think that these parameters are acceptable for other datasets at other resolutions? In the absence of this information, how can the user trust the solution? The authors should show that these values are robust across resolutions and datasets, and if they are not, provide a way for the user to obtain meaningful results in different conditions.

Response: Thank you for your comment. As addressed in our response to Reviewer 2 (point 5), in the revised manuscript, we have performed additional tests on GEM-FISH with different parameter settings. Fig. 11(a) shows the values of the scoring function when fixing $\lambda_E = 5 \times 10^{12}$ and changing λ_F from 1×10^{-6} to 1×10^{-10} . Similarly, Fig. 11(b) shows the values of the scoring function when fixing $\lambda_F = 1 \times 10^{-8}$ and changing λ_E from 5×10^{10} to 5×10^{14} .

To examine how different parameter settings may affect the reconstructed models, we ran GEM-FISH ten times with different combinations of parameters λ_E and λ_F . Fig. R12 shows the mean values of average relative errors obtained for Chrs 20, 21, and 22. Fig. R13 and R14 show the Pearson correlation coefficients between pairwise spatial distances between TADs obtained from the 3D models reconstructed by GEM-FISH vs. inverse Hi-C contact frequencies and experimental FISH distances, respectively. We also calculated the harmonic mean of Pearson correlation coefficients with both FISH distances and inverse Hi-C contact frequencies to measure the agreement between the reconstructed models with both Hi-C and FISH data (Fig. R15). All the above tests indicated that the values $\lambda_F = 1 \times 10^{-8}$ and $\lambda_E = 5 \times 10^{12}$ produced reasonably good modeling results.

To further test the robustness of the choice of parameters λ_E and λ_F on other Hi-C datasets, we also repeated all the above experiments using Hi-C data from [8] with resolution 40 Kbp. Fig. R17 - Fig. R20 show that the combination of parameters $\lambda_F = 1 \times 10^{-8}$ and $\lambda_E = 5 \times 10^{12}$ still yielded good results even on this different Hi-C

dataset. Thus, most likely our choice of the parameters is optimal and robust for Hi-C data at a range of resolution. In principle, the method provided in our manuscript (see Section “Parameter selection”) or a cross validation like procedure can be easily used to determine the optimal parameter setting for any input data. We have also included this additional analysis on parameter selection to the SI appendix in the revised manuscript (please refer to Section S2 “More analysis results on parameter selection”).

Comment 4: The models are validated by FISH data (see for instance Fig 2g and similar panels from supplement figures), but the same data was used to fit the model. This leads to a phenomenon known as overfitting that must be avoided. At the very least, the authors should use a leave-one-out strategy where they fit the model with all FISH data points except one, and then use only this point for validation. Alternatively, they could use k-fold cross-validation or any other approach to reduce overfitting. Ideally, they should test their model with orthogonal FISH data (from another laboratory for instance).

Response: Thank you for the perspective comment. As addressed in our responses to Reviewer 1 (point 1) and Reviewer 2 (point 4), we have conducted several additional tests with independent evidences to further validate the modeling performance of GEM-FISH. First, to make a more fair comparison, we conducted an additional 10-fold cross-validation procedure to further evaluate the performance of our approach. In each fold, we built the average FISH distance matrix, denoted by F_1 , of a specific chromosome using only 90% of the available FISH data and used it to optimize the 3D models in GEM-FISH. Then, we evaluated the reconstructed structures using the average FISH distance matrix obtained from the remaining 10% of FISH data, denoted by F_2 , of that chromosome (which can be considered completely independent data). We also assessed the structures reconstructed by GEM against the same distance matrix F_2 . The average relative errors obtained are listed in Table R1.

In addition, we measured the asphericity values of the 3D models reconstructed by both GEM-FISH and GEM [1], based on the eigenvalues of their gyration tensors [2]. We observed that the 3D models reconstructed by GEM-FISH tended to be spherical in shape, while those reconstructed by GEM tended to be more extended (Table R2 and Fig. R1). A potential reason for these extended structures obtained by GEM is the relatively weak restraints of Hi-C contact frequencies between those TADs far away in the genomic distance.

We also inspected the radial distributions of expressed genes in the final 3D models of Chrs 20, 21, and 22 obtained by GEM-FISH and GEM, respectively. We observed that the expressed genes tend to lie closer to the surface of the chromosome territories in the models reconstructed by GEM-FISH than those in the models reconstructed by GEM (Fig. R2). These observations are in agreement with the fact that active transcription regions tend to lie closer to the chromosome surface [3].

Moreover, we used a previous finding derived from an independent FISH dataset about the spatial arrangement of genomic loci related to the nuclear pore complex (NPC) com-

ponent Nup153 [4,5], to further verify the reasonableness of our reconstructed 3D models. In particular, we first derived a number of genomic positions that are known to interact with Nup153 in Chrs 20, 21, and 22 from the DNA adenine methyltransferase identification (DamID) data [4]. We inspected the radial distribution of these genomic sites in the final 3D models of Chrs 20, 21, and 22 reconstructed by GEM-FISH and compared them to that of randomly selected loci of the same genomic lengths (denoted by Control-Nup loci). We observed that most of the genomic sites interacting with Nup153 generally tend to lie closer to the surface of the chromosome territories than the Control-Nup loci (Fig. R3). Nevertheless, we also found that some sites interacting with Nup153 lie relatively close to the center of the chromosome. This observation was also consistent with the fact derived from the previous FRAP (fluorescence recovery after photobleaching) analysis that Nup153 is a nuclear basket component with a stable subpopulation at the nuclear envelope-embedded NPC, and a dynamic subpopulation exchanging between the NPC and the nucleoplasm [5].

Furthermore, we have conducted a new FISH experiment to validate the relative spatial distances between a triplet of genomic loci and show that loci belonging to the same subcompartment tend to lie spatially closer to each other than those loci belonging to different subcompartments even at a larger genomic distance. In particular, we designed probes for three genomic loci, denoted by L1, L2, and L3, respectively, where the genomic distance between L1 and L2 is larger than that between L1 and L3 (Fig. R4(a)). The two loci L1 and L2 are depleted from the marks H3K27me3, H3K36me3, H3K27ac, H3K4me1, and H3K79me2, and hence considered belonging to subcompartment B2 [6]. On the other hand, the locus L3 is enriched with H3K27me3 and depleted from H3K36me3, and hence considered belonging to subcompartment B1 [6] (Fig. R4(a)). We examined the spatial distances between L1-L2 and between L1-L3 derived from this new FISH experiment, and found that the spatial distance between the two loci L1-L2 is consistently smaller than that between L1-L3 (Fig. R4(b) and R4(c)). This result provided a new experimental evidence to show that the loci belonging to the same subcompartment tend to locate spatially closer to each other.

All the above additional validation tests have further supported the superiority of our new modeling approach with independent evidences. We have added the results of all the above tests to the revised manuscript (please refer to the highlighted parts in Section 2.2 and Section 2.5).

Comment 5: Another point related to the validation by FISH data is that Fig 2g and similar panels from supplement figures show a black curve labelled Exp. FISH that is perfectly smooth. The legend says that they are derived from the experimental FISH data, but I could not find the description of this derivation. Why not show the original data points? With the current representation, there is no information of the variability around the fitted line. It would also be interesting to see whether the line is better at fitting one outlier to the detriment of all other points. The authors should either represent the original data, or clearly explain why they think it is better not to. Either way, they

should explain what the fitted line is.

Response: Thank you for the comment. The fitting curves were obtained by fitting the data to a power-law function $S = a \times G^b$, where S is the spatial distance between two TADs, and G is the genomic distance between the centers of these two TADs. The coefficients a and b were obtained using the ‘fit’ function in Matlab.

The original data points and the fitting curves with their equations for experimental FISH data, GEM-FISH, and GEM models for different chromosomes are shown in Fig. R21 - R25. We have added these figures to the revised manuscript (please refer to Fig. S4-S9).

Comment 6: Figure 4 is interesting but it cannot be used for validation. Chromatin marks correlate with A/B compartments, but since the authors can compute the A/B compartments from the original Hi-C map, they learn nothing new about the structure of the chromosome by comparing the distributions of histone marks. The authors may present this result, but not as a validation of GEM-FISH. Instead, they should include tests that are orthogonal to the ones they have already included.

Response: Thank you for the comment. The purpose of Fig. 4 was to demonstrate that the 3D models computed by GEM-FISH displayed close relative enrichment patterns of different epigenetic marks in A/B compartments to those derived from experimental FISH data. This observation indicated that the few TADs that were wrongly assigned to the A and B compartments in the models reconstructed by GEM-FISH probably had more noisy epigenetic properties of one compartment over the other. In the revised manuscript, we have followed your suggestion and moved Fig. 4 to the SI appendix (Fig. S14). We have also added more validation tests with independent evidences as we described above in our response to Comment 4.

Comment 7: In section 2.3 the authors claim that GEM-FISH performs better than GEM on the X chromosome because FISH allowed to distinguish the two X chromosomes. The argument is interesting and it could be presented as a main feature of GEM-FISH (because the heterozygosity is probably not high enough to distinguish homologs by Hi-C in most cell lines). But in the current manuscript, it does not speak in favor of GEM-FISH against GEM: there exists allele-specific Hi-C data in mouse, so if the authors want to show that the GEM-FISH method surpasses the GEM method, they should test both on such data. Alternatively, they could rewrite the manuscript in order to show that the main advantage of GEM-FISH is that it makes better use than GEM of existing FISH data.

Response: Thank you for your suggestions. As generally it is not easy to collect a set of allele-specific mouse Hi-C data also with available large-scale FISH data, here we followed your latter suggestion and emphasized that one of the main advantages of GEM-FISH is that it can fully exploit existing FISH data to better calculate 3D models that are consistent with both Hi-C and FISH data, and biophysically reasonable at the same time (please refer to the blue highlighted part in the fourth paragraph of Section 2.3 in the revised manuscript). In addition to the advantage of being able to model the

structures of allele-specific chromosomes due to the nature of FISH data, as demonstrated through the tests on X chromosomes, we also showed that GEM-FISH displayed better modeling performance than GEM (which uses only Hi-C data as input) on Chrs 20, 21, and 22 (as shown in Table R1).

Comment 8: Section 2.6 about the super enhancers does not show any control. I would suggest to remove this section and the associated results altogether, unless the authors can show some evidence that the results they observe are specific to super enhancers versus typical enhancers. If they can show these controls, it would be appropriate to substantiate the claim that These novel patterns of the spatial distributions of super-enhancers can (...) provide important insights into revealing their functional roles in gene regulation... (last sentence of the section). The authors should explain in the discussion what kind of insight they are talking about. If the authors cannot or do not want to explain why this pattern is important, they should remove the claim.

Response: Thank you for your constructive comments. In the revised manuscript, we have further investigated this problem by adding the control analysis. Following your suggestion, we have also compared the spatial distribution of super-enhancers to that of regular enhancers. More specifically, we compared the normalized radial distances of super-enhancers to those of regular enhancers for Chrs 20, 21, and 22 (Fig. R8).

In this additional analysis, the list of regular enhancers in human genome was obtained from the FANTOM5 project [18,19]. To obtain the normalized radial distance of a certain point P corresponding to a specific locus of size 5 Kbp in our reconstructed 3D models, we first calculated the centroid of the 3D model, denoted as C . We then obtained R , the Euclidean distance of the farthest point in the 3D model from the centroid. After that, we divided the Euclidean distance between the two points P and C by the largest distance R to obtain the normalized radial distance of point P .

As shown in Fig. R8, the analysis on our reconstructed 3D models demonstrated that super-enhancers tend to lie closer to the chromosome surface than regular enhancers. Since super-enhancers are usually associated with cell-type-specific genes, their tendency to lie close to the chromosome surface, and hence their higher accessibility relative to regular enhancers, supported the hypothesis that local interactions between genomic loci are the driving forces that lead to particular chromosome conformations [3, 20, 21]. We think this finding is quite interesting and can provide useful insights into understanding the functional roles of super-enhancers. Thus, we kept this section in the revised manuscript. We have added the above analysis to the revised manuscript (please refer to the last paragraph in Section 2.7, highlighted in blue).

Comment 9: What is the evidence that there is no bug in GEM-FISH, or that he code actually does what the authors claim? The authors should provide some test code to show some evidence that the code is correct.

Response: Thank you for the comment. We have followed your suggestion and added a test code and a detailed Readme file describing how to use it to the program on: <https://github.com/ahmedabbas81/GEM-FISH>.

Minor comments

Comment 1: There is no discussion of the computational performance of GEM-FISH and of the hardware requirements. Even ballpark estimates would be useful. Due to the availability of FISH data, the largest chromosome used here is the X. It would be interesting to know if chromosome 1 can be modeled at the same Hi-C resolution.

Response: Thank you for the comment. Reviewer 1 (minor point 1) and Reviewer 2 (point 6) have raised a similar question. Table R4 shows the running time needed by GEM-FISH to reconstruct a 3D model at TAD-level resolution for each of Chrs 20, 21, 22, Xa, and Xi.

For reconstructing the intra-TAD models, the time taken by GEM-FISH depends on the resolutions of the intra-TAD Hi-C maps used. Since the 3D structures of all TADs of a chromosome can be calculated in parallel, we only need to focus on the maximum of running time needed by GEM-FISH to calculate individual intra-TAD structures. Table R5 summarizes the time needed by GEM-FISH to calculate the most time-demanding intra-TAD models at 5 Kbp, 10 Kbp, 25 Kbp, and 50 Kbp resolutions, respectively, for Chrs 20, 21, 22, Xa, and Xi.

In our framework, since the formats of scoring functions for computing the TAD-level resolution and intra-TAD models are similar to each other (see Eq. (1) and Eq. (8)) and the same optimization technique is used to solve them, their running time is basically comparable to each other. As for Chr1, it contains 237 TADs [8]. According to our previous analysis on the time required by GEM-FISH to construct the intra-TAD resolution model of Chr20, which contains 344 genomic loci of 5 Kbp size (Table R5), it is estimated that computing the TAD-level resolution model of Chr1 should take time in a range of tens of minutes.

In the revised manuscript, we added an analysis of the computational efficiency of GEM-FISH (Section 5.5).

Comment 2: In the Code availability section, it should be specified that GEM-FISH is written in Matlab (among others because Matlab is nonfree, so this is relevant for availability).

Response: Thank you for the comment. We have clarified this point in the “Data Availability” section.

Comment 3: The instructions on the page of GEM-FISH are unclear. For instance it says Download the program but there are over 40 files and none of them is obviously the program. Also, GitHub repositories are usually cloned, so the meaning of download is not clear. Statements like go into the folder are ambiguous, and so are statements like run the m-file. In all these cases, the authors should show the commands the users have to write.

Response: Thank you for the comment. We have added a more detailed Readme file with clear instructions for the users to use the program.

Reviewer 4

General comments

In this manuscript, the authors developed a computational framework to integrate Hi-C and multiplexed FISH data to generate 3D models of chromosome organization. Briefly, the authors took advantage of both Hi-C and FISH data to model chromosome conformation at the TAD-to-chromosome length scale, then used HiC data to guide the modeling at sub-TAD length scale, and finally combined the models at the two scales together using HiC data. This framework performs significantly better in capturing the correct large-scale organization than previous modeling using HiC data alone, and retains the ability to model sub-TAD structures without FISH data at the sub-TAD resolution. Using this framework, the authors obtained a series of structural features consistent with previous reports, and further reported new findings regarding the spatial localization of super enhancers and the G-band q22.3 of Chr21. The study is innovative, technically sound and clearly presented, and represents a significant progress in the computational modeling of chromosome organization that would be of broad interest to the genome biology and cell biology fields. I would recommend this manuscript for publication in Nature Communications as long as the following points are addressed:

Response: Thank you for the nice summary. In the revised manuscript, we have made the corresponding changes and added several new analyses to address your concerns. Below please find our point-to-point response to your comments.

Major comments

Comment 1: The authors clearly demonstrated that at the TAD-to-chromosome level, combing Hi-C data and FISH data in the modeling is better than using Hi-C data alone. But it is unclear how much the modeling at this scale benefits from the inclusion of Hi-C data in the GEM-FISH strategy. In other words, during the optimization of the cost function C_g , does the third term with C_3 dominate this process? If the authors use FISH data alone to model large scale chromosome conformation, will that give comparable results to the current GEM-FISH results? I would love to see in Fig. 2 comparisons among GEM-FISH, Hi-C only and FISH-only modeling. This comparison will not undermine the value of this work even if it shows that the FISH-only modeling is sufficient to address the large-scale fitting, but will only make the work more thorough and clear.

Response: Thank you for pointing this out. The goal of GEM-FISH is to reconstruct the 3D structures of chromosomes that are consistent with both FISH and Hi-C data, and biophysically stable at the same time. Following your suggestion, we have also implemented a modified version of GEM-FISH that used FISH data alone (i.e., without using the Hi-C data constraints). We ran this program ten times for each chromosome and compared the reconstructed 3D models with those obtained by GEM-FISH. We found that the average relative errors with respect to experimental FISH data were almost the same as in GEM-FISH for all five chromosomes (Table R16). However, the Pearson correlation coefficients between the inverses of Hi-C contact frequencies and the corresponding spatial distances derived from the 3D models reconstructed by GEM-FISH were significantly

higher than those with respect to the 3D models reconstructed using only FISH data (Fig. R26). These results demonstrated the main advantage of GEM-FISH in ensuring the consistency between the reconstructed models and both Hi-C and FISH data.

We have demonstrated this point in the revised manuscript (please refer to the last paragraph in Section 2.2).

Comment 2: Another analysis that could be very helpful is to show how much the individual chromosome conformations measured by the multiplexed FISH deviate from the authors averaged modeling result. This analysis could help evaluate how stereotypic/dynamic the individual chromosome conformations are at this scale.

Response: Thank you for the perspective comment. Following your suggestion, we have performed the following additional analysis in the revised manuscript. We first calculated the volumes of the minimum bounding boxes that enclosed the spatial positions of TADs experimentally measured in [7] for individual copies of the five chromosomes. We then grouped the copies of each chromosome measured in the FISH experiments into ten groups according to their volumes and calculated the average distance matrix for each group. After that, we compared the derived average distance matrix of each group with the average 3D model reconstructed by GEM-FISH. We found that in general the more the number of structures that each group contained, the less the relative error obtained when comparing the average 3D model reconstructed by GEM-FISH with the average distance matrix of that group (Fig. R27 - Fig. R31).

We have added detailed discussions about this point in the revised manuscript (please refer to the third paragraph in Section 3.2).

Comment 3: The authors should state why they used the KL divergence function to define C1 but instead used the sum-square function for C3. In both cases the authors are essentially comparing the differences between two distributions, so why not use the KL divergence for C3 as well, or adopt the sum-square function in the C1 definition?

Response: Thank you for the comment. In C1, we are indeed comparing the difference between two distributions. However, these two distributions are in different domains. The first distribution is in the Hi-C contact frequency domain, while the second one is in the Euclidean distance domain. It is generally not appropriate to use the L2 norm to measure their difference, as they are associated with different units. On the other hand, in C3, the two distributions are both in the Euclidean domain. Thus, in this case we preferred to use the L2 norm in C3, because, unlike the KL divergence, L2 norm can preserve the scale (i.e., the output is expected to be of the same scale as the input).

Comment 4: On Page 22, in order to infer the radius of gyration of individual TADs, the authors first plotted the distributions of volumes and sizes for individual copies of chromosomes, and selected the values corresponding to the highest values to be the expected volume and size of that chromosome. What is the rationale for choosing the highest values of the distributions as the expected values here, rather using the means or medians?

Response: Thank you for your comment. We selected the values of chromosome

volume and size corresponding to the highest scores because they were represented by the highest number of copies of chromosomes in the population. Yet, it was still just an estimate of the true volume or size to derive an approximate value of the radius of gyration for individual TADs.

Following your suggestion, we have further conducted an additional test, in which we also used both the mean and median values of volumes and sizes to infer the values of R_g for individual TADs. We found that the average relative errors with respect to experimental FISH data for the final 3D models reconstructed by GEM-FISH were close to each other when using the mode, mean, and median values to estimate the values of R_g for individual TADs (Table R17). We have added the result of this additional test to the revised manuscript (please refer to the last paragraph in Section 5.1, highlighted in blue).

Comment 5: One potential oversimplification in this framework could be the assumption that adjacent TADs are always well insulated and do not overlap. Recent super-resolution imaging investigation of TADs showed that adjacent TADs could substantially overlap, depending on the epigenetic states of the TADs (Nature 529 418-422 (2016)). I am not sure if adjacent TADs are allowed to overlap in the authors framework here. It would be great if the authors could clarify. If such an assumption was implemented, maybe the authors could discuss potential ways to improve this front in their future work.

Response: Thank you for the perspective comment. In principle, two adjacent TADs are allowed to overlap in GEM-FISH. The overlapping fraction between two adjacent TADs will depend on the locations of their centers and the shapes of their 3D structures. Following the same definition as in [10], here we defined those overlapping regions as the fractions of loci in one TAD that are within 80 nm (spatial distance) from the nearest loci in the adjacent TAD. We looked into the fractions of overlapping between sequential TADs along the genomic distance in Chrs 21 and 22, in which almost every pair of adjacent TADs were imaged in the original FISH data [7] that we used. We observed that adjacent TADs along the genomic distance that belong to the same compartment tend to display higher overlapping than those that belong to different compartments (Table R18), which was consistent with the previous findings [10].

We have added this analysis result in the revised manuscript (please refer to the last paragraph in Section 2.6, highlighted in blue).

Minor comments

Comment 1: On Page 3 the authors stated that Dixon et al [4], introduced the concept of topologically associated domains (TADs). TADs were independently discovered by four groups at about the same time: J. R. Dixon et al., Nature 485, 376380 (2012). E. P. Nora et al., Nature 485, 381385 (2012). T. Sexton et al., Cell 148, 458472 (2012). C. Hou, L. Li, Z. S. Qin, V. G. Corces, Mol. Cell 48, 471484 (2012). I recall the very terminology of TADs was first used in the Nora et al paper. The authors should cite all four papers.

Response: Thank you for the comment. We have added these four references to the revised manuscript.

Comment 2: On Page 9, Equation 7 is missing a square sign on the right end, given the common definition of radius of gyration.

Response: Thank you for the comment. We have corrected this typo in the revised manuscript.

Comment 3: On Page 14, the authors pointed out the lack of allele-specific Hi-C data for distinguishing ChrXa and ChrXi in IMR90 cells. How did the authors still manage to get 3D models calculated by GEM (which only takes Hi-C data as input) for ChrXa and ChrXi to perform the calculations in Table 2?

Response: Thank you for the comment. Reviewer 3 (point 7) has raised a similar point. For GEM, we calculated only one 3D model for ChrX. In Table 2 (and also in other related tables), we compared the results obtained from the reconstructed 3D model of ChrX to those obtained from the FISH data for both the two copies of ChrX (i.e., Chrs Xa and Xi). In the revised manuscript, we have emphasized that a main advantage of GEM-FISH lies in being able to fully exploit the available FISH data to derive more accurate higher-order arrangements of sub-chromosomal regions, especially for the X chromosomes.

Comment 4: On Page 17 and Fig. 7, the authors tried to show that a certain sub-compartment tend to colocalize in 3D models reconstructed by GEM-FISH. How much of this colocalization is due to the genomic proximity among regions in the same sub-compartment? Do regions far away on the genomic map still colocalize in a subcompartment? The authors should include a genomic map of the subcompartments and identify the genomic regions on the 3D model.

Response: Thank you for the perspective comment. Following your suggestion, we included the genomic map to show the genomic locations of subcompartments. Fig. R5, R6, and R7 show that those regions far away along the genomic distances still tend to colocalize in the 3D models of the chromosomes. We have added this result in the revised manuscript (please refer to the second paragraph in Section 2.5, highlighted in blue).

Comment 5: On Page 19, the authors stated that Hi-C contact frequency between a pair of genomic loci is normally inversely proportional to their spatial distance measured by FISH. I think the authors meant to say Hi-C contact frequency between a pair of genomic loci is normally negatively correlated with their spatial distance measured by FISH, since the exact scaling relationship is inversely proportional to the 4th power as the authors cited before.

Response: Thank you for the comment. We have corrected this statement.

Comment 6: The authors should quantify the surface distribution of the super-enhancers, and compare them with randomly picked regions of the same size in the 3D models.

Response: Thank you for the comment. Following your suggestion, we compared the radial distributions of super-enhancers to those of the randomly picked genomic loci of the

same sizes (which are referred to as “Control-SE”). We observed that super-enhancers tend to lie closer to the chromosome surface than the Control-SE loci (Fig. R32). In addition, as we described in our response to Reviewer 3 (point 8), we also compared the radial distributions of super-enhancers to those of regular enhancers. We also found that super-enhancers tend to lie closer to the chromosome surface than regular enhancers (Fig. R8). We have added this analysis to the revised manuscript (please refer to the last paragraph in Section 2.7, highlighted in blue).

Comment 7: On Page 22, the authors wrote n stands for the number of cells. n should be the total measured copies of the chromosome, given that IMR-90 is a diploid cell line.

Response: Thank you for the comment. Based on your suggestion, we changed the original description to “where n stands for the number of imaged copies of Chr_i ”.

Data Availability

Gene expression data for IMR90 cell line can be downloaded from NCBI GEO GSE87831.

References

1. Cremer T, Cremer M (2010) Chromosome territories. *Cold Spring Harbor Perspectives in Biology* 2.
2. Rudnick J, Gaspari G (1987) The shapes of random walks. *Science* 237: 384–389.
3. Di Pierro M, Zhang B, Aiden EL, Wolynes PG, Onuchic JN (2016) Transferable model for chromosome architecture. *Proceedings of the National Academy of Sciences* 113: 12168–12173.
4. Ibarra A, Benner C, Tyagi S, Cool J, Hetzer MW (2016) Nucleoporin-mediated regulation of cell identity genes. *Genes and Development* 30: 2253–2258.
5. Rabut G, Doye V, Ellenberg J (2004) Mapping the dynamic organization of the nuclear pore complex inside single living cells. *Nature Cell Biology* 6: 1114–1121.
6. Rao S, Huntley M, Durand N, Stamenova E, Bochkov I, et al. (2014) A 3D map of the human genome at kilobase resolution reveals principles of chromatin looping. *Cell* 159: 1665–1680.
7. Wang S, Su JH, Beliveau BJ, Bintu B, Moffitt JR, et al. (2016) Spatial organization of chromatin domains and compartments in single chromosomes. *Science* 353: 598–602.
8. Dixon JR, Selvaraj S, Yue F, Kim A, Li Y, et al. (2012) Topological domains in mammalian genomes identified by analysis of chromatin interactions. *Nature* 485: 376–380.
9. Di Pierro M, Cheng RR, Lieberman Aiden E, Wolynes PG, Onuchic JN (2017) De novo prediction of human chromosome structures: Epigenetic marking patterns encode genome architecture. *Proceedings of the National Academy of Sciences* 114: 12126–12131.
10. Boettiger AN, Bintu B, Moffitt JR, Wang S, Beliveau BJ, et al. (2016) Super-resolution imaging reveals distinct chromatin folding for different epigenetic states. *Nature* 529: 418–422.

11. Zhu G, Deng W, Hu H, Ma R, Zhang S, et al. (2018) Reconstructing spatial organizations of chromosomes through manifold learning. *Nucleic Acids Research* : gky065.
12. Schwieters CD, Kuszewski JJ, Tjandra N, Clore GM (2003) The xplor-nih nmr molecular structure determination package. *Journal of Magnetic Resonance* 160: 65 - 73.
13. Rohl CA, Strauss CE, Misura KM, Baker D (2004) Protein structure prediction using rosetta. In: *Numerical Computer Methods, Part D*, Academic Press, volume 383 of *Methods in Enzymology*. pp. 66 - 93. doi:[https://doi.org/10.1016/S0076-6879\(04\)83004-0](https://doi.org/10.1016/S0076-6879(04)83004-0). URL <http://www.sciencedirect.com/science/article/pii/S0076687904830040>.
14. Torgerson WS (1952) Multidimensional scaling: I. theory and method. *Psychometrika* 17: 401–419.
15. Lesne A, Riposo J, Roger P, Cournac A, Mozziconacci J (2014) 3D genome reconstruction from chromosomal contacts. *Nature Methods* 11: 1141-1143.
16. Adhikari B, Trieu T, Cheng J (2016) Chromosome3d: reconstructing three-dimensional chromosomal structures from hi-c interaction frequency data using distance geometry simulated annealing. *BMC Genomics* 17: 886.
17. Zhang Z, Li G, Toh KC, Sung WK (2013) 3d chromosome modeling with semi-definite programming and hi-c data. *Journal of Computational Biology* 20: 831-846.
18. Lizio M, Harshbarger J, Shimoji H, Severin J, Kasukawa T, et al. (2015) Gateways to the fantom5 promoter level mammalian expression atlas. *Genome Biology* 16: 22.
19. <http://fantom.gsc.riken.jp/5/datafiles/latest/extra/Enhancers/>.
20. Stevens TJ, Lando D, Basu S, Atkinson LP, Cao Y, et al. (2017) 3D structures of individual mammalian genomes studied by single-cell Hi-C. *Nature* 544: 59-64.
21. Gürsoy G, Liang J (2016) Three-dimensional chromosome structures from energy landscape. *Proceedings of the National Academy of Sciences* 113: 11991–11993.
22. Pettersen EF, Goddard TD, Huang CC, Couch GS, Greenblatt DM, et al. (2004) UCSF Chimera - A visualization system for exploratory research and analysis. *Journal of Computational Chemistry* 25: 1605-1612.

Table R1. Mean and standard deviation values of average relative errors when testing the 3D models reconstructed by GEM-FISH (using Hi-C data and only 90% of the available FISH data) and GEM (using only Hi-C data) against the remaining 10% of the available FISH data. In GEM, due to the availability of only the non-allele-specific Hi-C maps (which did not distinguish between ChrXa and ChrXi) for the IMR90 cell line [6], we only considered one 3D model for the X chromosome, and computed the average relative errors with respect to experimental FISH distances for ChrXa and ChrXi, respectively.

	GEM-FISH	GEM
Chr20	0.210 ± 0.014	0.352 ± 0.037
Chr21	0.188 ± 0.010	0.362 ± 0.026
Chr22	0.205 ± 0.011	0.237 ± 0.007
ChrXa	0.217 ± 0.021	1.407 ± 0.096
ChrXi	0.258 ± 0.016	1.995 ± 0.115

Table R2. Mean and standard deviation of asphericity values for the 3D models reconstructed by GEM-FISH and GEM for Chrs 20, 21, and 22.

	GEM-FISH	GEM
Chr20	0.026 ± 0.003	0.187 ± 0.009
Chr21	0.025 ± 0.001	0.308 ± 0.018
Chr22	0.027 ± 0.002	0.102 ± 0.005

Table R3. Mean and standard deviation values of average relative errors, numbers of correctly assigned TADs to A/B compartments, and asphericity values for the 3D models generated by GEM-FISH and GEM-FISH* (which also took into account the cell-to-cell variability of FISH data), respectively.

	Average relative error (for 100 runs)		Number of TADs correctly assigned to A/B compartments (for 100 runs)		Asphericity (for 100 runs)	
	GEM-FISH	GEM-FISH*	GEM-FISH	GEM-FISH*	GEM-FISH	GEM-FISH*
Chr20	0.178 ± 0.002	0.183 ± 0.002	26.89 ± 0.72	27.09 ± 0.933	0.025 ± 0.003	0.034 ± 0.005
Chr21	0.162 ± 0.002	0.164 ± 0.002	31.92 ± 0.47	32.41 ± 0.933	0.024 ± 0.002	0.030 ± 0.004
Chr22	0.175 ± 0.003	0.176 ± 0.002	21.86 ± 1.26	21.69 ± 0.872	0.027 ± 0.003	0.033 ± 0.003
ChrXa	0.173 ± 0.001	0.175 ± 0.001	38.96 ± 0.71	38.01 ± 1.04	0.071 ± 0.002	0.071 ± 0.003
ChrXi	0.222 ± 0.001	0.224 ± 0.001	36.19 ± 1.48	35.67 ± 1.74	0.005 ± 0.001	0.007 ± 0.001

Table R4. Time needed by GEM-FISH to reconstruct the 3D models of Chrs 20, 21, 22, Xa, and Xi at the TAD-level resolution on a server with 32 2.40GHz Intel(R) Xeon(R) CPUs and 128GB RAM.

	Number of TADs	Time
Chr20	30	55.4 sec
Chr21	34	91.1 sec
Chr22	27	82.5 sec
ChrXa	40	54.1 sec
ChrXi	40	67.1 sec

Table R5. Time needed by GEM-FISH to reconstruct the 3D models of the most time-demanding intra-TAD models for Chrs 20, 21, 22, Xa, and Xi at 5 Kbp, 10 Kbp, 25 Kbp, and 50 Kbp resolutions, respectively. The computations were carried on a server with 32 2.40GHz Intel(R) Xeon(R) CPUs and 128GB RAM.

	TAD ID	TAD size	5 Kbp resolution		10 Kbp resolution		25 Kbp resolution		50 Kbp resolution	
			Number of loci	Running time	Number of loci	Running time	Number of loci	Running time	Number of loci	Running time
Chr20	TAD 19	1.72 Mbp	344	0.46 hr	172	5.72 min	69	0.75 min	35	0.48 min
Chr21	TAD 2	2.88 Mbp	576	7.96 hr	288	77.41 min	116	4.86 min	58	1.67 min
Chr22	TAD 6	3.12 Mbp	624	1.03 hr	312	9.28 min	125	2.46 min	63	1.32 min
ChrXa	TAD 25	7.88 Mbp	1576	21.02 hr	788	69 min	316	7.41 min	158	2.76 min
ChrXi	TAD 25	7.88 Mbp	1576	25.21 hr	788	89.45 min	316	7.74 min	158	2.46 min

Table R6. The numbers and average sizes of TADs of the five chromosomes used in our computational experiments. The first column gives the chromosome number. The second column gives the original numbers of TADs derived from Hi-C data in [8]. The third and fourth columns list the numbers and average sizes of the TADs imaged in [7], respectively.

	Original number of TADs	Number of TADs imaged	Average TAD size
Chr20	60	30	0.97 Mbp
Chr21	34	34	0.93 Mbp
Chr22	27	27	1.16 Mbp
ChrXa	86	40	2.0 Mbp
ChrXi	86	40	2.0 Mbp

Table R7. The mean and standard deviation of the number of correctly assigned TADs to A and B compartments for the 3D chromosome models calculated by GEM-FISH (which uses both Hi-C and FISH data) and GEM (which uses only Hi-C data), using the ground truth assignment inferred from Hi-C data and FISH data, respectively.

	Ground truth derived from Hi-C data		Ground truth derived from FISH data	
	GEM-FISH	GEM	GEM-FISH	GEM
Chr20 (30 TADs)	25.95 \pm 0.71	23.19 \pm 0.41	26.89 \pm 0.72	22.21 \pm 0.41
Chr21 (34 TADs)	29.96 \pm 0.24	30.01 \pm 0.10	31.92 \pm 0.46	29.99 \pm 0.10
Chr22 (27 TADs)	21.14 \pm 0.80	18.51 \pm 0.68	21.86 \pm 1.25	20.51 \pm 0.68

Table R8. Average relative enrichment of active and repressive marks, and unmodified histone H3 in the regions belonging to B1 and B2 subcompartments, according to the annotation method adopted from [10]. The regions belonging to subcompartment B1 are enriched with repressive marks (shown in bold) and depleted from other marks. On the other hand, the regions belonging to subcompartment B2 are enriched with unmodified histone H3 (shown in bold) and depleted from other marks.

	Average relative enrichment for H3K36me3			Average relative enrichment for H3K4me2			Average relative enrichment for H3K27me3			Average relative enrichment for unmodified histones H3		
	Chr20	Chr21	Chr22	Chr20	Chr21	Chr22	Chr20	Chr21	Chr22	Chr20	Chr21	Chr22
B1	-0.65	-0.808	-0.585	-0.48	-0.389	-0.22	2.607	2.412	2.646	-0.47	-0.214	-0.839
B2	-0.588	-0.731	-0.308	-1.152	-0.852	-0.751	-1.099	-0.751	-1.048	3.836	3.336	2.699

Table R9. Relative enrichments of the four histone marks H3K27ac, H3K4me3, H3K4me1, and H3K9me3 in individual TADs of Chr20. The first column gives the TAD IDs. Starting from the second column, the relative enrichments of different histone marks are shown. The last column shows the compartment type (A or B) to which the corresponding TAD belongs based on the experimental FISH data.

TAD ID	H3K27ac	H3K4me3	H3K4me1	H3K9me3	Compartment type
1	1.2533	1.0769	2.3423	0.1792	B
2	1.0997	1.3349	1.9609	0.044422	A
3	1.5672	1.7151	1.2254	0.10597	A
4	0.25724	0.28628	0.16611	1.9484	B
5	1.3515	1.5201	0.4258	0.1678	B
6	1.4085	1.4533	0.75819	0.13018	A
7	0.21047	0.26546	0.18646	2.4291	B
8	2.3675	2.0946	2.0705	0.10707	A
9	3.678	2.7348	2.6779	0.1628	B
10	0.2554	0.29176	0.39154	2.3938	B
11	0.28823	0.46292	0.061763	0.36297	B
12	0.92925	1.0228	1.5356	0.27981	B
13	1.4302	1.4559	2.3535	0	A
14	2.5531	1.9316	1.914	0	A
15	2.1441	1.8826	2.8375	0.030241	A
16	1.1463	1.4809	2.9747	0.067632	A
17	1.983	1.9689	2.4053	0.08052	A
18	0.27708	0.38227	0.37573	1.036	B
19	0.043236	0.058956	0.041513	3.6944	B
20	1.4638	1.5144	1.5367	0.020581	B
21	1.7681	1.9125	0.74647	0.2581	A
22	0.64117	0.81788	0.67933	0.61837	B
23	4.6047	3.7281	2.357	0.014197	A
24	1.5525	2.4961	0.57214	0.082788	A
25	0.32346	0.34631	0.29235	1.6904	B
26	0.34689	0.48343	0.27621	1.3976	B
27	1.3685	0.76515	0.79919	0.75668	B
28	0.83384	1.0979	1.3842	0.94582	B
29	1.4894	1.9201	4.3024	0.64394	B
30	0.86723	1.2888	1.8641	0.25342	A

Table R10. Relative enrichments of the four histone marks H3K27ac, H3K4me3, H3K4me1, and H3K9me3 in individual TADs of Chr21. The first column gives the TAD IDs. Starting from the second column, the relative enrichments of different histone marks are shown. The last column shows the compartment type (A or B) to which the corresponding TAD belongs based on the experimental FISH data.

TAD ID	H3K27ac	H3K4me3	H3K4me1	H3K9me3	Compartment type
1	0.066121	0.11234	0.11188	0	B
2	0.35199	0.3326	0.42734	0.7882	B
3	0.42806	0.45713	0.45325	4.6223	B
4	0.035074	0.037609	0.029822	0	B
5	0.22153	0.2023	0.10736	3.4503	B
6	0.22973	0.24349	0.061523	1.9897	B
7	0.71911	0.73595	0.65694	1.1378	B
8	1.7157	1.1834	1.5848	0.096008	A
9	2.5705	1.3378	0.77653	0.078021	A
10	3.9996	2.6775	2.6235	0.14027	B
11	0.18502	0.22066	0.057273	2.0196	B
12	0.43046	0.7076	0.69793	0.1196	B
13	1.003	1.2654	1.5061	0.038453	B
14	1.608	1.5698	2.7392	0.044627	B
15	0.81681	0.97178	2.2287	0.072702	B
16	1.6472	1.7203	2.3571	0.066515	A
17	1.1825	1.5195	1.3719	0.22553	A
18	1.8373	1.8173	1.9218	0.12838	A
19	1.9488	2.0044	1.329	0.091923	A
20	1.6917	1.7521	2.4646	0	A
21	1.3239	1.6015	1.6348	0.061662	A
22	1.9747	1.7593	2.8412	0.14462	A
23	0.83026	1.228	0.14867	0.17169	B
24	2.1349	1.9168	1.9892	0.087209	B
25	0.068909	0.10956	0.0705	2.2964	B
26	0.61415	1.1013	0.7304	0.086245	B
27	1.6713	2.0781	1.7716	0.067741	B
28	0.46854	1.0518	0.49014	0.17288	B
29	2.0593	2.3235	3.0439	0.064511	A
30	1.2481	1.7221	2.1419	0.025235	A
31	0.57811	0.91977	1.3115	0.82738	A
32	3.4785	4.2077	4.6165	0.17265	A
33	2.5004	3.1646	1.3252	0.17626	A
34	3.0405	2.1949	3.7568	0.17125	A

Table R11. Relative enrichments of the four histone marks H3K27ac, H3K4me3, H3K4me1, and H3K9me3 in individual TADs of Chr22. The first column gives the TAD IDs. Starting from the second column, the relative enrichments of different histone marks are shown. The last column shows the compartment type (A or B) to which the corresponding TAD belongs based on the experimental FISH data.

TAD ID	H3K27ac	H3K4me3	H3K4me1	H3K9me3	Compartment type
1	0.11987	0.14318	0.33149	0.019873	B
2	1.644	1.5354	1.3694	0	B
3	0.75268	0.87985	1.5888	0.19873	A
4	2.0136	2.1152	2.0241	0.059932	A
5	1.1341	0.98852	1.5944	0.01929	A
6	0.77154	0.97012	0.86268	0.14898	B
7	0.25244	0.34606	0.28414	2.2569	B
8	1.1984	1.2885	1.5038	0.10302	B
9	0.77795	1.0254	0.26341	0.21236	B
10	1.1169	1.1106	0.95507	0.10159	B
11	2.0458	1.5014	1.1554	0.15699	A
12	1.4218	1.2623	1.7387	0.078958	A
13	1.2035	1.0784	0.99836	0.10466	B
14	0.29694	0.37824	0.30716	3.4149	B
15	1.8809	1.758	0.91581	0.060577	A
16	7.4318	4.3269	2.1933	0.066531	A
17	0.34163	0.48826	0.77141	0.1093	A
18	1.3156	1.9216	0.61794	0.14545	A
19	1.7049	1.4703	2.3014	0.050093	A
20	1.0276	1.1462	1.3895	0.13844	A
21	1.8558	1.3486	2.0468	0.020299	A
22	0.65483	0.79087	1.329	0.16252	A
23	2.0579	2.1285	1.7475	0.053529	A
24	1.0103	1.2647	0.61599	0.35916	B
25	1.4445	1.4849	1.3342	0.24128	A
26	0.14466	0.23419	0.060766	5.4926	B
27	1.3022	1.319	2.4121	0.12267	A

Table R12. Mean and standard deviation values of average relative errors and asphericity values for the 3D models reconstructed by running GEM-FISH and GEM 100 times.

	Average relative error (for 100 runs)		Asphericity (for 100 runs)	
	GEM-FISH	GEM	GEM-FISH	GEM
Chr20	0.178 ± 0.002	0.317 ± 0.009	0.025 ± 0.003	0.187 ± 0.009
Chr21	0.162 ± 0.002	0.326 ± 0.032	0.024 ± 0.002	0.308 ± 0.018
Chr22	0.175 ± 0.003	0.223 ± 0.009	0.027 ± 0.003	0.103 ± 0.005
ChrXa	0.173 ± 0.001	1.341 ± 0.022	0.072 ± 0.002	0.256 ± 0.053
ChrXi	0.222 ± 0.001	1.981 ± 0.033	0.005 ± 0.001	0.256 ± 0.053

Table R13. Mean and standard deviation values of the numbers of correctly assigned TADs to A/B compartments in the 3D models generated by running GEM-FISH and GEM 100 times.

	Number of TADs	Number of TADs correctly assigned to A/B compartments (for 100 runs)	
		GEM-FISH	GEM
Chr20	30	26.89 ± 0.72	22.21 ± 0.41
Chr21	34	31.92 ± 0.46	29.99 ± 0.10
Chr22	27	21.86 ± 1.25	20.51 ± 0.68
ChrXa	40	38.96 ± 0.71	35.86 ± 1.15
ChrXi	40	36.20 ± 1.48	23.36 ± 0.89

Table R14. Mean and standard deviation values of average relative errors and the numbers of TADs correctly assigned to A/B compartments for an ensemble of four 3D models at TAD-level resolution computed by GEM-FISH (using the same optimization technique as in [11]) and GEM. In total, 100 trials were repeated.

	Number of TADs	Average relative error (for 100 runs)		Number of TADs correctly assigned to A/B compartments (for 100 runs)	
		GEM-FISH (ensemble)	GEM (ensemble)	GEM-FISH (ensemble)	GEM (ensemble)
Chr20	30	0.176 ± 0.003	0.283 ± 0.021	26.89 ± 0.65	23.22 ± 1.05
Chr21	34	0.159 ± 0.002	0.262 ± 0.023	32.80 ± 0.67	28.50 ± 0.51
Chr22	27	0.175 ± 0.002	0.221 ± 0.112	22.12 ± 1.32	21.96 ± 0.60
ChrXa	40	0.163 ± 0.007	1.192 ± 0.019	38.94 ± 0.57	39.45 ± 0.74
ChrXi	40	0.203 ± 0.009	1.783 ± 0.032	36.39 ± 1.32	20.93 ± 0.81

Table R15. Comparison between different modeling approaches, including GEM-FISH, MDS, Shrec3D, chromosome3D, and ChromSDE in terms of average relative errors and the number of TADs correctly assigned to A/B compartments. For Shrec3D, chromosome3D, and ChromSDE, due to the availability of only the non-allele-specific Hi-C maps (which do not distinguish between ChrXa and ChrXi) for the IMR90 cell line [6], we calculated only one 3D model for ChrX, and compared the assignment of its TADs to A/B compartments against those obtained by experimental FISH data for Chrs Xa and Xi, respectively. To avoid unfair comparison, we did not calculate the average relative errors with respect to experimental FISH distances for the 3D models derived by Shrec3D, chromosome3D, and ChromSDE.

	Average relative error		Number of TADs correctly assigned					Asphericity				
	GEM-FISH	MDS	GEM-FISH	chromosome3D	MDS	Shrec3D	chromSDE	GEM-FISH	chromosome3D	MDS	Shrec3D	chromSDE
Chr20	0.178	0.356	26.89	17	16	19	25	0.025	0.24	0.089	0.245	0.288
Chr21	0.162	0.312	31.92	28.4	32	17	29	0.024	0.66	0.055	0.106	0.413
Chr22	0.175	0.355	21.86	18.6	24	18	23	0.027	0.13	0.072	0.218	0.113
ChrXa	0.173	0.362	38.96	38.4	40	38	39	0.07	0.23	0.244	0.591	0.762
ChrXi	0.222	0.522	36.19	21.4	33	22	21	0.005	0.23	0.069	0.591	0.762

Table R16. Mean and standard deviations of average relative errors, number of TADs correctly assigned to A/B compartments, and asphericity values for the 3D models obtained by GEM-FISH and a modified version of GEM-FISH that uses only FISH data.

	Average relative error (for 10 runs)		Number of TADs correctly assigned to A/B compartments (for 10 runs)		Asphericity (for 10 runs)	
	GEM-FISH	FISH only	GEM-FISH	FISH only	GEM-FISH	FISH only
Chr20	0.177 ± 0.002	0.177 ± 0.001	3.0 ± 0.47	4.1 ± 1.59	0.023 ± 0.003	0.016 ± 0.003
Chr21	0.162 ± 0.002	0.162 ± 0.002	2.0 ± 0.0	2.5 ± 1.17	0.024 ± 0.001	0.023 ± 0.002
Chr22	0.175 ± 0.003	0.175 ± 0.001	5.0 ± 1.33	4.2 ± 1.13	0.026 ± 0.002	0.021 ± 0.005
ChrXa	0.173 ± 0.001	0.175 ± 0.002	1.0 ± 0.66	3.2 ± 1.47	0.068 ± 0.002	0.06 ± 0.006
ChrXi	0.222 ± 0.001	0.220 ± 0.001	3.5 ± 0.97	5.0 ± 3.36	0.005 ± 0.001	0.003 ± 0.001

Table R17. Average relative errors with respect to experimental FISH data for the final 3D models reconstructed by GEM-FISH for Chrs 20, 21, 22, Xa, and Xi, when using the highest, median, and mean values of the volumes of the chromosomes and their sizes to estimate the values of the radius of gyration of individual TADs, respectively.

	Using highest value	Using median value	Using mean value
Chr20	0.163	0.163	0.162
Chr21	0.143	0.139	0.137
Chr22	0.159	0.157	0.157
ChrXa	0.157	0.157	0.157
ChrXi	0.211	0.211	0.210

Table R18. Average overlapping fractions between adjacent TADs along the genomic distance belonging to the same compartment (i.e., with the same epigenetic nature), and those belonging to different compartments (i.e., with different epigenetic nature).

	Average overlapping fraction	
	TADs belonging to the same compartment	TADs belonging to different compartments
Chr21	0.15	0.06
Chr22	0.06	0.02

(a)

(b)

(c)

Figure R1. The 3D models reconstructed by GEM-FISH tended to be more spherical than those reconstructed by GEM. (a, b, and c) The TAD-level resolution models derived by GEM-FISH (left) and GEM (right) for Chrs 20, 21, and 22, respectively.

Figure R2. The expressed genes tend to lie closer to the chromosome surface in the 3D models reconstructed by GEM-FISH than in those reconstructed by GEM. *: p-value $< 10^{-20}$, Wilcoxon rank sum test. The expressed genes were defined as those with expression values > 20 (as in [3]).

Figure R3. The genomic sites that are known to interact with the NPC component Nup153 tend to lie closer to the chromosome surface in the 3D models reconstructed by GEM-FISH than the randomly selected loci of the same genomic lengths (denoted by Control-Nup). *: p-value $< 10^{-8}$, Wilcoxon rank sum test.

(a)

(b)

(c)

Figure R4. An experimental evidence showing that the genomic loci belonging to the same subcompartment tend to lie spatially closer than those belonging to different subcompartments. (a) The ChIP-Seq profiles of different histone marks for the three loci L1, L2, and L3 examined in the FISH experiment. Locus L3 is relatively enriched with the mark H3K27me3 and depleted from the mark H3K36me3, and hence considered belonging to subcompartment B1. On the other hand, the two loci L1 and L2 are depleted from the marks H3K27me3, H3K27ac, H3K36me3, H3K4me1, and H3K79me2, and hence considered belonging to subcompartment B2. (b) One example of the experimental FISH images for the two loci L1 and L2 (top), and for the two loci L1 and L3 (bottom). (c) The cumulative distribution function (CDF) curves of the distances between L1 and L2 and between L1 and L3, indicating the tendency of the two loci L1 and L2 (belonging to the same subcompartment B2) to lie spatially closer than the other two loci L1 and L3 (belonging to subcompartments B2 and B1, respectively), in spite of the slightly smaller genomic distance between L1 and L3 than between L1 and L2.

Figure R5. Visualization of the regions belonging to subcompartments B1 and B2 in Chr20. Along the genomic distance, subcompartment B1 is represented by colors ranging from cyan to blue, while subcompartment B2 is represented by colors ranging from magenta to red. The colors become darker with the increase in genomic distance. Although the regions ‘1’, ‘2’, ‘3’, ‘4’, and ‘5’ are far away in the genomic distance, they tend to colocalize in the 3D model. Similarly, the regions ‘6’ and ‘7’ also tend to colocate in the 3D space in spite of the large genomic distance between them (~10 Mbp). The visualization was performed mainly using UCSF Chimera [22].

Figure R6. Visualization of the regions belonging to subcompartments B1 and B2 in Chr21. Along the genomic distance, subcompartment B1 is represented by colors ranging from cyan to blue, while subcompartment B2 is represented by colors ranging from magenta to red. The colors become darker with the increase in genomic distance. The regions ‘1’, ‘2’, ‘3’, ‘4’, and ‘5’ tend to colocalize in the 3D model in spite of being far away in the genomic distance. The visualization was performed mainly using UCSF Chimera [22].

Figure R7. Visualization of the regions belonging to subcompartments B1 and B2 in Chr21. Along the genomic distance, subcompartment B1 is represented by colors ranging from cyan to blue, while subcompartment B2 is represented by colors ranging from magenta to red. The colors become darker with the increase in genomic distance. The regions ‘1’, and ‘2’, and the regions ‘3’, and ‘4’ tend to colocalize in the 3D model in spite of being far away in the genomic distance. The visualization was performed mainly using UCSF Chimera [22].

Figure R8. Super-enhancers tend to lie closer to the chromosome surface than regular enhancers. The modeling results of Chrs 20, 21, and 22 were used in the analysis. *: p-value $< 10^{-7}$, one-tailed Wilcoxon rank sum test.

Figure R9. Assignments of TADs to A/B compartments for Chrs 20, 21, and 22. (a, b, and c) Assignments of TADs for Chrs 20 (a), 21 (b), and 22 (c) using experimental FISH data (obtained from [7]). (d, e, and f) Assignment of TADs for Chrs 20 (d), 21 (e), and 22 (f) using Hi-C data (obtained from [6]).

Figure R10. Expression values of the genes in the region q22.3 of Chr21 are significantly higher than those in the other regions of the same chromosome (referred to as control genes). *: p -value = 0.048, one-tailed two-sample t -test.

Figure R11. The values of the scoring function for Chr22 with different parameter settings. (a) The values of the scoring function when fixing the value of $\lambda_E = 5 \times 10^{12}$ and changing the value of λ_F . (b) The values of the scoring function when fixing the value of $\lambda_F = 1 \times 10^{-8}$ and changing the value of λ_E .

(a)

(b)

(c)

(d)

(e)

(f)

Figure R12. The mean values of average relative errors when running GEM-FISH ten times with different parameter settings for Chrs 20, 21, and 22. (a, c, and e) The mean values of average relative errors when fixing the value of $\lambda_E = 5 \times 10^{12}$ and changing the value of λ_F for Chrs 20 (a), 21 (c), and 22 (e). (b, d, and f) The mean values of average relative errors when fixing the value of $\lambda_F = 1 \times 10^{-8}$ and changing the value of λ_E for Chrs 20 (b), 21 (d), and 22 (f).

(a)

(b)

(c)

(d)

(e)

(f)

Figure R13. The mean values of Pearson correlation coefficients between pairwise spatial distances between TADs derived from the 3D models reconstructed by GEM-FISH vs. inverse Hi-C contact frequencies when running GEM-FISH ten times with different parameter settings for Chrs 20, 21, and 22. (a, c, and e) The mean values of Pearson correlation coefficients when fixing the value of $\lambda_E = 5 \times 10^{12}$ and changing the value of λ_F for Chrs 20 (a), 21 (c), and 22 (e). (b, d, and f) The mean values of Pearson correlation coefficients when fixing the value of $\lambda_F = 1 \times 10^{-8}$ and changing the value of λ_E for Chrs 20 (b), 21 (d), and 22 (f).

(a)

(b)

(c)

(d)

(e)

(f)

Figure R14. The mean values of Pearson correlation coefficients between pairwise spatial distances between TADs derived from the 3D models reconstructed by GEM-FISH vs. experimental FISH distances when running GEM-FISH ten times with different parameter settings for Chrs 20, 21, and 22. (a, c, and e) The mean values of Pearson correlation coefficients when fixing the value of $\lambda_E = 5 \times 10^{12}$ and changing the value of λ_F for Chrs 20 (a), 21 (c), and 22 (e). (b, d, and f) The mean values of Pearson correlation coefficients when fixing the value of $\lambda_F = 1 \times 10^{-8}$ and changing the value of λ_E for Chrs 20 (b), 21 (d), and 22 (f).

(a)

(b)

(c)

(d)

(e)

(f)

Figure R15. The mean values of the harmonic means of Pearson correlation coefficients with both FISH distances and inverse Hi-C contact frequencies when running GEM-FISH ten times with different parameter settings for Chrs 20, 21, and 22. (a, c, and e) The mean values of the harmonic means when fixing the value of $\lambda_E = 5 \times 10^{12}$ and changing the value of λ_F for Chrs 20 (a), 21 (c), and 22 (e). (b, d, and f) The mean values of the harmonic means when fixing the value of $\lambda_F = 1 \times 10^{-8}$ and changing the value of λ_E for Chrs 20 (b), 21 (d), and 22 (f).

Figure R16. Examples of the 3D models of inactive (a, b, and c) and active TADs (d, e, and f). Blue and red colors denote inactive and active loci (of 5 Kbp size), respectively. Green color denotes the loci whose read count densities for all the four marks (i.e., H3K27ac, H3K4me1, H3K4me3, and H3K9me3) are less than the mean read count density of the whole chromosome.

(a)

(b)

(c)

(d)

(e)

(f)

Figure R17. The mean values of average relative errors when running GEM-FISH ten times with different parameter settings for Chrs 20, 21, and 22 using Hi-C data from [8] with resolution 40 Kbp. (a, c, and e) The mean values of average relative errors when fixing the value of $\lambda_E = 5 \times 10^{12}$ and changing the value of λ_F for Chrs 20 (a), 21 (c), and 22 (e). (b, d, and f) The mean values of average relative errors when fixing the value of $\lambda_F = 1 \times 10^{-8}$ and changing the value of λ_E for Chrs 20 (b), 21 (d), and 22 (f).

(a)

(b)

(c)

(d)

(e)

(f)

Figure R18. The mean values of Pearson correlation coefficients between the pairwise spatial distances between TADs obtained from the 3D models reconstructed by GEM-FISH vs. inverse Hi-C contact frequencies when running GEM-FISH ten times with different parameter settings for Chrs 20, 21, and 22 using Hi-C data from [8] with resolution 40 Kbp. (a, c, and e) The mean values of Pearson correlation coefficients when fixing the value of $\lambda_E = 5 \times 10^{12}$ and changing the value of λ_F for Chrs 20 (a), 21 (c), and 22 (e). (b, d, and f) The mean values of Pearson correlation coefficients when fixing the value of $\lambda_F = 1 \times 10^{-8}$ and changing the value of λ_E for Chrs 20 (b), 21 (d), and 22 (f).

(a)

(b)

(c)

(d)

(e)

(f)

Figure R19. The mean values of Pearson correlation coefficients between the pairwise spatial distances between TADs obtained from the 3D models reconstructed by GEM-FISH vs. experimental FISH distances when running GEM-FISH ten times with different parameter settings for Chrs 20, 21, and 22 using Hi-C data from [8] with resolution 40 Kbp. (a, c, and e) The mean values of Pearson correlation coefficients when fixing the value of $\lambda_E = 5 \times 10^{12}$ and changing the value of λ_F for Chrs 20 (a), 21 (c), and 22 (e). (b, d, and f) The mean values of Pearson correlation coefficients when fixing the value of $\lambda_F = 1 \times 10^{-8}$ and changing the value of λ_E for Chrs 20 (b), 21 (d), and 22 (f).

(a)

(b)

(c)

(d)

(e)

(f)

Figure R20. The mean values of the harmonic means of Pearson correlation coefficients with both FISH distances and inverse Hi-C contact frequencies when running GEM-FISH ten times with different parameter settings for Chrs 20, 21, and 22 using Hi-C data from [8] with resolution 40 Kbp. (a, c, and e) The mean values of the harmonic means when fixing the value of $\lambda_E = 5 \times 10^{12}$ and changing the value of λ_F for Chr 20 (a), 21 (c), and 22 (e). (b, d, and f) The mean values of the harmonic means when fixing the value of $\lambda_F = 1 \times 10^{-8}$ and changing the value of λ_E for Chrs 20 (b), 21 (d), and 22 (f).

Figure R21. The scatter plots and the corresponding fitting curves of spatial vs. genomic distances between TADs for Chr20, which were derived from the experimental FISH data (a), and the final 3D models reconstructed by GEM-FISH (b), and by GEM (c), respectively.

Figure R22. The scatter plots and the corresponding fitting curves of spatial vs. genomic distances between TADs for Chr21, which were derived from the experimental FISH data (a), and the final 3D models reconstructed by GEM-FISH (b), and by GEM (c), respectively.

Figure R23. The scatter plots and the corresponding fitting curves of spatial vs. genomic distances between TADs for Chr22, which were derived from the experimental FISH data (a), and the final 3D models reconstructed by GEM-FISH (b), and by GEM (c), respectively.

Figure R24. The scatter plots and the corresponding fitting curves of spatial vs. genomic distances between TADs for ChrXa, which were derived from the experimental FISH data (a), and the final 3D models reconstructed by GEM-FISH (b), and by GEM (c), respectively.

Figure R25. The scatter plots and the corresponding fitting curves of spatial vs. genomic distances between TADs for ChrXi, which were derived from the experimental FISH data (a), and the final 3D models reconstructed by GEM-FISH (b), and by GEM (c), respectively.

Figure R26. Pearson correlation coefficients between the inverse of Hi-C contact frequencies and the corresponding spatial distances derived from the 3D models reconstructed by GEM-FISH and a modified version of GEM-FISH that used only FISH data for different chromosomes. ***: p -value $< 10^{-13}$, **: p -value $< 10^{-9}$, *: p -value $< 10^{-7}$, Wilcoxon rank sum test.

Figure R27. The agreement between the heterogeneity of individual chromosomes measured by FISH experiments and the average modeling results derived by GEM-FISH for Chr20. Top: the histograms showing the distribution of the number of structures within individual 10 groups classified according to the volumes of the minimum bounding boxes enclosing the spatial positions of TADs measured in FISH experiments. Bottom: the average relative errors obtained by comparing the average distance matrix of each corresponding group with the average 3D model reconstructed by GEM-FISH.

Figure R28. The agreement between the heterogeneity of individual chromosomes measured by FISH experiments and the average modeling results derived by GEM-FISH for Chr21. Top: the histograms showing the distribution of the number of structures within individual 10 groups classified according to the volumes of the minimum bounding boxes enclosing the spatial positions of TADs measured in FISH experiments. Bottom: the average relative errors obtained by comparing the average distance matrix of each corresponding group with the average 3D model reconstructed by GEM-FISH.

Figure R29. The agreement between the heterogeneity of individual chromosomes measured by FISH experiments and the average modeling results derived by GEM-FISH for Chr22. Top: the histograms showing the distribution of the number of structures within individual 10 groups classified according to the volumes of the minimum bounding boxes enclosing the spatial positions of TADs measured in FISH experiments. Bottom: the average relative errors obtained by comparing the average distance matrix of each corresponding group with the average 3D model reconstructed by GEM-FISH.

Figure R30. The agreement between the heterogeneity of individual chromosomes measured by FISH experiments and the average modeling results derived by GEM-FISH for ChrXa. Top: the histograms showing the distribution of the number of structures within individual 10 groups classified according to the volumes of the minimum bounding boxes enclosing the spatial positions of TADs measured in FISH experiments. Bottom: the average relative errors obtained by comparing the average distance matrix of each corresponding group with the average 3D model reconstructed by GEM-FISH.

Figure R31. The agreement between the heterogeneity of individual chromosomes measured by FISH experiments and the average modeling results derived by GEM-FISH for ChrXi. Top: the histograms showing the distribution of the number of structures within individual 10 groups classified according to the volumes of the minimum bounding boxes enclosing the spatial positions of TADs measured in FISH experiments. Bottom: the average relative errors obtained by comparing the average distance matrix of each corresponding group with the average 3D model reconstructed by GEM-FISH.

Figure R32. Super-enhancers tend to lie closer to the chromosome surface than randomly selected control loci (denoted as “Control-SE”) of the same genomic lengths . The modeling results of Chrs 20, 21, and 22 were used in the analysis. *: p-value < 10^{-5} , one-tailed Wilcoxon rank sum test.

Reviewers' comments:

Reviewer #1 (Remarks to the Author):

The authors have addressed most of my previous comments/concerns. I have only a few remaining suggestions.

- Re: Minor Comments 3

The authors provided a summary of TADs used in the study in Table R6. For chromosomes 20 and Xa/Xi, only a subset of TADs spanning at regular intervals was imaged. In that case, how did the authors handle the loci that were not included in imaged TADs in their modeling?

- Re: Minor Comment 4

The authors provided the genomic maps to show the genomic locations of subcompartments; visualization of the regions belonging to subcompartments B1 and B2 showed that loci within the same subcompartment tend to colocalize even if they are far away along the genomic distance.

* Figures R6 and R7 have the same title "Visualization of the regions belonging to subcompartments B1 and B2 in Chr21" but different images.

* In addition to visual inspection of the subcompartment colocalization, please provide some statistics about the genomic distance and spatial distance of loci in 3D models between same and different subcompartments.

Reviewer #2 (Remarks to the Author):

The authors have addressed all my comments. My only comment is about R12 and R13. Because of the strong spatial dependence in HiC data, these metrics might need to be computed in a distance-stratified way. Can the authors comment what happens in each distance bin?

Reviewer #3 (Remarks to the Author):

Transparency statement: this review was written by Guillaume Fillion (CRG, Barcelona). I was offered to review the updated manuscript on October 25, 2018. I asked 15 days to do the task instead of the usual 10 days. I worked approximately 4 hours on the review and turned it in on November 13, 2018.

In this much improved version of the manuscript, Abbas et al. have consolidated their results and they have responded to most of the requests from the reviewers. The principal improvements are the following:

They added a benchmark against other popular methods to reconstruct genome structure (section 2.4).

They use new FISH data (in section 2.5).

They compare super-enhancers to typical enhancers (section 2.7)

They added a section about dynamics and uncertainty of the models (section 3.2).

They have considerably expanded the methods section (S1).

The authors have put a lot of care in addressing the issues of the reviewers and they deserve acknowledgement for taking the comments seriously. The points above greatly contribute to increasing the quality of their manuscript. They give appropriate answers to almost all the comments (as far as I can understand the points of the other reviewers), but I would still like to raise some issues that deserve discussion.

1. The authors do not respond to an important question raised by reviewer #2 (major point 1), namely how sensitive GEM-FISH is to TAD calling. It is important for users to have this

information.

In the third paragraph of the answer to my point 4, the authors write "... in agreement with the fact that active transcription regions tend to lie closer to the chromosome surface [3]". The authors of reference [3] describe a structural model where active genes are at the periphery of the chromosome territory. To conclude the paragraph, the authors of reference [3] write "These observations are consistent with the findings of prior studies using both microscopy and Hi-C (9, 29, 30)". Reference (9) describes high resolution Hi-C data, (29) is a microscopy study in *Drosophila* and (30) is a review. None of them shows direct experimental evidence that active genes are on the outside of the chromosome territories.

I am aware of experimental evidence that escapees on the inactive X tend to lie outside the chromosome territory / Barr body (Chaumeil et al. 2006, A novel role for Xist RNA in the formation of a repressive nuclear compartment into which genes are recruited when silenced, *Genes Dev.*;20(16):2223-37), but for autosomes I do not know whether this is the case. Indeed, the vast majority of silent genes are in LADs (lamin-associated domains), i.e. at the periphery of the nucleus (see for instance: Guelen et al. 2008, Domain organization of human chromosomes revealed by mapping of nuclear lamina interactions, *Nature* 453(7197):948-51). It is hard to imagine how this is possible in a model where inactive genes are more on the inside of the chromosome territory than active genes.

2. The authors should clarify the nature of the evidence they refer to, and how it is compatible with other well known and established properties of gene distribution. Alternatively (or in addition), they can measure the relative position of escapees on the inactive X in their models.

In the rest of the answer, I admire the determination of the authors to test their model by performing FISH experiment. But I do not see how this constitutes a test of GEM-FISH. Indeed, the choice of the sites L1, L2 and L3 is not motivated by GEM-FISH (it is based on epigenetic marks), and the authors do not show the distances between the sites in their models. Even if the authors compared the distributions of distances in FISH and in their resampled model, it would still remain $n = 1$ observation without any statistical weight.

3. The authors should motivate the choice of L1, L2 and L3 from their model (assuming that the distance from L1 to L3 is greater than the distance from L1 to L2 in their structure) and compare the experimental FISH distances with distances in multiple simulated structures. In theory, the authors should test the validity of their model on a sample greater than $n = 1$ data point, but since their cross-validation analysis suggests that the GEM-FISH model is able to predict held out FISH data, they may justify that more experiments would not bring so much new insight.

I strongly advise the authors to address points 1, 2 and 3 before publication. The points are important but their implementation is relatively straightforward, so I do not request to see the updated manuscript.

Reviewer #4 (Remarks to the Author):

The authors thoroughly addressed all my previous comments. I only have two additional minor comments that are very easy to address, albeit important for the clarity of the paper:

1) In my previous report, I suggested in Comment 2 that the authors could compare the 3D conformations of single chromosomes measured by FISH with the averaged 3D model from the computation. The authors' attempt to address this comment is over-complicated and may confuse the readers. The authors grouped measured chromosome copies based on their spatial volumes, and then compared the averaged conformation of each group with the 3D model, and concluded that the N number of each group largely determines the similarity between the averaged group conformation and the 3D model. This basically indicates that the individual chromosome

conformation could have some level of deviation from the 3D model. I suggest the authors to include a supplementary figure plotting the 3D model conformation (as in Figure S10) and the FISH conformations of several randomly picked copies of the same chromosome. That would be a much more direct and clear visualization of how much individual chromosomes may deviate from the average conformation.

2) The authors referred to the Xa and Xi compartments as A-B compartments, such as in the caption of Table 2. This will mislead readers to assume that these X compartments correspond to open and close chromatin regions, as is associated with A-B compartments in the autosomes. I suggest the author to use alternative terminologies for the Xa- and Xi-specific compartments.

Reviewer 1**General comments**

The authors have addressed most of my previous comments/concerns. I have only a few remaining suggestions.

Response: Thank you for your efforts in reviewing our response and revised manuscript. Below please find our point-to-point response to your comments.

Comments

Re: Minor Comments 3: The authors provided a summary of TADs used in the study in Table R6. For chromosomes 20 and Xa/Xi, only a subset of TADs spanning at regular intervals was imaged. In that case, how did the authors handle the loci that were not included in imaged TADs in their modeling?

Response: Thank you for the comment. As illustrated in our manuscript, we are proposing a divide-and-conquer based method for modeling the 3D organizations of chromosomes. We first calculate a TAD-level resolution model, which acts as the backbone for the final structure. After that, we integrate the high-resolution 3D models of individual TADs to the initially calculated backbone structure. For Chrs 20, Xa, and Xi, since the imaged TADs span regular intervals of the chromosomes, we only used those imaged TADs to build the TAD-level resolution conformations (i.e., backbone structures). In addition, we calculated the high-resolution models of only those imaged TADs and integrated them to the TAD-level resolution 3D models to obtain the final 3D models.

To conclude our answer, as long as the imaged TADs span regular intervals of the chromosome, they are sufficient to build a backbone structure for the chromosome. Integrating this backbone structure with the high-resolution models of the imaged TADs will result in a reasonably accurate final 3D model.

Re: Minor Comment 4: The authors provided the genomic maps to show the genomic locations of subcompartments; visualization of the regions belonging to subcompartments B1 and B2 showed that loci within the same subcompartment tend to colocalize even if they are far away along the genomic distance.

Figures R6 and R7 have the same title Visualization of the regions belonging to subcompartments B1 and B2 in Chr21 but different images.

Response: Thank you for pointing this out. In fact, Fig. R7 was for Chr22. In the revised manuscript, we have corrected this typo. More specifically, we have corrected the caption of Fig. S22 (in the current revised manuscript) and clarified that it is relevant to Chr22.

In addition to visual inspection of the subcompartment colocalization, please provide some statistics about the genomic distance and spatial distance of loci in 3D models between same and different subcompartments.

Response: Thank you for the perspective suggestion. Following your suggestion, we compared the spatial distances between loci belonging to different subcompartments. We found that loci that belong to the same subcompartment are spatially closer to each other than loci that belong to different subcompartments, even when the genomic distances

between loci belonging to the same subcompartment are larger than those from different ones (Fig. R1-R3). We have also added these results into the revised manuscript (please refer to the highlighted parts in Section 2.5).

Reviewer 2**General comments**

The authors have addressed all my comments.

Response: Thank you for your efforts in reviewing our response and manuscript. Below please find our response to your comment.

Comment 1: My only comment is about R12 and R13. Because of the strong spatial dependence in HiC data, these metrics might need to be computed in a distance-stratified way. Can the authors comment what happens in each distance bin?

Response: Thank you for the comment. Here, we chose the values of the two parameters $\lambda_E = 5 \times 10^{12}$ and $\lambda_F = 10^{-8}$ because these values produced the highest scores of the scoring function (Eq. 18 in the manuscript) for Chr22 and reasonably good scores for Chr21 (as illustrated in Fig. R4 and in Section 5.5 in the manuscript).

Following your suggestion, we divided the experimental FISH distances equally into three bins (hereafter called small, medium, and large distances), starting from the minimum distance to the maximum one for each of the three autosomes 20, 21, and 22. We found that only a small portion (10%-15%) of the distances belong to the small range of distances (Table R1). We also examined the distributions of Hi-C contact frequencies for the three ranges of distances for the three autosomes. We found that Hi-C contact frequencies are generally high for small spatial distances, and low for medium and large spatial distances (Fig. R5). Thus, the Hi-C contact frequency restraints play a major role in calculating the output 3D model in small distances than in medium and large ones, in which the experimental FISH restraints play a more important role and compensate the relatively weak restraints of the Hi-C frequencies. This can be reflected by the high Pearson correlation coefficients between the inter-TAD distances in the reconstructed 3D models and the inverse Hi-C contact frequencies in case of small distances, and the small ones in case of medium and large distances for both $\lambda_E = 5 \times 10^{12}$ and $\lambda_F = 10^{-8}$ (Fig. R6-R8). Similarly, we can notice the small average relative errors with respect to FISH distances in case of the medium and large distances, and the higher one (yet still reasonable) in case of small distances (Fig. R9-R11). These results were in line with the small average relative errors and high Pearson correlation coefficients obtained for the parameter setting $\lambda_E = 5 \times 10^{12}$ and $\lambda_F = 10^{-8}$ when considering all ranges of distances together (Fig. R12 and R13).

Table R1. The ranges of the small, medium, and large distances (in μm) and their percentages in FISH data for Chrs 20, 21, and 22, respectively.

	Small distances		Medium distances		Large distances	
	Ranges (μm)	Percentages	Ranges (μm)	Percentages	Ranges (μm)	Percentages
Chr20	0.69-1.15	10.10%	1.15-1.60	47.20%	1.60-2.06	42.70%
Chr21	0.45-0.92	13.40%	0.92-1.39	59.80%	1.39-1.86	26.80%
Chr22	0.49-0.96	7.70%	0.96-1.43	60.30%	1.43-1.89	32.00%

Reviewer 3

General comments

Transparency statement: this review was written by Guillaume Filion (CRG, Barcelona). I was offered to review the updated manuscript on October 25, 2018. I asked 15 days to do the task instead of the usual 10 days. I worked approximately 4 hours on the review and turned it in on November 13, 2018.

In this much improved version of the manuscript, Abbas et al. have consolidated their results and they have responded to most of the requests from the reviewers. The principal improvements are the following:

They added a benchmark against other popular methods to reconstruct genome structure (section 2.4).

They use new FISH data (in section 2.5).

They compare super-enhancers to typical enhancers (section 2.7)

They added a section about dynamics and uncertainty of the models (section 3.2).

They have considerably expanded the methods section (S1).

The authors have put a lot of care in addressing the issues of the reviewers and they deserve acknowledgement for taking the comments seriously. The points above greatly contribute to increasing the quality of their manuscript. They give appropriate answers to almost all the comments (as far as I can understand the points of the other reviewers), but I would still like to raise some issues that deserve discussion.

Response: Thank you for your efforts in reviewing our response and revised manuscript, and many thanks for the encouraging comments. Below please find our point-to-point response to your comments.

Comment 1: The authors do not respond to an important question raised by reviewer 2 (major point 1), namely how sensitive GEM-FISH is to TAD calling. It is important for users to have this information.

Response: Thank you for the comment. In [1], the authors used the genomic locations of the TADs derived from the Hi-C data in [2] and imaged the central 100 Kbp regions of these TADs. Since we used the FISH data obtained from [1], here we used the same TAD partition.

We argue that GEM-FISH is not sensitive to that TAD calling method. The TAD-level resolution model reconstructed by GEM-FISH in the first step acts as the backbone for the final chromosome model, in which each 3D point corresponds to an imaged TAD. Thus, as long as the genomic locations of the TADs identified by the new TAD calling method are imaged, GEM-FISH should be able to build an accurate backbone model. In addition, the TADs identified by any calling method generally contain a sufficient number of Hi-C contact frequencies for GEM-FISH to reconstruct the high-resolution 3D models of individual TADs. Therefore, we believe that the modeling performance of GEM-FISH is robust to different TAD calling methods.

We have added this point in the Discussion section of the revised manuscript (please refer to the highlighted part in Section 3.2).

Comment 2: In the third paragraph of the answer to my point 4, the authors write ... in agreement with the fact that active transcription regions tend to lie closer to the chromosome surface [3]. The authors of reference [3] describe a structural model where active genes are at the periphery of the chromosome territory. To conclude the paragraph, the authors of reference [3] write These observations are consistent with the findings of prior studies using both microscopy and Hi-C (9, 29, 30). Reference (9) describes high resolution Hi-C data, (29) is a microscopy study in *Drosophila* and (30) is a review. None of them shows direct experimental evidence that active genes are on the outside of the chromosome territories.

I am aware of experimental evidence that escapees on the inactive X tend to lie outside the chromosome territory / Barr body (Chaumeil et al. 2006, A novel role for Xist RNA in the formation of a repressive nuclear compartment into which genes are recruited when silenced, *Genes Dev.*;20(16):2223-37), but for autosomes I do not know whether this is the case. Indeed, the vast majority of silent genes are in LADs (lamin-associated domains), i.e. at the periphery of the nucleus (see for instance: Guelen et al. 2008, Domain organization of human chromosomes revealed by mapping of nuclear lamina interactions, *Nature* 453(7197):948-51). It is hard to imagine how this is possible in a model where inactive genes are more on the inside of the chromosome territory than active genes.

2- The authors should clarify the nature of the evidence they refer to, and how it is compatible with other well known and established properties of gene distribution. Alternatively (or in addition), they can measure the relative position of escapees on the inactive X in their models.

Response: Thank you for the comment. When we referred to Di Pierro 2016 [3] (reference 3 in our previous response), we did not check the references 9, 29, and 30 in it. We should have been more careful concerning this point, and we are sorry about that.

As for the distribution of genes inside the chromosome, please refer to Stevens et al. 2017 [4]. In [4], the authors have shown that the chromosomes generally pack together to form an outer ring belonging to compartment B. In addition, the chromosomes form an inner ring belonging to compartment A and surrounding an interior ring belonging to compartment B which in turn encircles the nucleolus (Fig. 2(e, left) in [4]). Also, they have shown that the highly expressed genes prefer to lie within the inner ring belonging to compartment A, and the constitutive lamina-associated domains usually lie in the compartment B regions (Fig. 2(e, middle) in [4]). Moreover, they have shown that highly expressed genes tend to lie closer to the surface of the chromosomes and within compartment A ring (Fig. 3(c) in [4]). Although in [4], the authors mainly studied mouse embryonic stem (ES) cells, we believe that human chromosomes should display similar behavior, given that, in general, the systems controlling gene activity in the two species have many similarities [5]. We also believe that this behavior is compatible with the well-known properties of gene distributions as described in your comment.

In addition, looking into the literature, we have found that active genes usually lie on the surface of the chromosome territories or the surface of the condensed chromatin

subdomains, and certain active genes may even reside on the large loops extending out of the chromosome territories (for review, see [6]).

We have also looked into the list of genes that escape the inactivation in ChrXi [7], and we found that only five of them are both active in the IMR90 cell line (with expression levels > 20 as described in [3]) and belonging to the imaged 40 TADs of ChrXi in [1]. Thus, instead of checking the radial distribution of these escapees in the 3D model of ChrXi, we have inspected the radial distributions of the expressed genes of ChrX in the final 3D models reconstructed by GEM-FISH for both ChrXa (in which the genes are active) and ChrXi (in which the same set of genes are inactive, except for those that escape the X-inactivation). We found that the expressed genes tend to lie closer to the chromosome surface in ChrXa than in ChrXi (Fig. R14). This result agrees well with the previous finding that the silenced genes in ChrXi tend to be shifted into a more internal location within the XIST RNA compartment [8].

We have added the above results to the revised manuscript (please refer to the highlighted part in Section S1). In addition, in the revised manuscript, we have referred to Stevens et al. 2017 [4] and Williams 2003 [6] instead of Di Pierro 2016 [3] to show the evidence that active gene regions tend to lie closer to the chromosome surface (please refer to the highlighted part in Section 2.7).

Comment 3: In the rest of the answer, I admire the determination of the authors to test their model by performing FISH experiment. But I do not see how this constitutes a test of GEM-FISH. Indeed, the choice of the sites L1, L2 and L3 is not motivated by GEM-FISH (it is based on epigenetic marks), and the authors do not show the distances between the sites in their models. Even if the authors compared the distributions of distances in FISH and in their resampled model, it would still remain $n = 1$ observation without any statistical weight.

3. The authors should motivate the choice of L1, L2 and L3 from their model (assuming that the distance from L1 to L3 is greater than the distance from L1 to L2 in their structure) and compare the experimental FISH distances with distances in multiple simulated structures. In theory, the authors should test the validity of their model on a sample greater than $n = 1$ data point, but since their cross-validation analysis suggests that the GEM-FISH model is able to predict held out FISH data, they may justify that more experiments would not bring so much new insight.

Response: Thank you for the comment. In fact, the goal of this new FISH experiment was to provide another evidence that the regions belonging to the same subcompartment tend to be spatially closer than those belonging to different subcompartments. Thus, we tested the positions of three loci L1, L2, and L3, in which both L1 and L2 carry the epigenetic properties of subcompartment B2, while L3 carries those of subcompartment B1. We found that the spatial distance between L1-L2 was smaller than that between L1-L3, in spite of the larger genomic distance between L1-L2 compared to that between L1-L3.

We have further compared the distances between these three loci in the final 3D

models reconstructed by GEM-FISH with resolutions 25 Kbp and 50 Kbp (using Hi-C data from [9]), and 40 Kbp (using Hi-C data from [2]). We found that the distance between L1-L2 is consistently smaller than that between L1-L3 in all the models that we inspected with different resolutions. We have added this result in the revised manuscript (please refer to the highlighted part in the last paragraph of Section 2.5.)

Reviewer 4

General comments

The authors thoroughly addressed all my previous comments. I only have two additional minor comments that are very easy to address, albeit important for the clarity of the paper:

Response: Thank you for your efforts in reviewing our response and manuscript. Below please find our point-to-point response to your comments.

Minor comments

Comment 1: In my previous report, I suggested in Comment 2 that the authors could compare the 3D conformations of single chromosomes measured by FISH with the averaged 3D model from the computation. The authors attempt to address this comment is over-complicated and may confuse the readers. The authors grouped measured chromosome copies based on their spatial volumes, and then compared the averaged conformation of each group with the 3D model, and concluded that the N number of each group largely determines the similarity between the averaged group conformation and the 3D model. This basically indicates that the individual chromosome conformation could have some level of deviation from the 3D model. I suggest the authors to include a supplementary figure plotting the 3D model conformation (as in Figure S10) and the FISH conformations of several randomly picked copies of the same chromosome. That would be a much more direct and clear visualization of how much individual chromosomes may deviate from the average conformation.

Response: Thank you for the comment. Following your suggestion, for each of the five chromosomes, we plotted its average 3D model reconstructed by GEM-FISH, and then compared it visually with the FISH conformations of five randomly picked imaged copies (Fig. R15). This visual comparison demonstrated that, due to the dynamic nature of chromosomes, the corresponding conformations may show a certain level of deviation from each other and also from the average 3D model reconstructed by GEM-FISH. We have also added this result into the revised manuscript (please refer to the highlighted part in Section 3.2).

Comment 2: The authors referred to the Xa and Xi compartments as A-B compartments, such as in the caption of Table 2. This will mislead readers to assume that these X compartments correspond to open and close chromatin regions, as is associated with A-B compartments in the autosomes. I suggest the author to use alternative terminologies for the Xa- and Xi-specific compartments.

Response: Thank you for the comment. In the revised manuscript, we have emphasized that for Chrs Xa and Xi, the A/B compartments do not refer to active and inactive compartments as in the autosomes. Instead, they refer to the p and q arms in case of ChrXa, and to the genomic regions before and after the macrosatellite DXZ4 in case of ChrXi. Please refer to the highlighted part in Section 2.3 in the revised manuscript. We have also rephrased the corresponding sentences in Table 2 and other related tables.

References

1. Wang S, Su JH, Beliveau BJ, Bintu B, Moffitt JR, et al. (2016) Spatial organization of chromatin domains and compartments in single chromosomes. *Science* 353: 598–602.
2. Dixon JR, Selvaraj S, Yue F, Kim A, Li Y, et al. (2012) Topological domains in mammalian genomes identified by analysis of chromatin interactions. *Nature* 485: 376–380.
3. Di Pierro M, Zhang B, Aiden EL, Wolynes PG, Onuchic JN (2016) Transferable model for chromosome architecture. *Proceedings of the National Academy of Sciences* 113: 12168–12173.
4. Stevens TJ, Lando D, Basu S, Atkinson LP, Cao Y, et al. (2017) 3D structures of individual mammalian genomes studied by single-cell Hi-C. *Nature* 544: 59-64.
5. Yue F, Cheng Y, Breschi A, Vierstra J, Wu W, et al. (2014) A comparative encyclopedia of dna elements in the mouse genome. *Nature* 515: 355-364.
6. Williams RR (2003) Transcription and the territory: the ins and outs of gene positioning. *Trends in Genetics* 19: 298 - 302.
7. Carrel L, Willard HF (2005) X-inactivation profile reveals extensive variability in x-linked gene expression in females. *Nature* 434: 400-404.
8. Chaumeil J, Le Baccon P, Wutz A, Heard E (2006) A novel role for xist rna in the formation of a repressive nuclear compartment into which genes are recruited when silenced. *Genes and Development* 20: 2223-2237.
9. Rao S, Huntley M, Durand N, Stamenova E, Bochkov I, et al. (2014) A 3D map of the human genome at kilobase resolution reveals principles of chromatin looping. *Cell* 159: 1665–1680.

(a)

(b)

Figure R1. Genomic loci belonging to the same subcompartment are spatially closer than those belonging to different subcompartments in Chr20. (a and b) Spatial distances between loci in genomic regions ‘1’ and ‘2’ (which both belong to subcompartment B2) are smaller than those between loci in genomic regions ‘1’ and ‘3’ (which belong to subcompartments B2 and B1, respectively), although regions ‘1’ and ‘3’ are closer to each other than regions ‘1’ and ‘2’. *: negligible p-value, one-tailed Wilcoxon rank sum test.

(a)

(b)

Figure R2. Genomic loci belonging to the same subcompartment are spatially closer than those belonging to different subcompartments in Chr21. (a and b) Spatial distances between loci in genomic regions ‘1’ and ‘2’ (which both belong to subcompartment B2) are smaller than those between loci in genomic regions ‘1’ and ‘3’ (which belong to subcompartments B2 and B1, respectively), although regions ‘1’ and ‘3’ are closer to each other than regions ‘1’ and ‘2’. *: negligible p-value, one-tailed Wilcoxon rank sum test.

(a)

(b)

Figure R3. Genomic loci belonging to the same subcompartment are spatially closer than those belonging to different subcompartments in Chr22. (a and b) Spatial distances between loci in genomic regions ‘1’ and ‘2’ (which both belong to subcompartment B2) are smaller than those between loci in genomic regions ‘1’ and ‘3’ (which belong to subcompartments B2 and B1, respectively), although regions ‘1’ and ‘3’ are closer to each other than regions ‘1’ and ‘2’. *: negligible p-value, one-tailed Wilcoxon rank sum test.

Figure R4. The values of the scoring function for Chr22 with different parameter settings. (a) The values of the scoring function when fixing the value of $\lambda_E = 5 \times 10^{12}$ and changing the value of λ_F . (b) The values of the scoring function when fixing the value of $\lambda_F = 1 \times 10^{-8}$ and changing the value of λ_E .

Figure R5. The mean and standard deviation values for Hi-C contact frequencies for different ranges of distances for Chr20 (a), Chr21 (b), and Chr22 (c), respectively.

Figure R6. The mean values of Pearson correlation coefficients between different ranges of inter-TAD spatial distances derived from the 3D models reconstructed by GEM-FISH vs. the corresponding inverse Hi-C contact frequencies, when running GEM-FISH ten times with different parameter settings for Chr20. (a, b, and c) The mean values of Pearson correlation coefficients when fixing the value of $\lambda_E = 5 \times 10^{12}$ and changing the value of λ_F for small (a), medium (c), and large distances (e). (d, e, and f) The mean values of Pearson correlation coefficients when fixing the value of $\lambda_F = 1 \times 10^{-8}$ and changing the value of λ_E for small (d), medium (e), and large distances (f).

Figure R7. The mean values of Pearson correlation coefficients between different ranges of inter-TAD spatial distances derived from the 3D models reconstructed by GEM-FISH vs. the corresponding inverse Hi-C contact frequencies, when running GEM-FISH ten times with different parameter settings for Chr21. (a, b, and c) The mean values of Pearson correlation coefficients when fixing the value of $\lambda_E = 5 \times 10^{12}$ and changing the value of λ_F for small (a), medium (b), and large distances (c). (d, e, and f) The mean values of Pearson correlation coefficients when fixing the value of $\lambda_F = 1 \times 10^{-8}$ and changing the value of λ_E for small (d), medium (e), and large distances (f).

Figure R8. The mean values of Pearson correlation coefficients between different ranges of inter-TAD spatial distances derived from the 3D models reconstructed by GEM-FISH vs. the corresponding inverse Hi-C contact frequencies, when running GEM-FISH ten times with different parameter settings for Chr22. (a, b, and c) The mean values of Pearson correlation coefficients when fixing the value of $\lambda_E = 5 \times 10^{12}$ and changing the value of λ_F for small (a), medium (b), and large distances (c). (d, e, and f) The mean values of Pearson correlation coefficients when fixing the value of $\lambda_F = 1 \times 10^{-8}$ and changing the value of λ_E for small (d), medium (e), and large distances (f).

Figure R9. The mean values of average relative errors for small, medium, and large distances of Chr20, when running GEM-FISH ten times with different parameter settings. (a, b, and c) The mean values of average relative errors when fixing the value of $\lambda_E = 5 \times 10^{12}$ and changing the value of λ_F for small (a), medium (b), and large distances (c). (d, e, and f) The mean values of average relative errors when fixing the value of $\lambda_F = 1 \times 10^{-8}$ and changing the value of λ_E for small (b), medium (d), and large distances (f).

Figure R10. The mean values of average relative errors for small, medium, and large distances of Chr21, when running GEM-FISH ten times with different parameter settings. (a, b, and c) The mean values of average relative errors when fixing the value of $\lambda_E = 5 \times 10^{12}$ and changing the value of λ_F for small (a), medium (b), and large distances (c). (d, e, and f) The mean values of average relative errors when fixing the value of $\lambda_F = 1 \times 10^{-8}$ and changing the value of λ_E for small (b), medium (d), and large distances (f).

Figure R11. The mean values of average relative errors for small, medium, and large distances of Chr22, when running GEM-FISH ten times with different parameter settings. (a, b, and c) The mean values of average relative errors when fixing the value of $\lambda_E = 5 \times 10^{12}$ and changing the value of λ_F for small (a), medium (b), and large distances (c). (d, e, and f) The mean values of average relative errors when fixing the value of $\lambda_F = 1 \times 10^{-8}$ and changing the value of λ_E for small (b), medium (d), and large distances (f).

(a)

(b)

(c)

(d)

(e)

(f)

Figure R12. The mean values of average relative errors when running GEM-FISH ten times with different parameter settings for Chrs 20, 21, and 22. (a, c, and e) The mean values of average relative errors when fixing the value of $\lambda_E = 5 \times 10^{12}$ and changing the value of λ_F for Chrs 20 (a), 21 (c), and 22 (e). (b, d, and f) The mean values of average relative errors when fixing the value of $\lambda_F = 1 \times 10^{-8}$ and changing the value of λ_E for Chrs 20 (b), 21 (d), and 22 (f).

(a)

(b)

(c)

(d)

(e)

(f)

Figure R13. The mean values of Pearson correlation coefficients between pairwise spatial distances between TADs derived from the 3D models reconstructed by GEM-FISH vs. inverse Hi-C contact frequencies when running GEM-FISH ten times with different parameter settings for Chrs 20, 21, and 22. (a, c, and e) The mean values of Pearson correlation coefficients when fixing the value of $\lambda_E = 5 \times 10^{12}$ and changing the value of λ_F for Chrs 20 (a), 21 (c), and 22 (e). (b, d, and f) The mean values of Pearson correlation coefficients when fixing the value of $\lambda_F = 1 \times 10^{-8}$ and changing the value of λ_E for Chrs 20 (b), 21 (d), and 22 (f).

Figure R14. According to the final 3D models reconstructed by GEM-FISH, the expressed genes of ChrX tend to lie closer to the chromosome surface in ChrXa (in which the genes are active) than in ChrXi (in which the genes are inactive, except for those that escape the X-inactivation). *: p-value < 0.002, one-tailed Wilcoxon rank sum test.

(a)

(b)

(c)

(d)

Figure R15. FISH conformations of different imaged copies of the same chromosome show a certain level of deviation from each other and from the average 3D model reconstructed by GEM-FISH. (a-e) Plots of the average 3D model reconstructed by GEM-FISH (in magenta) and the FISH conformations of five randomly picked imaged copies (in cyan) of Chr20 (a), Chr21 (b), Chr22 (c), ChrXa (d), and ChrXi (e).